# Long-term observations of atmospheric $CO_2$ and $CH_4$ trends and comparison of two measurement systems at Pallas-Sammaltunturi station in Northern Finland

Antti Laitinen[1], Hermanni Aaltonen[1], Christoph Zellweger[2], Aki Tsuruta[1], Tuula Aalto[1], and Juha Hatakka[1]

[1]Finnish Meteorological Institute, Helsinki, Finland
[2]Empa, Swiss Federal Laboratories for Materials Science and Technology, Laboratory for Air Pollution/Environmental Technology, Duebendorf, Switzerland

**Correspondence:** Antti Laitinen (antti.laitinen@fmi.fi)

**Abstract.** Accurate and precise observations of atmospheric greenhouse gas mole fractions are crucial for understanding the carbon cycle. However, challenges can arise when comparing data between different observation sites, due to different measurement routines and data formats used. To combat these challenges, different research infrastructures have been established in order to harmonize measurement routines and data processing and to make the data from different stations readily available.

One of the few stations in the boreal region that observes atmospheric greenhouse gas mole fractions is the Pallas station, located atop Sammaltunturi fell in Finnish Lapland. The station's location above the arctic circle, far away from large settlements, makes it ideal for measurement of background mole fractions. The station hosts instrumentation for two different research infrastructures, Integrated Carbon Observation System (ICOS) and Global Atmosphere Watch (GAW), with completely independent measurement instruments, calibration standards and sampling systems. We present the long-term time series of

the mole fractions of $CO_2$ and $CH_4$ and their evolution measured at the station, as well as a long-term comparison of the two instruments during the period when both have been installed. We find that the average difference in the hourly values for $CO_2$ is <0.01 ppm and 0.47 ppb for $CH_4$. The trends and growth rated calculated for both instruments agree well. For a more detailed comparison, the ICOS and GAW systems were simultaneously audited by ICOS Mobile Laboratory and the World Calibration Centre (WCC-EMPA) of GAW, respectively. The audit results show good agreement between the different systems, with the

differences ranging from -0.06 ppm to 0.02 ppm for $CO_2$ and from -0.24 ppb to 0.30 ppb for $CH_4$. No significant dependence on mole fraction values was found for the differences between the systems. However, for one of the instruments we found a clear influence of sample drying, especially for $CH_4$. We also compared the long time series with the marine boundary layer (MBL) reference values, derived by NOAA based on the weekly air sample measurements, in the Northern Hemisphere. For $CO_2$, the values measured at Pallas are on average 1.9 ppm higher than the MBL for Northern Hemisphere, and 54 ppb higher

for $CH_4$. The difference is larger during summer for $CO_2$, but not significantly for $CH_4$.

# 1 Introduction

Accurate, long-term observations of the atmospheric greenhouse gases (GHGs) are important for predicting climate change, validating models and satellite observations, and for detecting changes in atmospheric composition. Especially in situ measurements of greenhouse gas mole fractions are needed for quantifying the long-term trends of the greenhouse gases, as well as annual and interannual variations. While remote sensing techniques can also be used for this purpose, only in situ measurements can be directly calibrated to the WMO scales for $CO_2$ and $CH_4$, and can be used to link the remote sensing observations to accepted scales (Byrne et al., 2023). They are also crucial for top-down emission estimates using atmospheric inverse models, which aim to optimize fluxes based on measured mole fractions (McGrath et al. (2023); Petrescu et al. (2023); Lauerwald et al. (2024); Saunois et al. (2024); Friedlingstein et al. (2025)). Together with the bottom-up estimates they give the best estimates for the emissions, crucial for policy makers to understand the sources and sinks of the greenhouse gases. In order to assure consistent measurement quality, WMO has defined compatibility goals that should be reached between different stations and laboratories (WMO, 2024).

One of the few stations conducting atmospheric observations in the boreal region is the station located on top of Sammaltunturi fell in the Pallas-Ylläsjärvi national park in Finland. The Pallas site is operated by the Finnish Meteorological Institute (FMI). The station hosts a wide variety of instruments ranging from meteorological measurements to greenhouse gases, aerosols and air quality observations.

The area's initial meteorological observations were conducted near Lake Pallasjärvi, commencing in 1931. Subsequently, in 1991, the measurement station atop Sammaltunturi began its operations. Starting in 1998, the first greenhouse gas measured at Sammaltunturi was carbon dioxide ($CO_2$). Over time, the measurement repertoire expanded to include methane ($CH_4$) in 2004, carbon monoxide (CO) in 2012 and nitrous oxide ($N_2O$) in 2022. In terms of the greenhouse gas measurements, the station is affiliated with two international measurement networks: the Global Atmosphere Watch (GAW) programme of World Meteorological Organization (WMO), and the European-wide Integrated Carbon Observation System (ICOS) (Heiskanen et al., 2022). Within the GAW network, the station is referred to as Pallas-Sammaltunturi (station id: PAL) and it reports data on $CO_2$ and $CH_4$. Meanwhile, under the ICOS network, the station is named Pallas (station id: PAL), and it provides data not only on $CO_2$ and $CH_4$ but also on CO and $N_2O$. This data is also submitted to the WMO World Data Center for Greenhouse Gases (WWDCGG), as ICOS is a contributing network to GAW. Contributing networks have signed a Letter of Agreement with WMO, detailing the list and characteristics of the stations to be included in the GAW network as contributing stations. The data from these stations is subsequently available through the GAW data portal.

More recently, the station has diversified its focus to encompass various features of atmospheric composition. Furthermore, it benefits from the support of multiple measurement sites dedicated to studying atmosphere-ecosystem interactions around the fell. Combined with the different atmosphere-ecosystem interactions stations, the measurement area of Pallas (Pallas supersite), including Sammaltunturi station, provide a comprehensive insight into the different processes and dynamics of the atmosphere and its interaction with ecosystems. An overview of the Pallas site is given in Hatakka et al. (2003), and up to date information

can be found on the FMI website [1]. While the term Pallas can, in a broader context, refer to the entire supersite, in this paper we use the term Pallas to refer to the atmosphere station atop Sammaltunturi.

The measurement networks ICOS and GAW both aim to achieve high accuracy and comparable observations of the atmospheric composition. While the GAW network focuses on a wider variety of atmospheric components and global coverage, ICOS aims to capture the entire carbon cycle. This includes atmospheric mole fraction observations of $CO_2$ and $CH_4$, as well as atmosphere-ecosystem interactions through observations of ecosystem fluxes and oceanic carbon. The ICOS Atmospheric Thematic Center (ICOS ATC) oversees the atmospheric measurements of the ICOS network. Within the ICOS ATC, the stations are classified into Class 1 and Class 2 stations (Yver-Kwok et al., 2021). The requirements for the Class 2 stations are continuous measurements of $CO_2$ and $CH_4$ complemented by basic meteorological parameters: Air temperature, relative humidity, wind speed and direction as well as atmospheric pressure. The Class 1 stations are required, in addition to the requirements of the Class 2, to have continuous CO and boundary layer height measurements as well as to operate the ICOS flask sampler (described more in detail in (Levin et al., 2020)). Furthermore, the stations are classified to three types based on their location: continental stations targeting mainly continental air-masses, coastal stations targeting mainly marine air-masses and mountain stations targeting mainly free troposphere during night (ICOS RI, 2020).

Assessing the compliance to the WMO network compatibility goals requires comparison of station measurements with other laboratory's measurements (Andrews et al., 2014). Such comparisons have been made with travelling cylinders (Zhou et al., 2009) and flask-sampling at the site (Levin et al., 2020). More recently, travelling instruments have been employed at stations to obtain consistent parallel measurements with good results (Hammer et al., 2013; WMO, 2013; Zellweger et al., 2016). While travelling cylinders can be used to ensure that the measurement scale is transferred correctly, they do not account for potential biases arising from the sampling system (WMO, 2024). With a co-located measurements with a travelling instrument, the whole sampling system can be evaluated. The ICOS ATC is composed of various components, including the ICOS Mobile Laboratory, which is tasked with this exact purpose: auditing the different atmospheric stations through parallel measurements and cross-comparisons. The Mobile Lab aims to ensure high quality and accuracy of the ICOS atmospheric measurements. A similar quality management framework exists for the WMO/GAW program. Central Calibration Laboratories (CCLs) maintain and distribute the calibration scales, and World and Regional Calibration Centres (WCCs/RCCs) ensure traceability through independent system and performance audits.

In this paper we give a detailed description of the WMO and ICOS setups used for the atmospheric greenhouse gas measurements at the Pallas station and presents trends, growth rates, seasonal and daily variations of the mole fractions as well as comparison of the two setups. We focus on $CO_2$ and $CH_4$, which are available from both the ICOS and GAW networks at the Pallas station. In addition, we explore the quality of the Pallas station measurements through comparisons of the two networks as well as the Mobile Laboratory audit and the GAW audit, which was conducted by the WCC for Surface Ozone, CO, $CH_4$ and $CO_2$ (WCC-Empa). We also show how the mole fractions of $CO_2$ and $CH_4$ at the Pallas station have evolved compared to the global trend in the Northern Hemisphere, and how well the two separate measurement systems compare over the long term.

---

[1]https//en.ilmatieteenlaitos.fi/pallas-atmosphere-ecosystem-supersite, last access: 07.10.2024

## 2 Measurement station

This section presents the details of the Pallas station, location, and instrumentation with a focus on greenhouse gas measurements.

### 2.1 Location

The Pallas station is located in the Pallas-Yllästunturi national park in Northern Finland, approximately 860 km from the capital city of Helsinki. The station is on top of a subartic round-topped mountain (Sammaltunturi) (Fig. 1), 566 m above sea level (ASL) and about 100 m above the tree line. It is above the boundary layer most of the time during the winter season and summer nights. On the fell, the vegetation is sparse and mostly consists of low vascular plants, moss and lichen. There are no large cities near the station, with the biggest town, Muonio (approximately 2000 inhabitants), about 20 km west and Kittilä (approximately 6500 inhabitants) about 50 km to south-east from the station. The region around Sammaltunturi has no significant local or regional sources of pollution. The Pallas region lies at the edge of the northern boreal and subartctic climate zones. Mean annual temperature atop the Sammaltunturi (1981-2010) is -1.0 °C, and the mean monthly temperatures vary from -14 °C in January to +14 °C in July. The lowest temperatures are usually measured in February and the highest in July, and the relative humidity is lowest in June and highest in November - January (Fig. 2 (a)). The prevailing wind direction atop Sammaltunturi is along the west-south axis (Fig. 2 (b)), with very little wind coming in from North. The mean wind speed (1996 - 2022) is 6.9 m/s ($\pm$ 0.5 m/s). The fell of Sammaltunturi is composed of mafic volcanic rock types, which provides a nutritient-rich soil on the fell slopes. The top of the Sammaltunturi fell is treeless and the treeline is mostly composed of Norway Spruce. Due to its remote location far away from any local pollution sources, the station measurements are representative for unpolluted background air (Hatakka et al., 2003), and it fulfills the requirements for ICOS Class 1 Mountain Atmosphere Station.

### 2.2 Instrumentation

During the last 25 years, greenhouse gas instrumentation has undergone substantial improvements in terms of of precision, measurement frequency and user-friendliness (Zellweger et al., 2016, 2019). The $CO_2$ measurements at Pallas began with a non-dispersive infrared (NDIR) analyzer in July 1998. For $CH_4$ measurements, a gas-chromatography (GC) based instrument was first used, starting in February 2004. Later, in January 2009, both instruments were replaced by a single cavity ring-down spectroscopy (CRDS) instrument capable of measuring both species simultaneously. These instruments were producing data for the GAW network, which was in 2017 supplemented by a separate CRDS instrument producing data for the ICOS network.

Today, the greenhouse gas measurements at Pallas for the GAW and ICOS networks are still completely independent, but both rely on the use of Picarro G2401 and Picarro G5310 (ICOS only) instruments. These commercially available CRDS instruments are capable of measuring dry mole fractions of $CO_2$ (G2401), $CH_4$ (G2401), CO (G2401 and G5310), $N_2O$ (G5310) and $H_2O$ (G2401 and G5310). To validate the instrument performance, both ICOS instruments have been tested in ICOS Atmosphere thematic centre (ATC) before being set up at the station. The Picarro G5310 was installed in 2022, adding $N_2O$ to the list of continuously measured components and at same time significantly improving CO measurement precision.

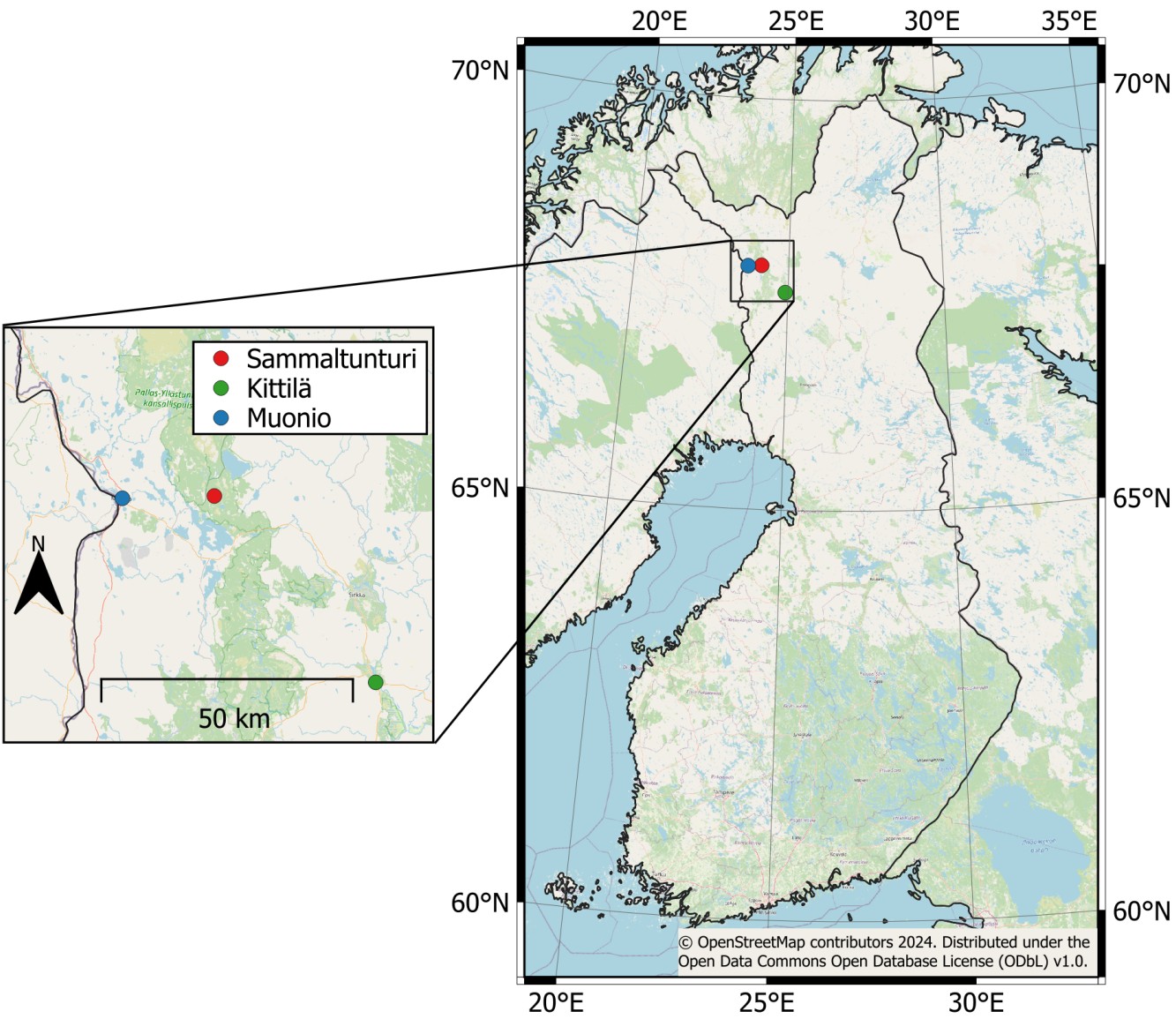

**Figure 1.** Location of Sammaltunturi in Finland. The location of Sammaltunturi in relation to the biggest municipalities Kittilä and Muonio are shown in the inset. © OpenStreetMap contributors 2024. Distributed under the Open Data Commons Open Database License (ODbL) v1.0.

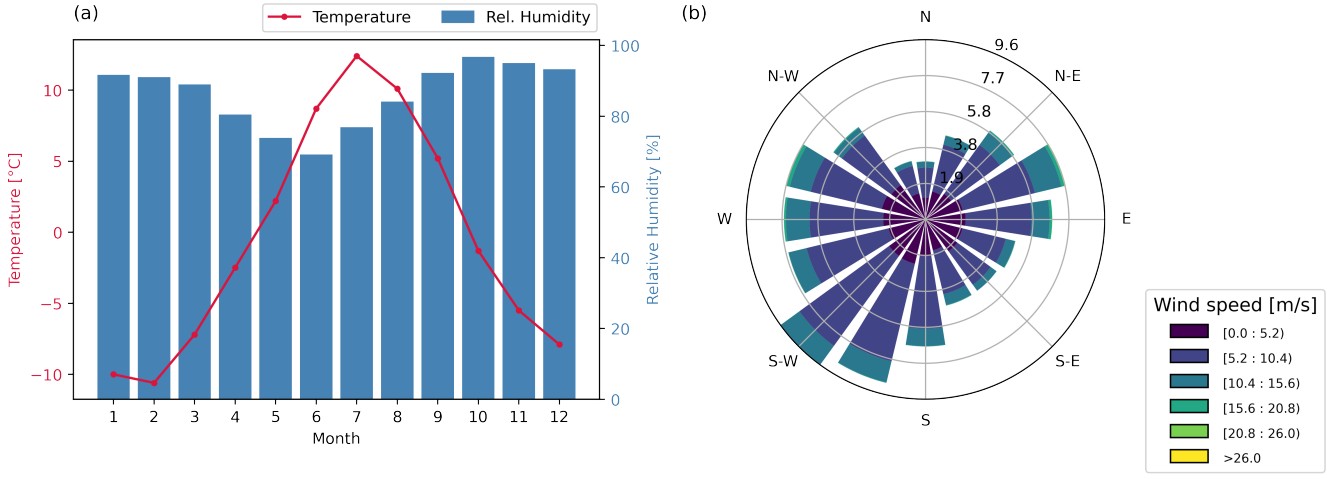

**Figure 2.** Monthly mean temperature and relative humidity (a) and windrose (b) of Sammaltunturi

Although CO is not a greenhouse gas, it is used as a proxy for emissions from anthropogenic sources. Since 2017, when ICOS measurements started at Pallas, ICOS specifications for atmosphere stations have been followed to meet the strict measurement compatibility goals set by the WMO. In Table 1, the network compatibility goals (the maximum bias between different datasets tolerable when measuring well-mixed background air) and the measurement ranges are presented in WMO (2024). The ICOS network aims for the same goals, however covering a wider range (ICOS RI, 2020). All the data presented in this paper, including the data measured during the ICOS Mobile Laboratory and WCC-Empa audits, are reported on the same scale for each gas. The scale used for $CO_2$ is the WMO $CO_2$ X2019 (Hall et al., 2021) and for $CH_4$ the scale is WMO $CH_4$ X2004A (Dlugokencky et al., 2005).

### 2.2.1 ICOS

Pallas was labelled as an ICOS Class 1 atmosphere station (AS) in 2017. To maintain accuracy, ICOS instruments are automatically calibrated every 360 hours (15 days) and short-term target (ST) cylinder is automatically measured every 15 hours, and also immediately before and after calibration. A long-term target (LT) cylinder is measured directly after each calibration. The purpose of the short-term target is to ensure quality on daily basis, while the long-term target can ensure the continuity of the quality control as the cylinder should last over a decade (Yver-Kwok et al., 2021). As of April 2022, a set of four calibration standards is used (C1, C2, C3, C4) (three calibration standards before) in addition to the LT and ST cylinders. The use of these cylinders is in accordance with the ICOS atmosphere station specifications. The calibration and target gases used at the station were prepared by the ICOS Flask and Calibration Laboratory (CAL-FCL). For $N_2O$ and CO measurements done with the G5310 instrument, a working standard cylinder (ST WS) is used for short-term variability correction, as recommended by the ICOS ATC (ICOS RI, 2020). The ICOS sampling inlet is located about five meters from the measurement hut, on a mast 12 meters above the ground level (inlet 1, Fig. A1). The sampling inlet collects air samples at a flow of 2 lpm through 1300 Synflex

| Component | $CO_2$ | $CH_4$ |
|---|---|---|
| Compatibility goal | 0.1 ppm (NH) <br> 0.05 ppm (SH) | 2 ppb |
| Extended compatibility goal | 0.2 ppm | 5 ppb |
| Range in unpolluted troposphere | 380 - 450 ppm | 1750 - 2100 ppb |

**Table 1.** WMO Compatibility goals for $CO_2$ and $CH_4$ measurements. For $CO_2$, the goal is separated to Northern Hemisphere (NH) and Southern Hemisphere (SH).

1/4" tubing with a length of 17 m, which are subsequently partially dried using a Nafion dryer (Perma Pure MD-070-144S-2). The Nafion dryers were installed to the inlets of the ICOS G2401 and ICOS G5310 instruments in December 2020. Before that, the air was measured as wet. Without the dryer, the sample water content was, on average, 0.59 v% ($\pm$ 0.33 v%). With the dryer installed, the remaining water content was on average 0.06 v% ($\pm$ 0.01 v%). A Valco SD12MWE valve sequencer is used to switch the sample from ambient air to the calibration and target cylinders (Until April 2022 a solenoid valve sequencer was used). As the dryer is installed directly to the instrument inlet, the sample drawn from the cylinders is carried through the dryer as well. The setup of the ICOS instrumentation is illustrated in Fig. 3.

The water vapor present in the sample air dilutes the mole fractions of $CO_2$ and $CH_4$ as well as broadening the absorption peaks. In order to make the measured mole fractions of $CO_2$ and $CH_4$ comparable between stations with varying water content, the effect of water vapor in the sample must be removed. The resulting dry mole fraction is the comparable physical quantity to report. The dry mole fractions can be obtained by sufficiently drying the sample (dew point of at most -50 °C (WMO, 2013)), e.g. using cryogenic traps. Another way to account for the water vapor is to correct for the dilution and spectroscopic effects and determine the dry mole fractions computationally. All Picarro instruments are capable of correcting the water vapor effect of the sample and report the dry mole fractions. However, as the pressure broadening effect caused by the water vapor in the sample is different for each instrument, the ICOS strategy is to determine the correction coefficient for each instrument individually and apply the correction in the ICOS database (Hazan et al., 2016). For the ICOS instrument, the correction coefficients are determined by the ATC during the initial instrument test by first measuring a dry gas stream from a cylinder, and then humidifying the stream for 20 minutes a step with 0.25 v% steps from 0.5 v% to 2v % and an additional steps at 2.5v % and 3 v%. The coefficients for $CO_2$ and $CH_4$ are then determined with the following equation:

$$\frac{C_w}{C_d} = 1 + aH + bH^2, \tag{1}$$

where $C_w$ is the measured wet mole fraction, $C_d$ is the dry mole fraction (measured when H = 0), H is the measured water vapor concentration and $a$ and $b$ are the correction factors. These coefficients are evaluated during the ICOS audit, as well as approximately once per year by the station PI, and updated if deemed necessary by the ATC. The method for calculating the correction functions and correcting the measured mole fractions is presented in detail in Rella et al. (2013). Is it also possible to employ a combination of partly drying the sample, for example with a Nafion dryer, and correcting the data for the remaining water vapor. This approach is used at Pallas for the ICOS system since December 2020.

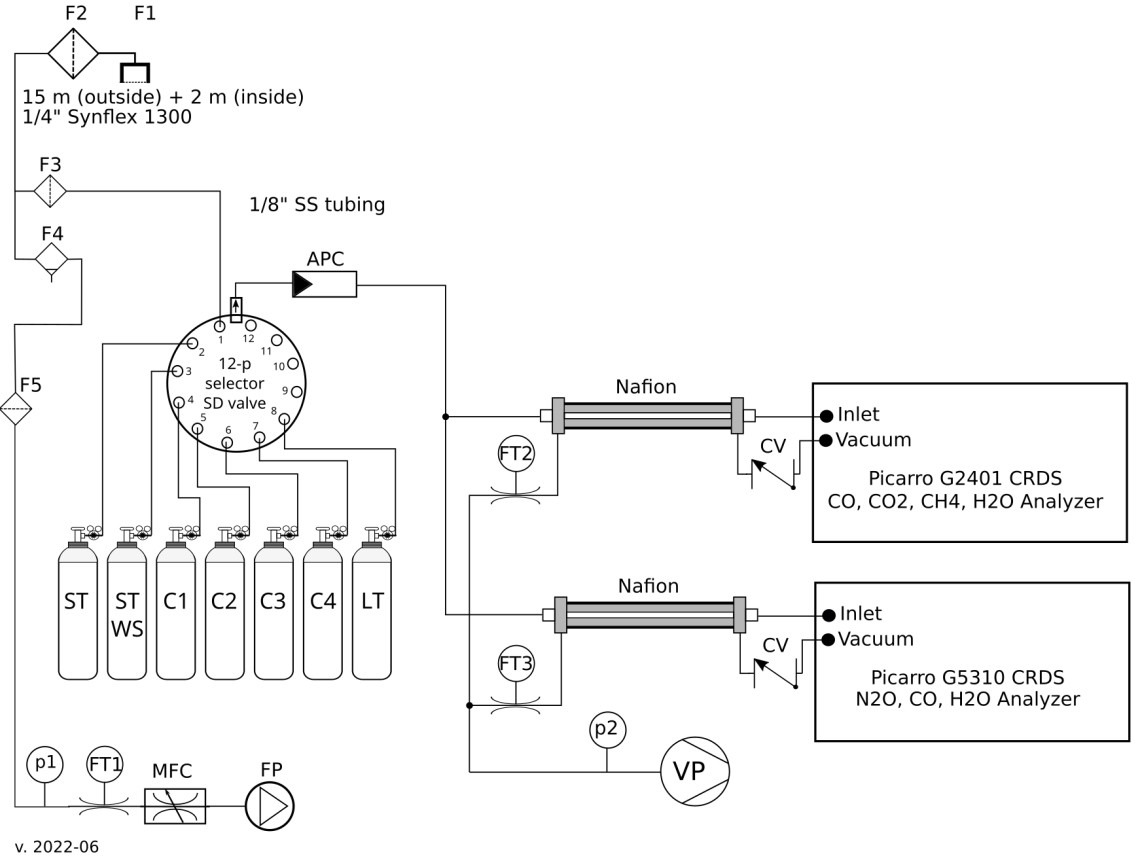

**Figure 3.** Schematic of the ICOS inlet system and manifold of the Pallas station. Abbreviations and specifications as follow: APC - Absolute pressure controller (MKS 640A61PS1M22M), CV - check valve (Swagelok SS-4C-1/3), F1 - inlet protector (vent filter Swagelok SS-MD-4), F2 - 2 µm pleated mesh filter (Swagelok SS-4FW-2), F3 - 0.5 µm sintered filter (Swagelok SS-2F-T7-05), F4 - drain separator for vacuum (SMC AMJ3000-N02B), F5 - filter (SMC ZFB100-06), FP - vacuum (flushing) pump (Edwards nXDS20i), FT1 - flow transmitter (SMC PFM710S-N01-C-MA), FT2, FT3 - flow transmitter (SMC PFMV505-1), MFC - mass flow controller (Bronkhorst F-201CV-5K0-BAD-22-V), p1, p2 - differential pressure transmitter (SMC ZSE30AF-N01-E-L), VP - vacuum pump (Vacuubrand MD 1), Nafion - Nafion dryer (Permapure MD-070-144-S-2), SD valve - Valco SD12MWE. Cylinders: ST - short-term target, ST WS - working standard, C1-C4 - calibration cylinders, LT - long-term standard. All cylinders are Luxfer L6X (AA6061 T6) 50l/WP 200, cylinder valves are ROTAREX D20030473 (brass) and pressure regulators are CALGAZ 1002 (nickel-plated brass).

To ensure the reliability of the acquired data, a sophisticated two-stage quality control process is implemented. Initially, an automatic quality control algorithm is employed by the ICOS ATC, followed by a manual flagging procedure conducted by the station's principal investigator (PI). The data processing chain implemented the ICOS ATC for $CO_2$ and $CH_4$ is presented in detail in Hazan et al. (2016). All the data measured by the ICOS-related instruments are submitted to the ICOS ATC servers and the processed data is available at the ICOS Carbon portal (Hatakka, 2024c, d).

### 2.2.2 GAW

The Pallas GAW Picarro is calibrated manually 4-5 times a year. To uphold the accuracy of the measurements between the calibrations, the GAW instrument automatically measures the short-term target cylinder automatically every 7 hours and 15 minutes and the long-term target cylinder every 25 hours 15 minutes. The calibration is done by measuring 9 standard cylinders. The GAW instrument measures humid air and a water vapor correction is applied to the data to calculate the dry mole fractions. The air inlet system of the GAW instrumentation is similar to the ICOS sampling system (Fig. 3), with the difference being in the calibration gases and the absence of the Nafion dryer at the instrument. For the GAW sampling system, the main manifold consists of 60 mm diameter stainless steel tubing which is continuously flushed with a nominal flow rate of 150 $m^3$ $h^{-1}$. The GAW instrument is connected to the main sampling manifold with a stainless steel tube. The sampling inlet is heated and located on the roof of the measurement building, approximately 7 m above ground level and 3 m above the roof (inlet 2, Fig. A1) and approximately 10 m from the ICOS inlet. All the standard cylinders used for the GAW instrument are filled by the FMI, and calibrated at the FMI laboratory against a set of four standard cylinders prepared at the NOAA Global Monitoring Laboratory (GML) before being sent to the station. The GML is the GAW Central Calibration Laboratory (GAW-CCL). These cylinders are regularly calibrated at the GML and the latest calibration for the FMI standards was in July 2018. The target cylinders used for quality assurance QA are presented in tables 3 and 5. For $CO_2$, cylinder D489486 is originally calibrated to older X2007 $CO_2$ scale, and for the purpose of this paper later converted to X2019 scale using the conversion equation determined by Hall et al. (2021):

$$X2019 = 1.00079 \times X2007 - 0.142 (\text{ppm}). \tag{2}$$

All other GAW cylinders are calibrated directly to X2019 scale.

Similarly to the ICOS instrument, the instrument specific water vapor correction factors are determined as well, but by the FMI. The approach used by the FMI is similar to that of ATC; a dry gas stream is humidified using a self-build instrument, ranging from 0 to 3.5 v% (Aaltonen et al., 2016). The coefficient are then calculated using Eq. 1. The processing of the GAW data is done by the FMI, and the data is submitted to the GAW database where it is available (Hatakka (2024a, b)). The GAW quality control process includes regular system and performance audits carried out by WCC-EMPA for $CO_2$ and $CH_2$ (Zellweger et al., 2016), and described here in Sect. 3.2.

 ## 2.3 Auxiliary measurements

In addition to GHG measurements, meteorological parameters are also measured atop Sammaltunturi. Measured parameters include ambient temperature, relative humidity, air pressure, wind speed and wind direction. Air temperature and relative humidity are measured at 7 m height from the ground using a Vaisala HMP155 sensor. The barometric pressure is measured at 2 m height using a Vaisala PTB220 sensor, and the wind speed and direction are measured at 9 m height with a Thies Ultrasonic 2D sensor.

## 3 Methods

The methods used for the time series analysis are presented in Sect. 3.1. The setup and procedure of the ICOS Mobile Laboratory audit is described in Sect.3.2.

### 3.1 Time series

The hourly time series measured with GAW setup was averaged to daily values. A curve was then fitted to the time series using a method developed by Thoning et al. (1989). The curve is fitted to the data in the form

$$f(t) = at^2 + bt + c + c_1 \sin(2\pi t + \vartheta_1) + c_2 \sin(4\pi t + \vartheta_2) + c_3 \sin(6\pi t + \vartheta_3) + c_4 \sin(8\pi t + \vartheta_4). \tag{3}$$

After fitting the function $f(t)$ to the data, the residuals of the fit are calculated. The residuals are then filtered with a low-pass filter to remove any remaining, unwanted oscillations. The filter equation is

$$H(f) = \exp(-\ln(2) \cdot (\frac{f}{f_c})^6), \tag{4}$$

where $f_c$ is the cutoff frequency (in days). Two different cutoff frequencies were used, $f_c = 667$ for long-term cutoff and $f_c = 80$ for short-term cutoff. The short-term cutoff is used for smoothing the curve, and the long-term cutoff is used for removing any remaining oscillations that might be present after the fitting. The trend curve, without seasonal oscillations is then calculated by subtracting the polynomial part of the Eq. 3 and adding the long-term filtered residuals to that curve. The smoothed curve is calculated by adding the short-term filtered residuals to the Eq. 3.

The yearly growth rate of the time series is calculated from the trend curve by taking the difference of the values of the last day (31.12) and the first day (01.01) of the given year. For example, the growth rate of 2020 would be calculated by taking the difference of the daily trend values of 31.12.2020 and 01.01.2020. This is approximately equal to the derivative of the trend curve, and gives information on how fast the concentrations are changing. Lastly, the seasonal variation is calculated from the smoothed curve by detrending it, i.e., subtracting the trend part from the smoothed line. The remaining curve shows the seasonal changes in the mole fractions.

The mole fraction time series presented in the results section are from the GAW instrumentation, as it is the longest time series available from Pallas. The $CO_2$ time series spans from July 1998 until the end of 2023 and the $CH_4$ time series spans from February 2004 until the end of 2023.

## 3.2 ICOS and GAW audit

As an ICOS station, the Pallas station was audited by the ICOS Mobile Laboratory during spring 2021. The ICOS Mobile Laboratory, operated by the FMI, is designed for visiting the ICOS atmosphere stations, ensuring the quality of their measurement through parallel ambient air comparison measurements as well as cross-comparison of the calibration standard cylinders. This type of additional quality control has been shown to be beneficial in maintaining and improving the quality of the greenhouse gas observations (Hammer et al., 2013; Zellweger et al., 2016).

The Mobile Laboratory is equipped with two Picarro models, G2401 and G5310, and in addition the Ecotech Spectronus FTIR instrument for $CO_2$, $CH_4$, $N_2O$ and CO. For calibration purposes, the Mobile Laboratory carries a set of 6 standard cylinders (hereafter travelling cylinders, TCs) filled by the ICOS FCL. Three of the cylinders are used for calibration purposes, two are used as the short-term and long-term targets cylinders for G2401 and G5310 and one is used as the target for the FTIR instrument. In order to account for a potential drift of the TCs, they are regularly compared to the same NOAA GML standards used to calibrate the GAW instrument standard cylinders between the audit campaigns. The Mobile Laboratory is also equipped with a freeze dryer (model ICOS) for drying the sample air, as well as a water-bench for evaluating the humidity correction coefficient of the stations' G2301/G2401 instruments.

The Mobile Laboratory audit consists of two visits to the station, one at the beginning and one at the end of the approximately six weeks long parallel measurement period. During this period, all the Mobile Laboratory instruments are sampling ambient air in parallel with the station instruments, through a dedicated spare sampling line. During this time the Mobile Laboratory instruments are calibrated automatically every 10 days, the short-term target cylinder is measured every 11 hours and the long-term target cylinder every 10 days. The station instrument is operating according to its normal operation schedule.

During the two visits, the station's calibration cylinders and the TCs are cross-compared to trace any possible issues related to the cylinders or the instrument. The Mobile Laboratory also performs a water vapor test on the station's instrument in order to assess the validity of the water vapor correction coefficients determined by ATC. For Pallas ICOS instrument the coefficient were deemed valid and no change was required. As all of the measurements and tests are performed during both visits, any possible drift in the cylinder concentrations, instrument performance or the water vapor correction coefficients can be tracked.

At the same time as the audit of the ICOS Mobile Laboratory, which focused on the ICOS system, an audit of the GAW system was carried out by WCC-Empa. The GAW measurements at Pallas have been audited three times in the last 2 decades: in 2007, in 2012 and most recently in 2021 (latest report: (WMO, 2022)). The WCC instrumentation consists of a single Picarro G2401 for $CO_2$, $CH_4$ and CO measurements. The zero reading of the WCC-Empa travelling instrument (TI-WCC) has been calibrated with $CO_2$ and $CH_4$ free air (or nitrogen 6.0) prior to field use by adjusting the offsets in the user calibration file of the instrument. During the field use of the TI-WCC, only one working standard (WS) is used to calibrate the instrument for $CO_2$ and $CH_4$. In a first step, a Loess function is fitted to the WS (measured every 1445 min) to correct for drift. The resulting drift correction is then applied to all TI-WCC data in a second step. The drift corrected WS is then used to apply a calibration factor to the data using the assigned value of the WS based on calibration against the CCL standards before and after field use. Two target standards are measured to verify the drift correction. The sample air was dried using a Nafion dryer (Permapure,

Model MD-070-48S-4), and the WCC-Empa Picarro was calibrated every 1445 minutes. The WCC-Empa instrument sampled air from the same inlet as the ICOS and the ICOS Mobile Laboratory instruments inlets using a 1/4" Synflex-1300 line flushed by an external pump at 3 l/min. The joint audit period ran from 05.03.2021 until 19.04.2021. During the GAW-audit, the calibration cylinders of are cross-calibrated as well. The GAW instrument was calibrated at the beginning and at the end of the joint audit period.

The instruments and their sampling locations during the audit are illustrated in Fig. A1. In the figure, ML refers to the ICOS Mobile Laboratory instrument. Outside audit campaign, the sampling of the ICOS and GAW instruments is done as during the audit.

### 3.3 Data comparison

The data of the ICOS and GAW instruments were compared at both hourly and daily resolutions, starting in September 2017, when the ICOS instrument was installed at Pallas. In order to focus on the regional signal, the data was filtered based on the wind speed and the standard deviation of the hourly measurements based on wind statistic, as defined by Aalto et al. (2015). Due to the differing wind speeds between summer and winter, the criteria is defined separately for the seasons. The lower limit for the wind speed is 3 m/s during summertime (June-August) and 4 m/s during wintertime, and the standard deviation less than 0.5 ppm ($CO_2$) or 3 ppb ($CH_4$). Based on this criterion, approximately 31 % of the $CO_2$ and 23 % of $CH_4$ hourly data were discarded.

In addition, to remove any hourly means with possible biased sampling (i.e. if a calibration sequence starts in the middle of the hour, causing the hourly mean to represent only part of the hour), only hours with 60 minutes of measurements from both instruments were considered. When comparing the data on hourly and daily resolution, it could be expected that the mean difference remains the same on both resolutions. However, as we filter the hourly data and later aggregate this filtered time series to daily values, days with different amount of hourly data points are represented differently in the final daily timeseries, leading to small differences in the comparison of daily and hourly values (i.e., a day with 24 hourly data points would be weighted twice as much as a day with 12 hourly data points in hourly means, while after aggregating to daily means both days would be weighted equally).

To quantify the effect of the different systems on the fitted trend lines and growth rates, a curve according to Eq. 3 was fitted to the time series from both systems for the time period of concurrent measurements. The mean difference between the trend lines is calculated as well as the confidence intervals. From the trend lines, the annual growth rate was calculated for both systems and their differences are reported.

Often, the convention is to calculate the daily averages from the afternoon hours in order to maximize the boundary layer mixing (Resovsky et al., 2021). We also calculated the daily means using this method and compared the differences with the data filtered by wind speed and hourly standard deviation. The exact hours chosen may vary from station to station, here the hours 12:00 - 17:00 (EET) are used.

## 3.4 Quality assurance

For QA of the different measurement instruments, we analyzed their respective calibrated target cylinder measurements. Using the calibrated values allows for a comprehensive evaluation of the instrument as well as the calibration cylinders and calibration method, which also varies by instrument. We use two different measures for the measurement stability, similar to those used by Yver-Kwok et al. (2021). As a measure of long-term repeatability (LTR), we calculate the deviation of the target cylinder measurement from the assigned value of the cylinder, and calculate the standard deviation of these values. This approach is used to account for different target cylinders used over time. For situations where only one cylinder is used, this value is equal to simply taking the standard deviation of the target cylinder measurements. For short-term repeatability (STR), we use the standard deviation of the individual cylinder measurement sequences. In addition, we calculate the mean bias of the measured values to the assigned target mole fractions for each instrument. For stabilization, only the last minutes of the injection are used for the analysis. For ICOS and GAW instruments, each measure is calculated for the whole time period of concurrent measurements as well as for the audit period. For the ICOS instrument, the LT is used for the long-term comparison QA and the ST for the audit period. For GAW instrument, one target cylinder is used for QA and a short-term working standard is used for drift correction. For the audits, one target cylinder is analyzed for each travelling instrument. In order to to get a stable measurement, only the last minutes of each cylinder measurement is used for calculating the means. The different cylinders are presented in tables 3 ($CO_2$) and 5 ($CH_4$). The assigned values, measurement times as well as the number of minute data points used for averaging are presented, as well as STR, LTR and mean biases.

## 4 Results and discussion

The results of the time series analysis are presented in this section. For each measured component a time series is presented along with a fitted curve to the data and a long-term time series without seasonal oscillations. In addition to the mole fractions, a growth rate for each component is calculated according to method described Sect. 3. Average diurnal and seasonal cycles are also presented.

### 4.1 $CO_2$

In this section the results for long-term mole fraction measurements, comparisons of ICOS and GAW instruments and results of the audit for $CO_2$ are presented.

#### 4.1.1 Long-term measurements of $CO_2$

The observed daily $CO_2$ mole fractions from the GAW measurements at PAL (Hatakka, 2024b) as well as the Northern Hemisphere mean marine boundary layer (MBL) from NOAA (Lan et al., 2024a) are presented in Fig. 4 (a) along with the annual growth rates for GAW observations and MBL data (b).

Consistent with the global trend, the $CO_2$ levels at Pallas have risen steadily at a rate of approximately 2 ppm/year from 373 ppm (1999 mean) to 423 ppm (2023 mean).

Likely due to the location, the background mole fractions measured at Pallas are, in general, higher than the average in the Northern Hemisphere MBL. The mean difference in the daily average is 1.9 ppm (95% CI: [-8.0,11.8] ppm. This difference is significantly higher during the cold season (approximately September-April) with a mean difference of 4.1 ppm (95% CI:
[-3.1, 12.8] ppm than during the warm season (approximately May - August), when the mean difference is -2.7 ppm on average (95% CI: [-9.9,3.0] ppm).

The average growth rate of about 2 ppm/year at Pallas is comparable to the globally observed changes in $CO_2$ (Fig. 4). Measurements at Pallas show, however, larger deviations in the $CO_2$ growth rate than the Northern Hemisphere averages. This can partly be explained by noting that we compare measurements from one location to an averaged product, which
naturally leads to higher variation. A negative growth rate is observed at Pallas in 2001; this is caused by elevated $CO_2$ mole fractions during late 2000, and lower values during late 2001. The exact reason is difficult to quantify based on atmospheric measurements alone, however the fall 2000 was warm with little precipitation, which could influence the $CO_2$ emissions.

Measured $CO_2$ mole fractions at Pallas station are representative for a large area due to its remote location, and no significant anthropogenic sources are present near the station. $CO_2$ sinks at Pallas are mostly vegetation, and the effect can be seen in the
diurnal cycle (Fig. 5 (a)). The seasonal cycle is well defined (Fig. 5, (b)) the yearly maximum is, on average, on day 37 (beginning of February) and the minimum on day 220 (beginning of August) (calculated as the annual minima and maxima of the smoothed curve). During the vegetation period (approximately May-September) a diurnal cycle is also visible (5, (a)) with a mean amplitude of 4.2 ppm, indicating the influence of local vegetation. During the winter months no diurnal variation is visible.

### 4.1.2   Comparison of GAW and ICOS $CO_2$ measurements

The hourly and daily biases between the ICOS (Hatakka, 2024d) and GAW measurement systems are presented in Fig. 6. The mean $CO_2$ mole fraction measured was 416.90 ppm for both instruments, with the standard deviations of 3.80 ppm.

The differences of the instruments fit well within the WMO/GAW compatibility goals: for daily measurements, 94.7 % of the days are within the assigned limits, and for hourly measurements, 93.3%. The mean difference is <0.01 ppm (95% CI:
[-0.07, 0.10] ppm for the daily means and <0.01 ppm (95% CI: [-0.10, 0.13] ppm) for the hourly means. Before a Nafion dryer was installed to the ICOS instrument inlet at the end of 2020, there is a seasonal variation in the differences between the instrument (Fig A3, (a)).

With the addition of the Nafion dryer to the ICOS instrument, the seasonal variation is reduced and the differences between the two instruments were slightly changed: before the Nafion was added, the average hourly difference was 0.01 ppm (95%
CI: [-0.10, 0.13] ppm) and the average daily difference was 0.02 ppm (95% CI: [-0.07, 0.11] ppm). After the addition of the Nafion dryer, the hourly differences was -0.02 ppm (95% CI: [-0.10, 0.12] ppm) and the daily difference was -0.02 ppm (95% CI: [-0.07, 0.10] ppm). Thus, the Nafion dryer appears to slightly increase the absolute difference between the measurements, but at the same time it reduces the spread of the differences. There could be a remaining seasonal variation after the ICOS

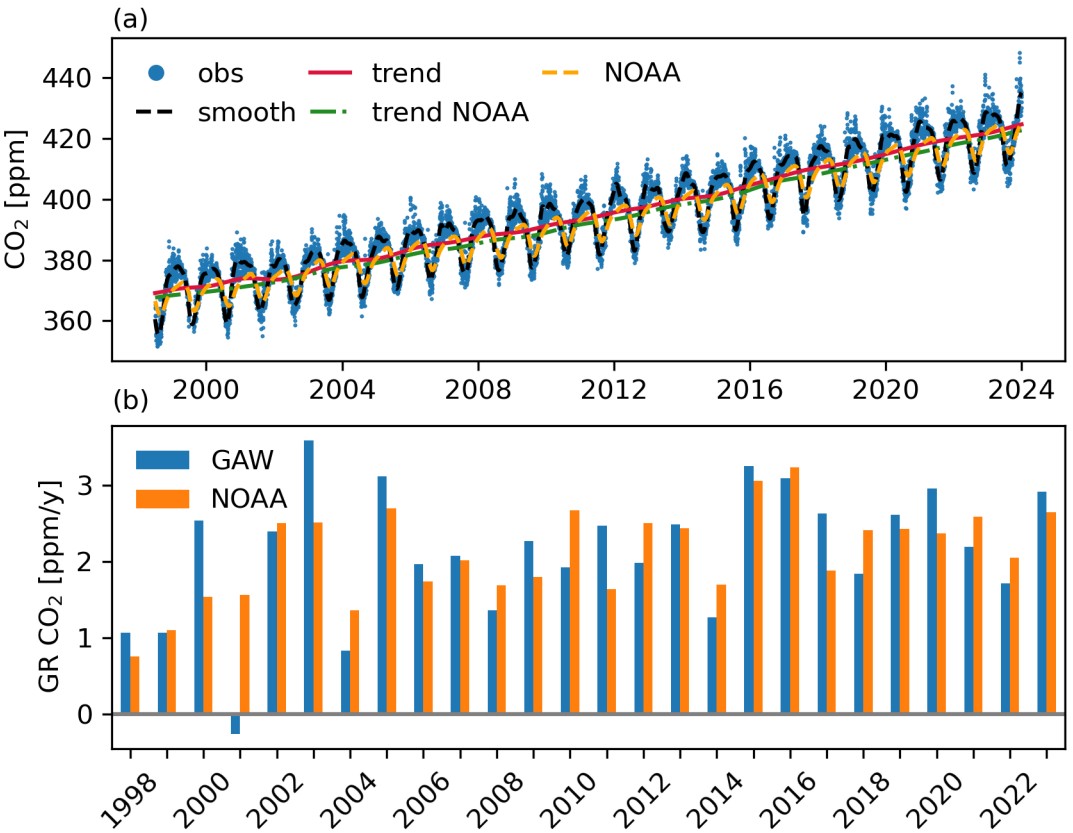

**Figure 4.** (a) Time series of $CO_2$. The blue dots indicate the daily observed values from the GAW instrument, the red line the trend value derived from the GAW observations, and the black dashed line the smoothed line from the GAW observations. The yellow dashed line indicates the NOAA mean MBL data, and the green dotted line the trend derived from the NOAA mean MBL. (b) Yearly growth rates of $CO_2$ for GAW measurements (blue) and the NOAA mean MBL data (orange).

sample is dried, however the stronger variation during summer in $CO_2$ masks this effect. Before the the addition of the Nafion dryer, 91.9% of the hourly data and 94.1% of the daily data fit within the WMO/GAW limits. After the addition of the Nafion 94.8% of the hourly data and 95.3% of the daily data fit within the limits. Overall, the drying the sample air with a Nafion dryer seems to be beneficial for the $CO_2$ measurements. There is a negative correlation between the $\Delta CO_2$ and the water vapor concentration of about -0.05 ppm/v% (intercept 0.05 v%, p < 0.05, $R^2$: 0.12) when both of the instruments measure wet air (Fig. A3, (a)). The correlation changes to slightly positive when the ICOS sample air is dried (slope 0.04 ppm/v%, intercept = -0.04, p < 0.05, $R^2$ = 0.08). Additionally, only a very weak correlation with mole fraction was observed on the differences (Fig A3, (b)). The agreement of the trend lines fitted for both time series was excellent, with the mean difference being 0.02 ppm. Calculating the yearly growth rates from both trend lines also agree well with mean difference of 0.01 ppm/year.

The target cylinder $CO_2$ measurements of both instruments are presented in Fig 7. The values are give in bias to the assigned cylinder values. For ICOS instrument both ST and LT tanks are shown, and one target tank for GAW. Especially the first ICOS ST cylinder (D348368) shows a strong drift while in use at the station. The cylinder was changed in the beginning of 2019 and send to CAL for recalibration, and taken again in use in 2020. Similarly, the first LT cylinder (D348367) shows a slight drift over time. The GAW cylinders seem to be more stable, showing less drift.

For each cylinder, the STR, LTR and assigned values as well as the mean bias to the assigned value for $CO_2$ are given in table 3. The STR for $CO_2$ measurements is very consistent across the different instruments and cylinders, but the LTR shows some deviation. Especially the ST cylinder of the ICOS instrument D348368 show a higher LTR than other cylinders. For the ICOS instrument, the biases between the different ST cylinders is consistent, as well as between the two LT cylinders, however the biases of the LT cylinders are clearly deviating from the ST cylinder biases. For GAW the biases are more consistent between the two last cylinders, however the first cylinder (D489486) shows a higher bias. The biases could also be caused by calibration offsets as the cylinder mole fraction values vary. This can be confirmed with the cross-calibrations with TCs during the audit, presented in tables A1 for GAW and A2 for ICOS. For the GAW-instrument, a slight negative slope exists when comparing the bias of the TC measurements to the mole fractions, and for ICOS instrument a slight positive slope exists.

When the data are filtered to include afternoon hours only, the differences between the two systems is 0.01 ppm (95% CI: [-0.10, 0.09] ppm), which is almost identical to the filtering based on wind speed and hourly standard deviation.

### 4.1.3  Audit results for $CO_2$

Results of the combined ICOS and GAW audit are presented in Fig. 8. In the analysis the ICOS Mobile Laboratory data is measured with G2401 instrument. During the audit period, both the ICOS Mobile Laboratory instrument and the WCC-Empa travelling instrument were sampling from a dedicated sampling line and inlet, located next to the inlet of the local ICOS system. As the ICOS inlet is in a slightly different location than the GAW inlet, the measured time series can be at times inconsistent in case of local emissions episodes. In order to provide comprehensive overview of the differences in the measured time series, no distinction has been made between regional and local signals, in contrast to the long-term comparison between the ICOS and GAW systems.

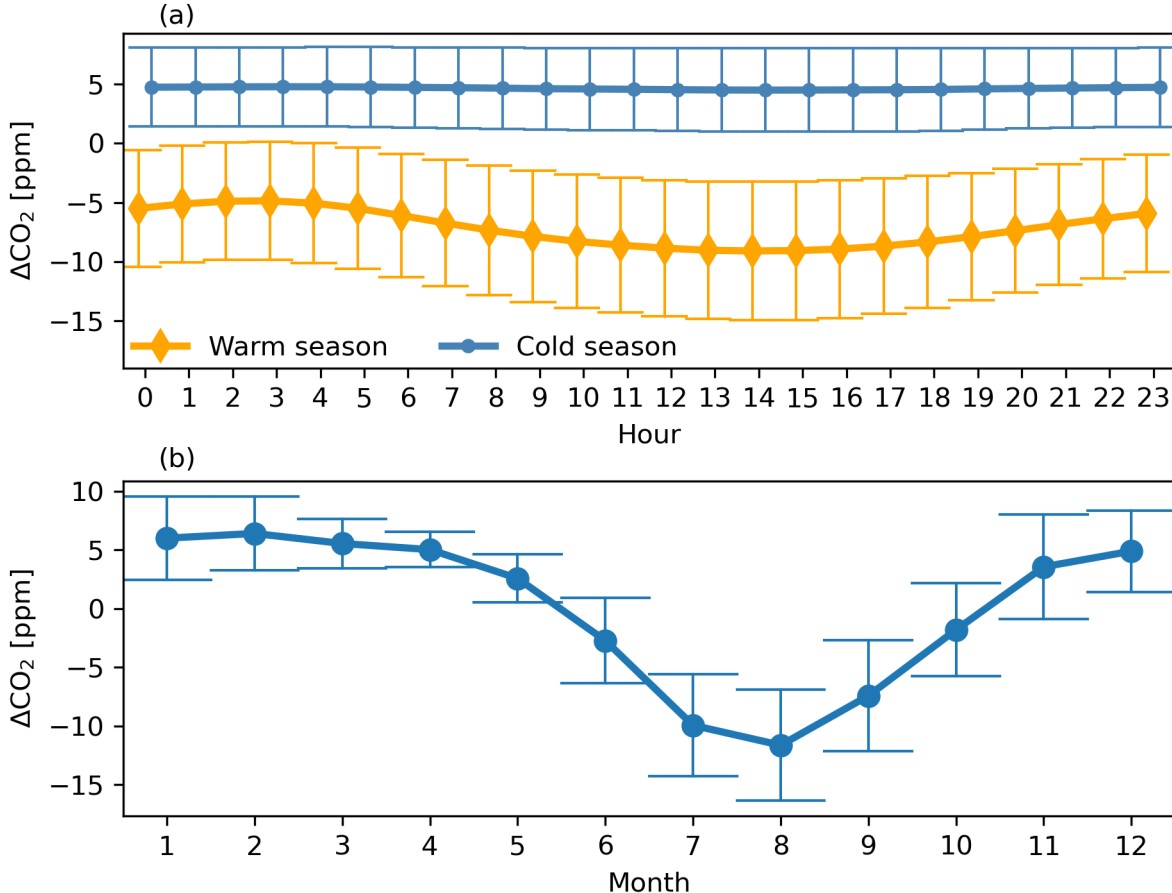

**Figure 5.** (a) Average diurnal cycle of during the warm and cold seasons (deviation from the trend line) for $CO_2$ and (b) the average seasonal variation of $CO_2$ (deviation from the trend line) (b). Data points are hourly (a) and monthly (b) means with associated standard deviations.

The summarized results of each $CO_2$ comparison is presented in Table 2. The differences between the instrumentation are consistent in time, and are mostly within the WMO/GAW compatibility goals. The best agreement is found between the ICOS Mobile Laboratory and the ICOS system, but also all the other comparisons fit within the compatibility goals. The largest spread in the confidence intervals are found between the GAW and Mobile Laboratory and WCC-Empa and the ICOS Mobile Laboratory measurements. The former is probably due to the different inlet locations. For the latter, it is difficult to find a definitive reason. While the flow rates of the different instruments are slightly different, the effect of the flow rate is likely small as the lines at the station are rather relatively short and we are comparing hourly aggregated data. In general, the spread of the confidence intervals between the different comparisons is consistent, and with the exception of the WCC comparison with the ICOS Mobile Laboratory, all 95% CIs fall within the compatibility goals. When the data are filtered for wind speed and hourly standard deviation, as done for the analysis for trend and long-term differences, the spread between GAW and ICOS Mobile

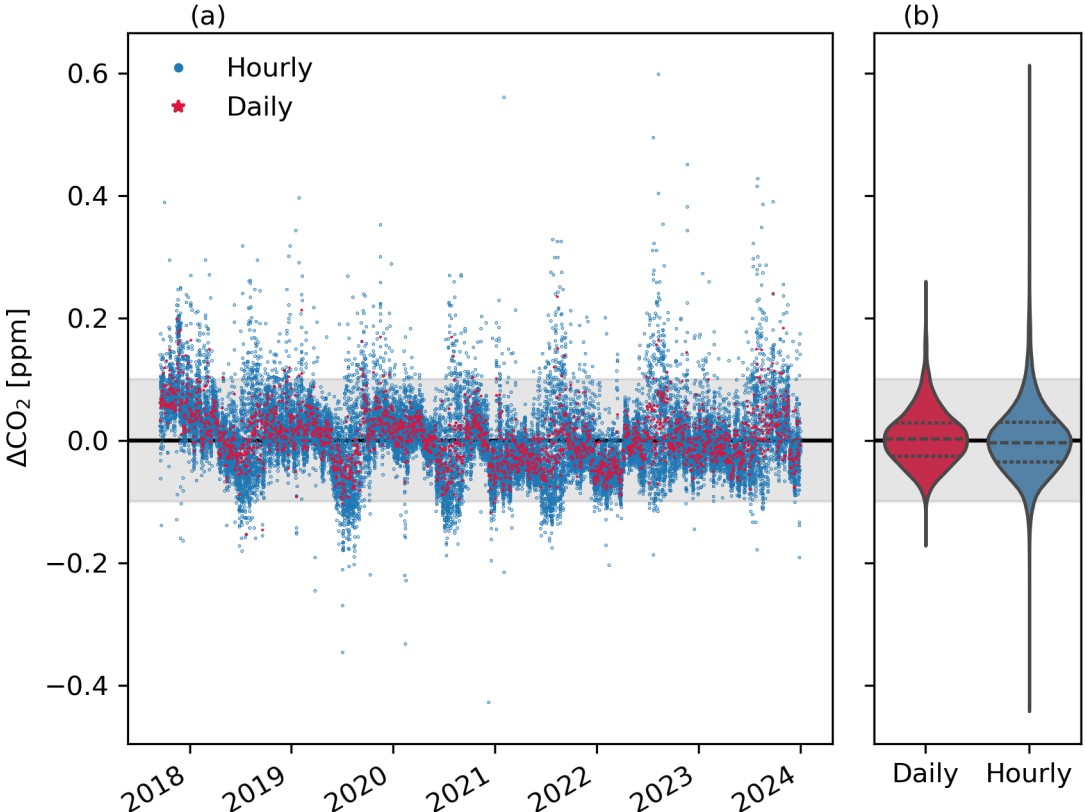

**Figure 6.** Panel (a): Time series of the differences between the ICOS and GAW instruments for $CO_2$ (GAW-ICOS). The blue dots indicate the hourly mean values and red dots the daily mean values. Panel (b): distribution of the data points. The gray shaded areas indicate the WMO/GAW compatilibity goals.

Laboratory decreases to 0.14 ppm, between GAW and WCC to 0.09 ppm and between ICOS and GAW to 0.12 ppm, indicating that the different inlet location affects the comparison when the air is not well mixed. The difference between WCC and ICOS Mobile Laboratory decreases as well to 0.15 ppm, but a decrease is observed for comparison of ICOS and WCC. Each of the comparison pairs and the mole fraction dependency of the their respective differences are presented in Fig A4. Comparisons against the WCC instrument (a, d, e) show a weak dependency on mole fractions, while the other comparisons do not. To account for possible differences in the calibration standards, the ICOS Mobile Laboratory cylinders were measured with the ICOS instrument, and the WCC TCs were measured with the GAW instrument. The results of the measurements are presented in table A1 and table A2. The calibration cylinder measurements show that for $CO_2$ the ICOS instruments measurements differ on average from -0.09 ppm to 0.01 ppm to the assigned cylinder values, and the GAW instrument measurements differ from

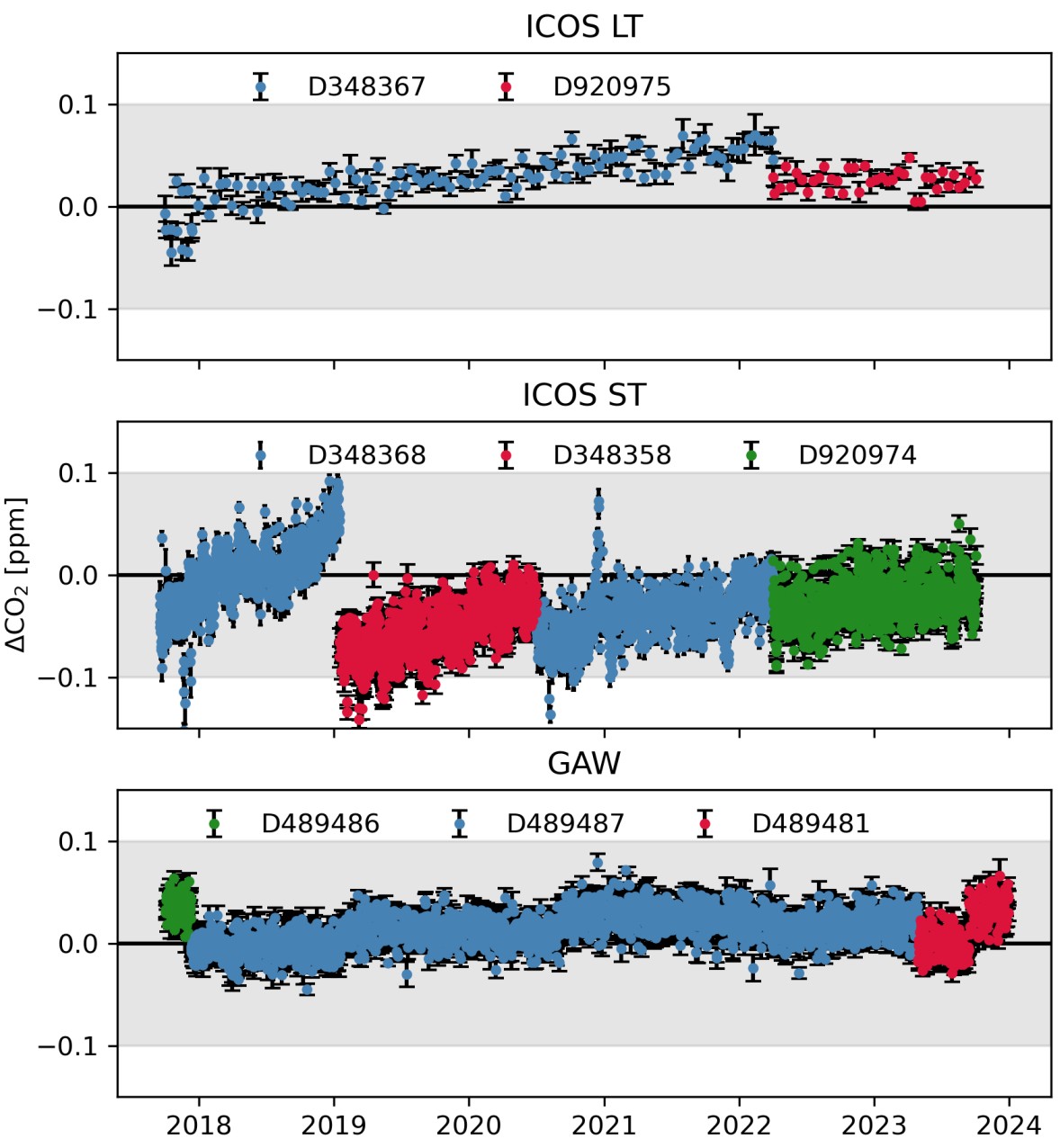

**Figure 7.** $CO_2$ target measurements of ICOS LT, ICOS ST and GAW target cylinders over the whole comparison period. Different cylinders used are marked with distinct colors. The data are given as means of each sequence with the associated standard deviation.

-0.05 ppm to 0.03 ppm. The QA of the $CO_2$ measurements using the target cylinders for ICOS and GAW instruments are presented in table 3.

The $CO_2$ measurements of the target cylinders during the audit are presented in Fig. 9. For ICOS instrument, both ST (D348368) and LT (D348367) are shown. No signficant drift is evident from the time series for any of the instruments. The
410 STR and LTR, as well as the cylinder biases are presented in table 3. During the audit, the STR is compatible between all the instruments, and the LTR of GAW, WCC and Mobile Laboratory instrument are comparable. The ICOS ST and LT cylinders show slightly higher LTR then the rest.

A study by Hammer et al. (2013) compared a fourier transform infrared (FTIR) based travelling instrument (TI) to a GC reference instrument at Heidelberg (HEI) as well as to local CRDS instruments at two field stations, Cabauw (CBW) and
415 Houdelaincourt (OPE) on 3-minute aggregated data. Their results show median differences between the traveling instrument and the local instrument being -0.02 [-1.13, 1.49] ppm at HEI, 0.21 [0.08, 0.40] ppm at CBW and 0.13 [-0.28, 1.15] ppm at OPE (brackets referring to 5% - 95% quantiles). Similarly, a study by Vardag et al. (2014) compared similar FTIR instrument at HEI and at a field station in Mace Head (MHD); they found median differences ($\pm$ interquantile range) of $0.04 \pm 0.22$ ppm and $0.03 \pm 0.31$ ppm at HEI (before and after the MHD campaign) and $0.14 \pm 0.04$ ppm at MHD. Our results thus show a rather
good agreement, comparable even with the results at Heidelberg where the TI was compared against a reference instrument. However, our results compare hourly means while Hammer et al. (2013) and Vardag et al. (2014) compare 3-minute means.

In Zellweger et al. (2016) an overview of results of GAW audits performed in Danum Valley (DMV), Cape Verde (CPVO), MHD, as well as an earlier audit at Pallas, are presented. The audits were performed in 2012 (PAL & CBO) and 2013 (DMV & MHD). The comparison for $CO_2$ was made in PAL, DMV and CVO. The measurement methods were non-dispersive infrared
(NDIR) in PAL and DMV, and off-axis integrated cavity output spectroscopy (OA-ICOS) in CVO, compared against the traveling CRDS instrument. Results of the audits show a median deviation (1h aggregation, $\pm$ standard deviation) of $0.08 \pm$ 0.03 ppm at PAL, $0.03 \pm 0.21$ ppm at DMV, 0.06 ppm $\pm 0.06$ at CVO. For Pallas, a comparison of the local CRDS against the traveling CRDS is presented in (Rella et al., 2013). For $CO_2$, the mean deviation was $-0.025 \pm 0.034$ ppm.

## 4.2 $CH_4$

In this section the results for long-term mole fraction measurements, comparisons of ICOS and GAW instruments and results of the audit for $CH_4$ are presented.

### 4.2.1 Long-term $CH_4$ measurements

The measured daily $CH_4$ mole fractions (Hatakka, 2024a) as well as the mean marine boundary layer means in the Northern Hemisphere (NOAA $CH_4$ MBL, Lan et al. (2024b)) are presented in Fig. 10.
As for $CO_2$, the $CH_4$ mole fractions measured at Pallas are higher than the average for the Northern Hemisphere due to the station's location at high latitude. At Pallas the mole fractions have increased from 1865 ppb in 2004 to 2023 ppb in 2023, while in the Northern Hemisphere on average the mole fractions have increased from 1819 ppb in 2004 to 1969 ppb in 2023. On average the mole fractions at Pallas are 54 ppb higher than on the average in the Northern Hemisphere. Unlike for $CO_2$, there

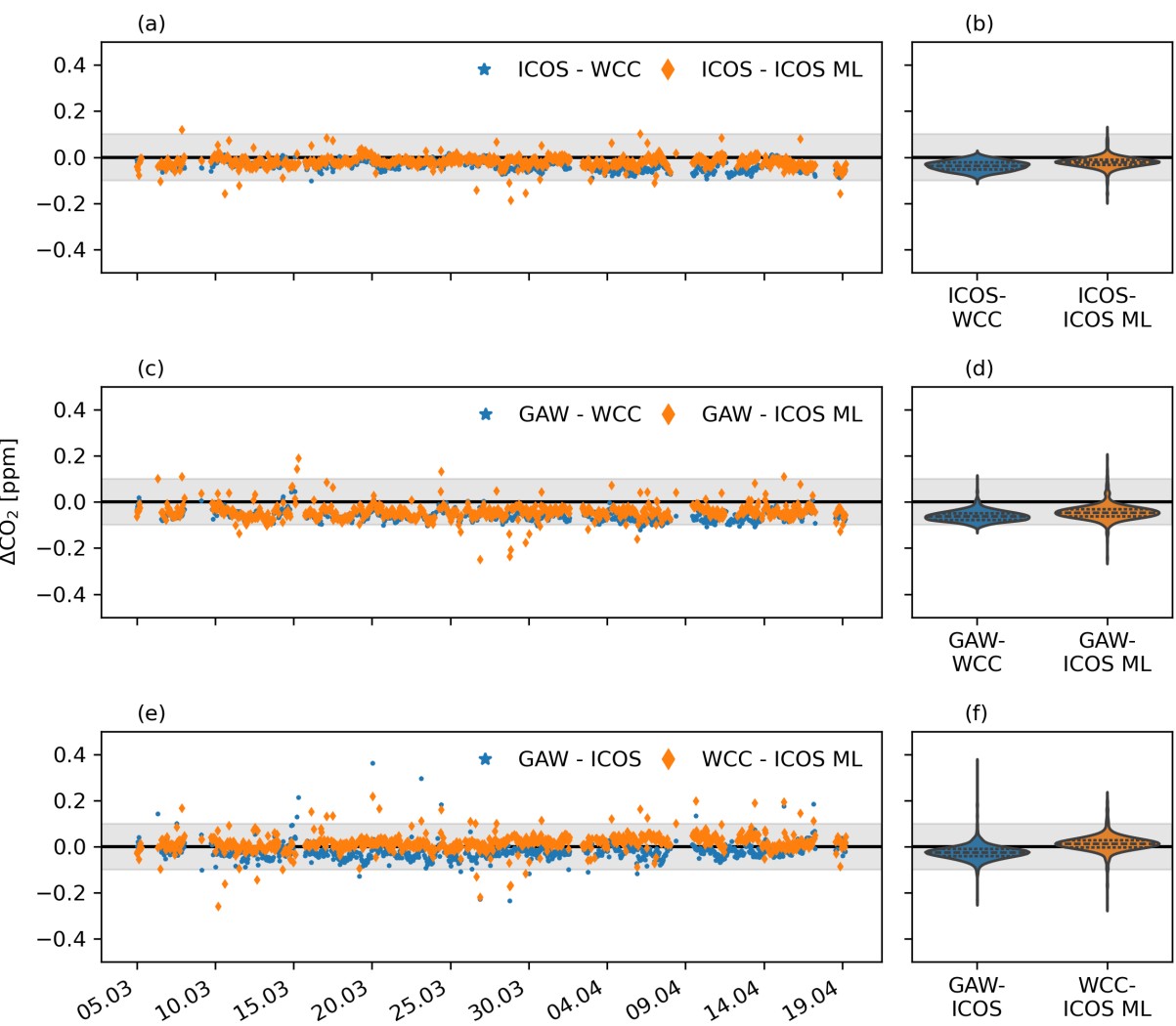

**Figure 8.** Time series of the differences for different comparison pairs (a, c and e), and their associated distributions (b, d and f) for $CO_2$. In the legend ML refers to Mobile Laboratory.

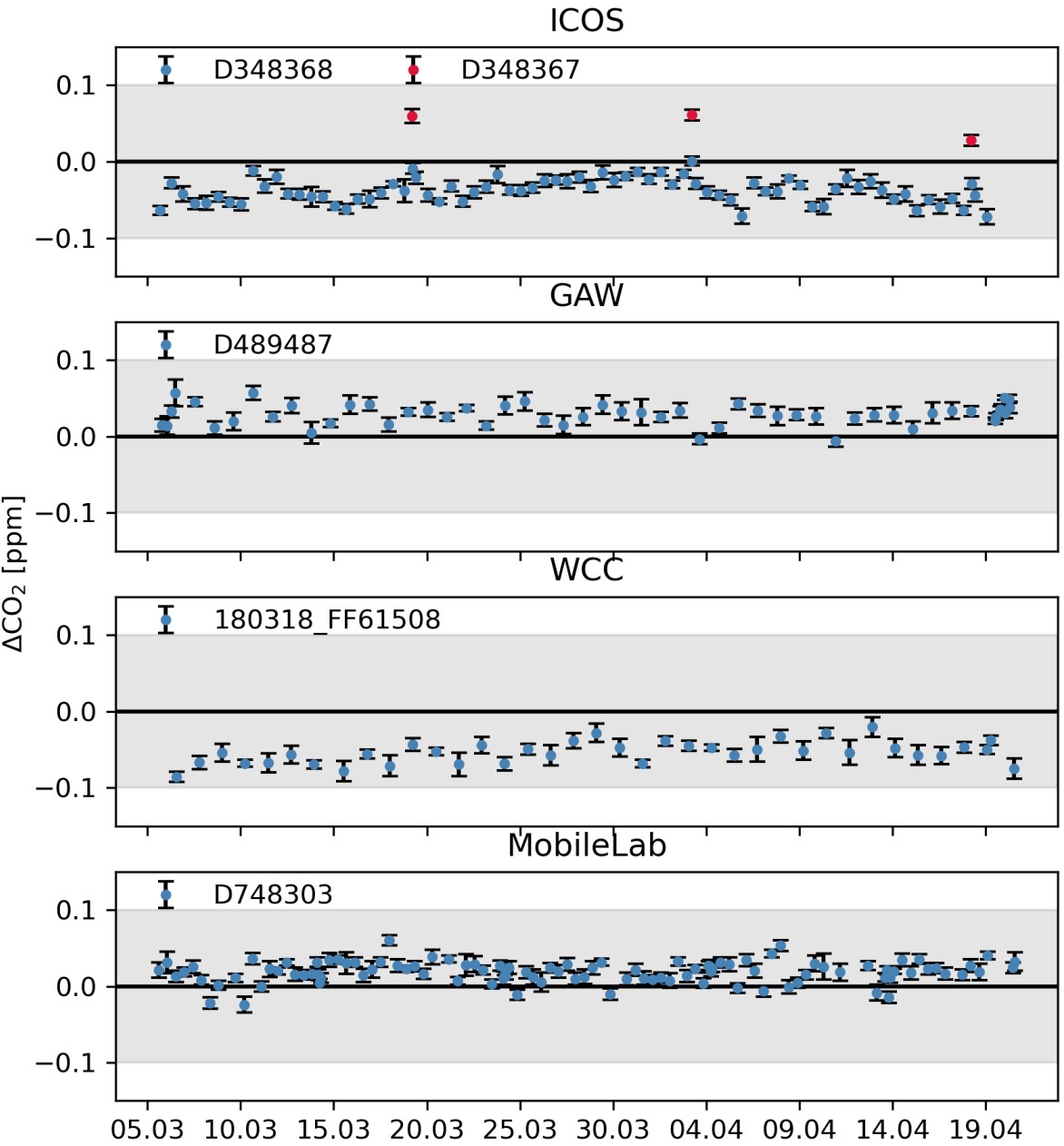

**Figure 9.** CO$_2$ target measurements for ICOS, GAW, WCC and Mobile Laboratory instruments during the audit period. For ICOS both ST and LT are presented. The data are given as means of each sequence with the associated standard deviation.

| | Mean (ppm) | 95% CI (ppm) | CI range (ppm) | Slope | Intercept |
|---|---|---|---|---|---|
| **Full period** | | | | | |
| GAW - ICOS | <0.01 | -0.10 to 0.13 | 0.23 | 0.999 | 0.32 |
| **Audit period** | | | | | |
| ICOS-Mobile Lab | -0.02 | -0.07 to 0.02 | 0.09 | 1.000 | 0.19 |
| GAW-Mobile Lab | -0.04 | -0.10 to 0.07 | 0.17 | 1.002 | -0.69 |
| WCC-Mobile Lab | 0.02 | -0.06 to 0.11 | 0.17 | 0.998 | 0.97 |
| ICOS-WCC | -0.04 | -0.08 to 0.00 | 0.08 | 1.001 | - 0.67 |
| GAW-WCC | -0.06 | -0.10 to 0.01 | 0.11 | 1.002 | -0.84 |
| GAW-ICOS | -0.02 | -0.08 to 0.10 | 0.18 | 0.998 | 0.97 |

**Table 2.** Results of the cross-comparisons of different instruments on hourly mean data over the full period and during the audit period for $CO_2$. For the full period, the data is filtered for wind speed, however during the audit period no wind speed filtering is applied. Slope and intercept are fitted to the mole fraction dependency of the difference.

| | Cylinder | Purpose | LTR (ppm) | STR (ppm) | Bias (ppm) | Conc (ppm) | Nb | Measure time (min) |
|---|---|---|---|---|---|---|---|---|
| **Full period** | | | | | | | | |
| ICOS | D348367 | LT | 0.02 | 0.01 | 0.03 | 450.80 | 10 | 20 |
| | D920975 | LT | 0.01 | 0.01 | 0.03 | 461.97 | 10 | 20 |
| | D348358 | ST | 0.02 | 0.01 | -0.05 | 409.76 | 10 | 20 |
| | D348368 | ST | 0.03 | 0.01 | -0.03 | 399.71 | 10 | 20 |
| | D920974 | ST | 0.02 | 0.01 | -0.03 | 415.02 | 10 | 20 |
| GAW | D489481 | | 0.02 | 0.01 | 0.01 | 418.30 | 9 | 18 |
| | D489486 | | 0.01 | 0.01 | 0.04 | 396.61 | 9 | 18 |
| | D489487 | | 0.02 | 0.01 | 0.01 | 411.07 | 9 | 18 |
| **Audit period** | | | | | | | | |
| ICOS | D348367 | LT | 0.02 | 0.01 | 0.05 | 450.80 | 10 | 20 |
| ICOS | D348368 | ST | 0.02 | 0.01 | -0.04 | 414.64 | 10 | 20 |
| GAW | D489487 | | 0.01 | 0.01 | 0.03 | 411.07 | 9 | 18 |
| WCC | 180318_FF61508 | | 0.01 | 0.01 | -0.05 | 417.57 | 4 | 9 |
| MobileLab | D748303 | | 0.01 | 0.01 | 0.02 | 411.94 | 8 | 20 |

**Table 3.** Results of the target measurements of $CO_2$ for each instrument for the full period of comparisons between ICOS and GAW instruments and for the audit period. Nb refers to the number of data points used for averaging and measure time is the total time each cylinder is measured during one injection.

is no significant variation in the differences between the Pallas observations and the Northern Hemisphere averages between the cold and the warm season, indicating little influence of the local vegetation to the $CH_4$ mole fractions.

Definite diurnal and seasonal cycles during the warm period are also visible in $CH_4$ timeseries (Fig 11, a and b, respectively). The amplitude of the diurnal cycle during the warm period is approximately 6.5 ppb and the amplitude of the seasonal variation 35.7 ppb. The seasonal cycle has the highest values usually in January, and a second peak in September. Lowest values for the seasonal cycle are usually in June.

Major $CH_4$ sources in the Arctic that influence the mole fractions at Pallas are anthropogenic sources and wetlands followed by freshwater systems. Of these Arctic sources, during wintertime anthropogenic emissions contribute up to 56% and during summertime the wetland emissions contribute up to 70% and freshwater systems up to 26% (Thonat et al., 2017). However, major contribution is still from emissions originating outside of the Arctic area.

The $CH_4$ concentrations have been increasing rapidly after 2019, and the growth rate peaked in 2020. The increase in the methane in 2007-2017 can likely be attributed to the increased emission from wetlands (Nisbet et al., 2016, 2019). The more recent increase in 2020 could be attributed to the increased wetland or anthropogenic emissions locally (Yuan et al. (2024); Tenkanen et al. (2024); Ward et al. (2024)) as well as decrease in atmospheric sinks (Peng et al., 2022; Stevenson et al., 2022; Qu et al., 2022; Feng et al., 2023). However, the increase measured at Pallas is significantly higher than on the average in the Northern Hemisphere, indicating a strong increase of local and regional emissions. Based on results from inversion model results, these emissions are most likely caused by increase in wetlands or anthropogenic sources (Tenkanen et al., 2024). As for $CO_2$, the growth rate of $CH_4$ varies more for Pallas than for the northern hemisphere average. This can be expected, as local variations in sources and sinks affect the mole fractions at the site more than the hemisphere averages. Especially in 2006 and 2010 the growth rates observed at Pallas were negative. A study by Tsuruta et al. (2019) found that during those years the $CH_4$ emissions in Finland were lower than usually, which is a likely explanation for the lower mole fractions observed at Pallas during those years. The summer 2006 and autumn 2010 were both dry with less precipitation than normally, likely influencing the emissions.

### 4.2.2 Comparison of GAW and ICOS $CH_4$ measurements

The hourly and daily biases between the ICOS (Hatakka, 2024c) and GAW instruments for $CH_4$ are presented in Fig. 12.

As for $CO_2$, the measurements of the ICOS and GAW instruments for $CH_4$ agree very well. The mean difference for the hourly measurements is 0.47 ppb (95% CI: [-0.40, 1.53] ppb) and for daily measurements the mean is also 0.47 ppb (95% CI: [-0.36, 1.39] ppb).

The differences show a seasonal variation with higher differences in the winter compared to summer until the end of 2020 (Fig A3, (b)), after which the variation is significantly reduced, indicating the effect of a Nafion dryer installed to the inlet of the ICOS instrument. In addition, the variation in the differences especially during summer is reduced when the ICOS sample is dried. This suggests that the seasonal variation is driven by varying humidity from summer to winter. As for $CO_2$, there is a negative correlation between $\Delta CH_4$ and the water vapor concentration of about -0.80 ppb/v% (intercept 1.35 v%, p < 0.05, $R^2$ = 0.40) when both of the instruments measure wet air (Fig. A5, (a)). After the ICOS instrument sample is dried, the

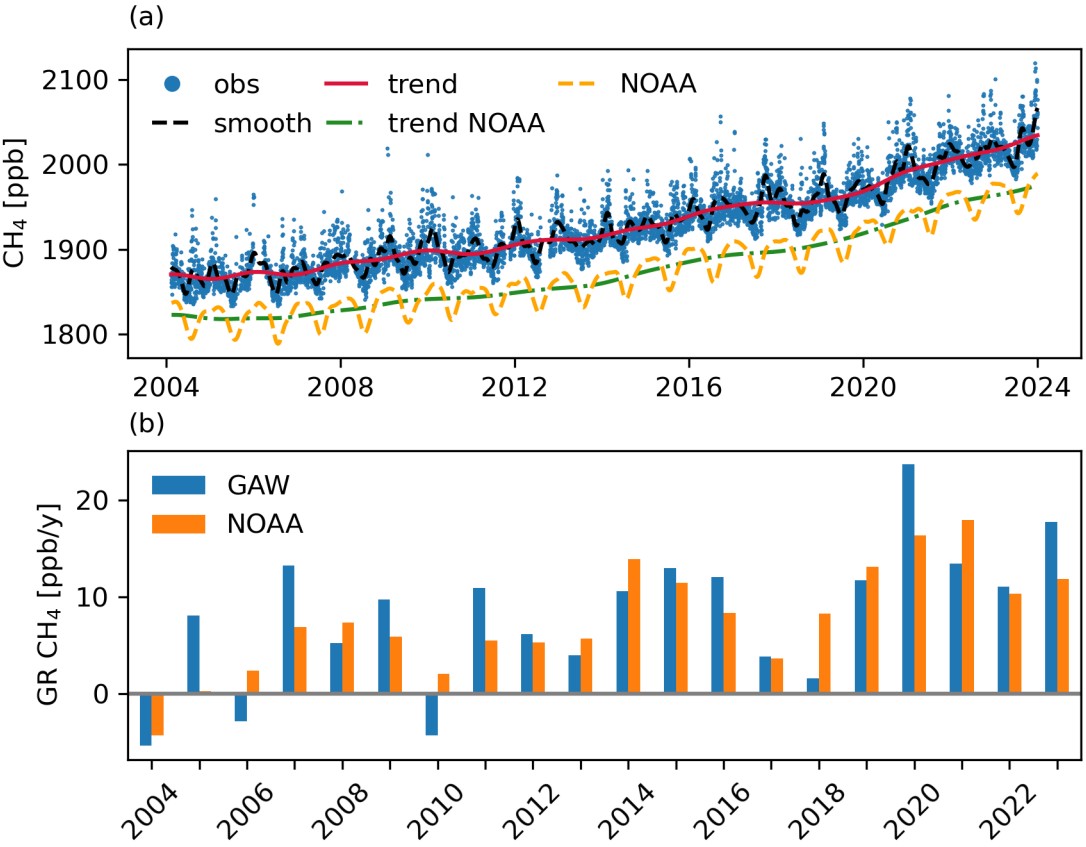

**Figure 10.** Panel (a): Time series of $CH_4$. The blue dots indicate the daily observed values from the GAW instrument, the red line the trend value derived from the GAW observations and the black dashed line the smoothed line from the GAW observations. The yellow dashed line indicates the MBL data and the green dotted line the trend derived from the MBL. Panel (b): The yearly growth rates of $CH_4$ for GAW measurements (blue) and the NOAA MBL (orange).

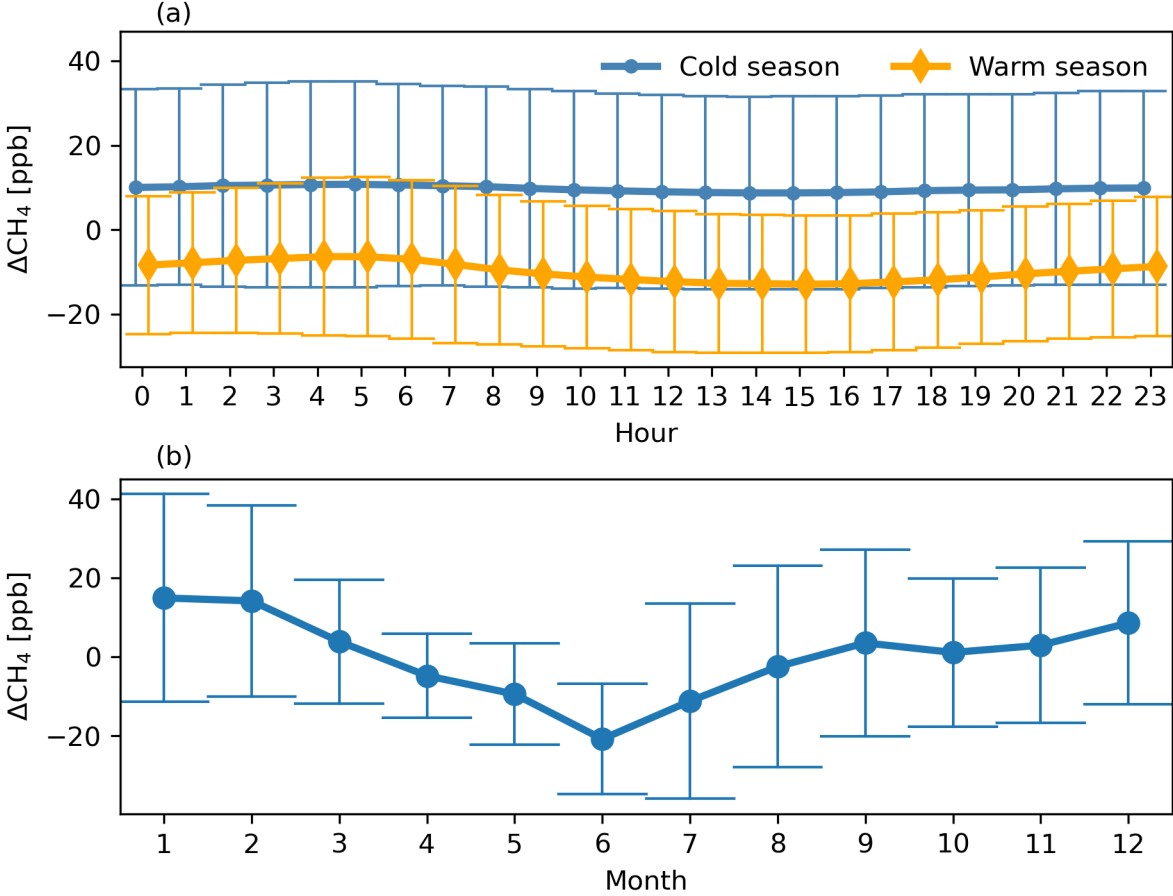

**Figure 11.** Average diurnal cycle of the during the warm and cold seasons for $CH_4$ (a) and average seasonal variation of $CH_4$ (b). Both are calculate from de-trended timeseries. Data points are hourly (a) and monthly (b) means with associated standard deviations.

correlation is slightly positive of about 0.22 ppb/v% (intercept -0.02, $p < 0.05$, $R^2 = 0.06$). When measuring wet samples, there is a slight correlation with mole fraction of about 0.01 ppb/ppb (intercept -9.37, $R^2 = 0.06$), and when the ICOS instrument
sample is dried no significant mole fraction dependency is observed (Fig. A5, (b)). As the drying process eliminates most of the moisture from the sample, the variation is reduced. However, the strong variation due to the sample humidity seems to be solely caused by the ICOS instrument, since the GAW instrument always measures the sample air wet. The discrepancy could also be caused by better performance of water vapor correction of the GAW instrument compared to the ICOS instrument. Before the installation of the Nafion dryer the mean hourly difference is 0.77 ppb (95% CI: [-0.34, 1.69] ppb) and the mean
daily difference is 0.76 ppb (95% CI: [-0.27, 1.54] ppb); after the Nafion installation the mean hourly difference drops to 0.14 ppb (95% CI: [-0.53, 0.90] ppb) and the daily mean difference drops to 0.16 ppb (95% CI: [-0.38, 0.74] ppb). Before the Nafion installation, 99% of the measured hours and 99.7% of the measured days were within the WMO compatibility goals;

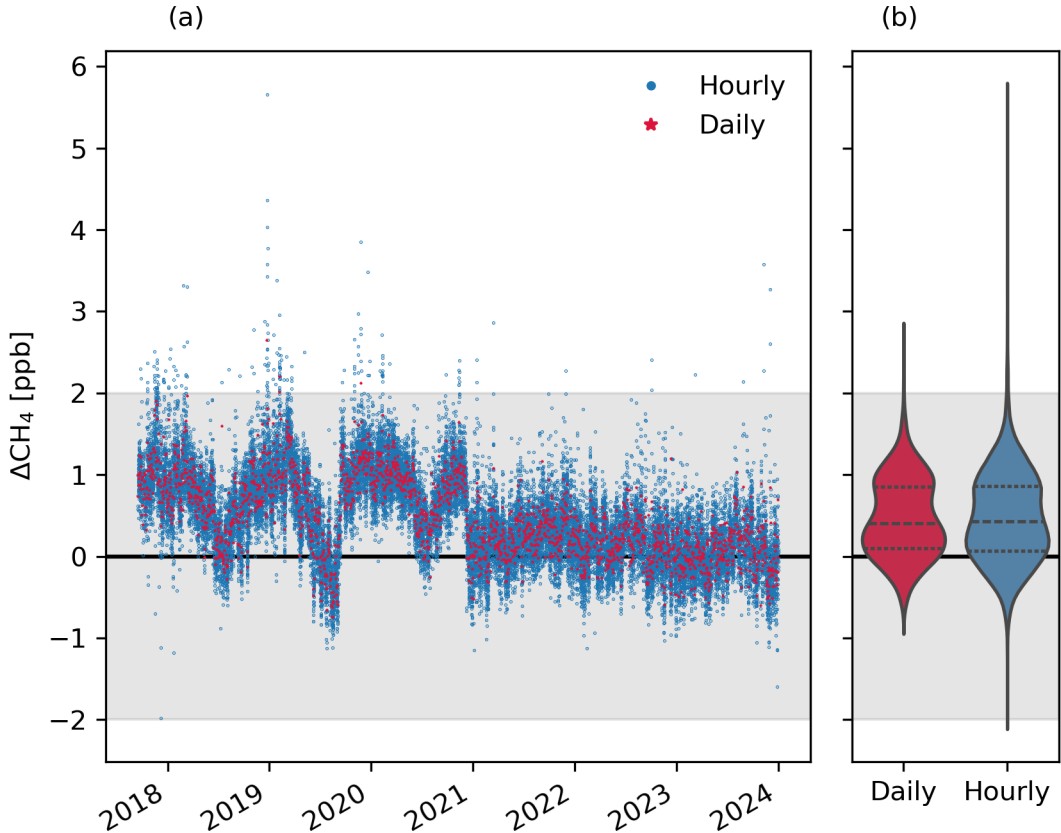

**Figure 12.** Panel (a): Differences between the ICOS and GAW instruments for $CH_4$ (GAW-ICOS). The blue dots indicate the hourly mean values and red dots the daily mean values. Positive/negative values indicates $CH_4$ measured by the ICOS instruments are higher/lower than those from GAW. Panel (b): distribution of the data points. The gray shaded areas indicate the WMO/GAW compatibility goals

after installation 99.9% of the hours and virtually all (100%) of the days were within limits. The target measurements of $CH_4$, presented in table 5 show a consistent STR across the different instruments and cylinders. The LTR for the ICOS ST cylinder D348368 is rather high compared to the rest of the target cylinders, similarly to $CO_2$.

The agreement of the two fitted trend lines was good, with the mean difference being 0.3 ppb. Calculating the yearly growth rates from both trend lines also agree well with mean difference of 0.1 ppb/year.

When the data are filtered to include afternoon hours only, the differences between the two systems is 0.46 ppb (95% CI: [-0.43, 1.52] ppb), which is slightly higher but not significantly different from the filtering based on wind speed and hourly standard deviation.

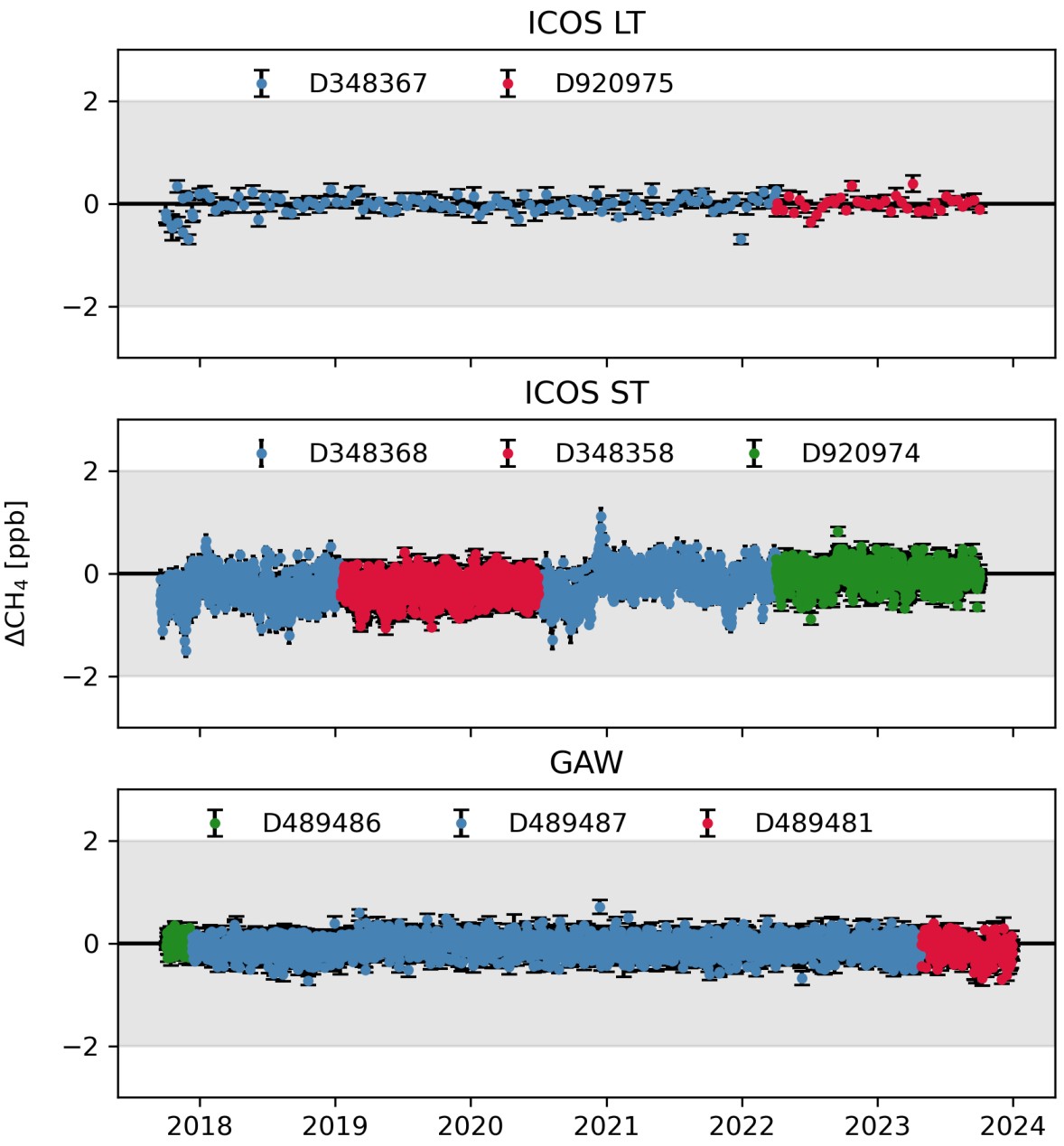

**Figure 13.** CH$_4$ target measurements of ICOS LT, ICOS ST and GAW target cylinders over the whole comparison period. Different cylinders used are marked with distinct colors. The data are given as means of each sequence with the associated standard deviation.

### 4.2.3 Audit results for $CH_4$

The results of the combined ICOS and GAW audit for $CH_4$ are presented in Fig. 14. As for the $CO_2$ comparison, the measurements are on an hourly resolution. The ICOS Mobile Laboratory hourly data is calculated from the minute data matching the ICOS data (i.e., minutes not measured by the local ICOS instrument due to calibration etc. are not included in the Mobile Laboratory hourly means), and the WCC-Empa hourly data is similarly matched to the GAW Picarro data. In the analysis, the ICOS Mobile Laboratory data is measured with G2401 instrument.

Results of the audit are summarized in table 4. The mean difference of each comparison is well within the WMO/GAW compatibility goals. The largest differences are observed between GAW and WCC and between GAW and Mobile Laboratory. It seems that the GAW instrument is measuring slightly higher values compared to the other instruments. This could be an issue of the sampling line or inlet, as the GAW instrument is the only instrument sampling from a different location than the rest of the instruments. Filtering the audit data for $CH_4$ for wind speed and hourly standard deviation leads to similar results as for $CO_2$, decreasing the spread between GAW and ICOS Mobile Laboratory to 1.40 ppb, between ICOS and WCC to 0.88 ppb and between ICOS and GAW to 1.29 ppb. Between WCC and ICOS Mobile Laboratory, the spread is only reduced to 1.31 ppb and between GAW and WCC to 0.98 ppb, and no significant difference in the spread is noticed between ICOS and ICOS Mobile Laboratory. Furthermore, all the instruments are calibrated using a separate set of calibration standards, leading to slight differences in the calibrated values. To quantify this effect, during the audit the ICOS instrument measured the ICOS Mobile Laboratory calibration standards, and the GAW instrument measured the WCC travelling standards. The results of the calibrations are presented in tables A1 and A2. For $CH_4$, the ICOS instrument measured 0.12 to 0.54 ppb lower values than the assigned values of the cylinder, and the GAW instrument measured 0.16 to 0.72 ppb higher values. However, despite these differences in the setup, sampling and calibration, the differences between the instruments remain small. The spreads of the differences are also consistent between the instruments as well, and all the 95% intervals are within the compatibility goals. The largest spreads are found between the GAW and the ICOS Mobile Laboratory and between WCC and ICOS Mobile Laboratory instruments. The mole fraction dependency of the differences between the different comparison pairs for $CH_4$ is presented in Fig A3. For $CH_4$, none of the comparison pairs show a significant mole fractional dependency.

The $CH_4$ measurements of the target cylinders during the audit are presented in Fig. 15. For ICOS instrument, both ST (D348368) and LT (D348367) are shown. No signficant drift is evident from the time series for any of the instruments. The STR and LTR, as well as the cylinder biases are presented in table 5. During the audit, the STR is comparable between all the instruments, and the LTR of GAW, WCC and ICOS ST and LT cylinders show similar values. Slightly higher LTR is found in the ICOS Mobile Laboratory cylinder measurements.

As for $CO_2$, Hammer et al. (2013) compared $CH_4$ measurements as well. For $CH_4$ the results show median bias of -0.3 [-0.51, 0.51] ppb against the reference instrument at HEI, 0.41 [-0.77, 1.78] ppb at CBW and 0.44 [-0.28, 1.15] ppb at OPE (brackets referring to 5% and 95% quantiles). Our comparisons for $CH_4$ show similar results between GAW-MobileLab, GAW-WCC and ICOS-GAW comparisons. The other study by Vardag et al. (2014) found median differences of -0.04 $\pm$ 3.38 ppb and 0.12 $\pm$ 0.25 ppb at MHD, when comparing the traveling instrument to the two local instruments at MHD. At HEI the median

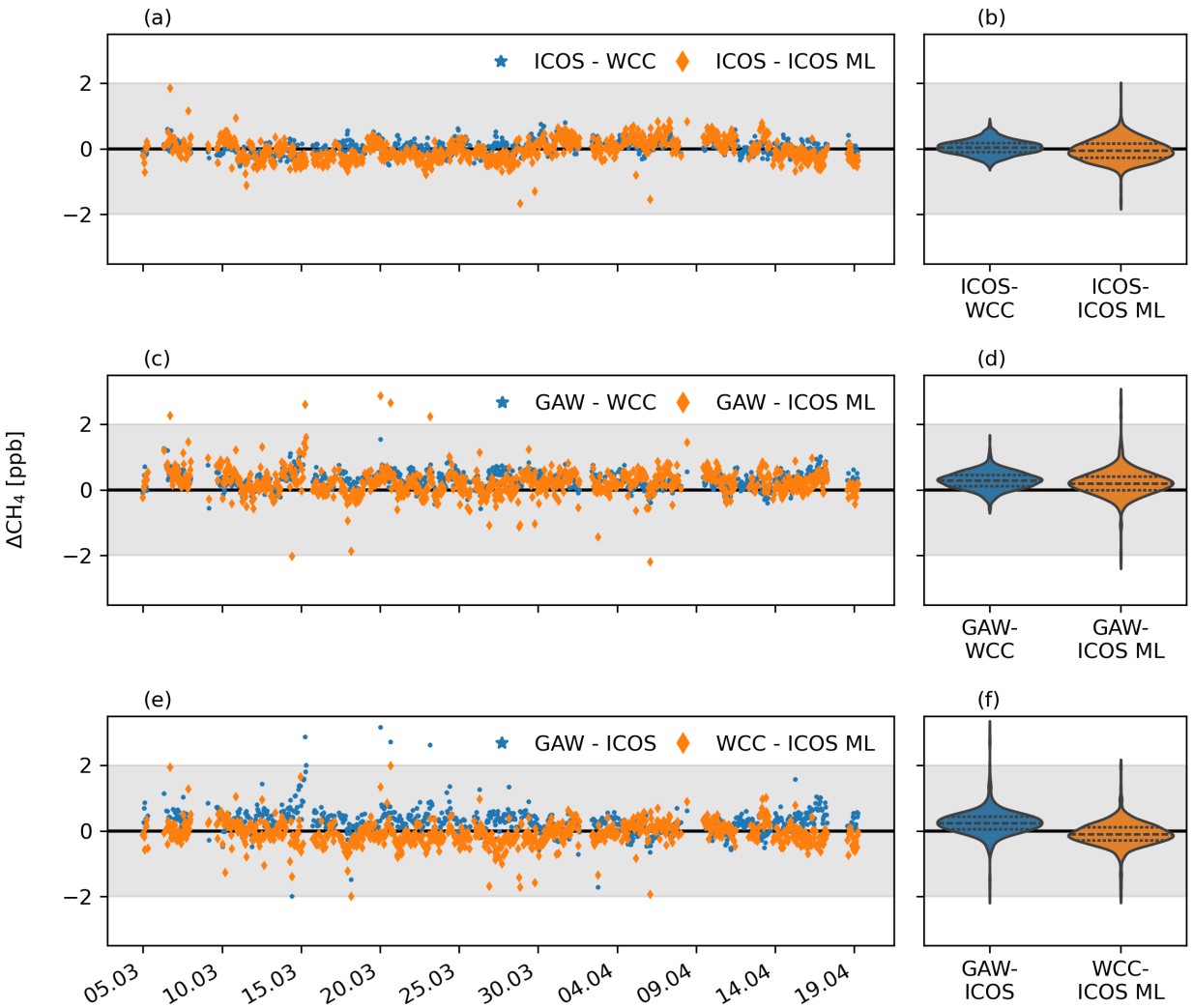

**Figure 14.** Time series of the differences for different comparison pairs (a, c and e), and their associated distributions (b, d and f) for CH$_4$. In the legend ML refers to Mobile Laboratory.

difference of CH$_4$ was -0.25 $\pm$ 3.16 ppb before the campaign at MHD and -0.24 $\pm$ 2.43 ppb after. Earlier GAW audits in CVO and MHD for CH$_4$ are presented in Zellweger et al. (2016). In CVO the same instrument (OA-ICOS) was used as for CO$_2$. In MHD the measurement method was GC-FID. The results show a mean deviation ($\pm$ standard deviation) of -0.61 $\pm$ 0.32 ppb at CVO and 0.22 $\pm$ 3.59 ppb and MHD. Comparison of the traveling CRDS against a local CRDS at Pallas in 2012 is presented in Rella et al. (2013), with a mean deviation of -0.032 $\pm$ 0.367 ppb.

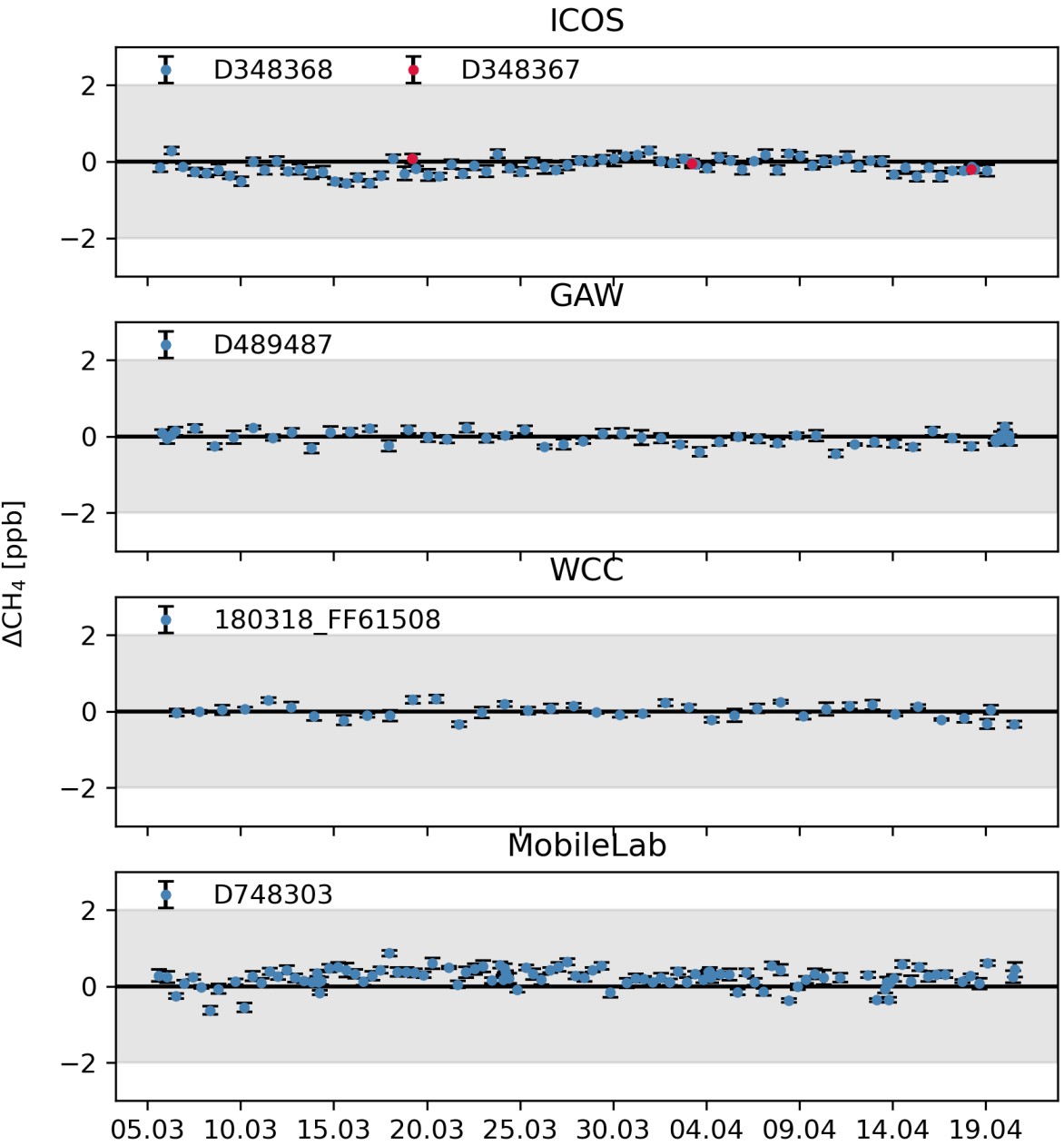

**Figure 15.** $CH_4$ target measurements for ICOS, GAW, WCC and Mobile Laboratory instruments during the audit period. For ICOS both ST and LT are presented. The data are given as means of each sequence with the associated standard deviation.

|  | Mean (ppb) | 95% CI (ppb) | CI range (ppb) | Slope | Intercept |
|---|---|---|---|---|---|
| **Full period** | | | | | |
| ICOS - GAW | 0.55 | -0.4 to 1.16 | 1.56 | 0.994 | 11.62 |
| **Audit period** | | | | | |
| ICOS-Mobile Lab | -0.04 | -0.59 to 0.62 | 1.20 | 0.999 | 2.25 |
| GAW-Mobile Lab | -0.24 | -0.44 to 1.16 | 1.60 | 1.002 | -4.06 |
| WCC-Mobile Lab | -0.08 | -0.76 to 0.58 | 1.34 | 1.000 | -0.66 |
| ICOS-WCC | 0.05 | -0.44 to 0.54 | 0.98 | 0.999 | 2.89 |
| GAW-WCC | 0.30 | -0.25 to 0.85 | 0.98 | 1.001 | -1.26 |
| GAW-ICOS | 0.27 | -0.38 to 1.02 | 1.40 | 0.996 | 6.93 |

**Table 4.** Results of the cross-comparisons of different instrumentations on hourly mean data over the full period and during the audit period for $CH_4$. For the full period, the data is filtered for wind speed, however during the audit period no wind speed filtering is applied.

|  | Cylinder | Purpose | LTR (ppb) | STR (ppb) | Bias (ppb) | Conc (ppb) | Nb | Measure time (min) |
|---|---|---|---|---|---|---|---|---|
| **Full period** | | | | | | | | |
| ICOS | D348367 | LT | 0.18 | 0.11 | -0.03 | 2097.11 | 10 | 20 |
|  | D920975 | LT | 0.14 | 0.10 | -0.01 | 2196.16 | 10 | 20 |
|  | D348358 | ST | 0.24 | 0.11 | -0.31 | 1949.49 | 10 | 20 |
|  | D348368 | ST | 0.30 | 0.12 | -0.24 | 1948.65 | 10 | 20 |
|  | D920974 | ST | 0.22 | 0.10 | -0.04 | 1936.69 | 10 | 20 |
| GAW | D489481 | | 0.21 | 0.11 | -0.12 | 1961.42 | 9 | 18 |
|  | D489486 | | 0.15 | 0.10 | -0.01 | 1686.86 | 9 | 18 |
|  | D489487 | | 0.18 | 0.10 | -0.06 | 1938.79 | 9 | 18 |
| **Audit period** | | | | | | | | |
| ICOS | D348367 | LT | 0.14 | 0.11 | -0.06 | 2097.11 | 10 | 20 |
| ICOS | D348368 | ST | 0.20 | 0.11 | -0.12 | 1974.51 | 10 | 20 |
| GAW | D489487 | | 0.17 | 0.10 | -0.04 | 1938.79 | 9 | 18 |
| WCC | 180318_FF61508 | | 0.18 | 0.09 | <0.01 | 1963.81 | 4 | 9 |
| MobileLab | D748303 | | 0.25 | 0.10 | 0.23 | 1937.38 | 8 | 20 |

**Table 5.** Results of the target measurements of $CH_4$ for each instrument for the full period of comparisons between ICOS and GAW instruments and for the audit period. Nb refers to the number of data points used for averaging and measure time is the total time each cylinder is measured during one injection.

## 5 Summary

In this article, we present the measurements of $CO_2$ and $CH_4$ at the Pallas station located in the Pallas-Yllästunturi national park in Finnish Lapland, as well as a comparison of the two measurement setups at the station. A comprehensive description of the measurement system is presented in Sect. 1, including the used instruments, calibration and target cylinders. The time series of the GAW measurements for $CO_2$ and $CH_4$ are presented with a smoothed and trend curves fitted to the time series. We also present the diurnal and seasonal variations of the de-trended time series. The time series show, as expected, a rise in $CO_2$ and $CH_4$, in agreement with global tendencies. The measured mole fractions of both $CO_2$ and $CH_4$ at Pallas are higher than the average in the Northern Hemisphere, owing to the station's location in high latitude where the greenhouse gas mole fractions are generally higher than the average. The growth rates at Pallas and the average in the Northern Hemisphere agree in general. However, especially the growth rate of $CH_4$ shows large year-to-year variation, and these variations are more pronounced at Pallas compared to the NOAA MBL. Especially in 2020, the growth rate of $CH_4$ is significantly higher than the average in the Northern Hemisphere.

The observed differences between the two separate measuring systems at Pallas show a mean difference of less than 0.01 ppm for daily $CO_2$ averages and 0.55 ppb for daily $CH_4$ averages when filtering the measurements to only contain the background signal. An improvement in the agreement between the systems was observed with the addition of Nafion dryer on the intake line of the ICOS instrument. Especially for the $CH_4$ measurements the improvement is clear, the difference before drying the sample is 0.76 ppb on average and 0.21 ppb after. In the $CO_2$ measurements the effect is less pronounced, with the difference before adding the Nafion being 0.02 ppm and -0.02 ppm afterward. However, a larger proportion of the data fits within the WMO/GAW compatibility goals after the addition of a dryer, emphasizing the benefit of sample drying. At Pallas, the agreement between the instruments could likely be further improved by drying the sample air of the GAW instrument as well.

Furthermore, the biases observed between the different systems during the ICOS Mobile Laboratory and WCC audits at the station are all shown to be within the WMO/GAW goals. The largest differences were observed between the GAW and WCC systems in both $CO_2$ and $CH_4$, however the largest spread (CI range) in the differences was between ICOS and GAW and between WCC and ICOS Mobile Laboratory for $CO_2$ and between GAW and the ICOS Mobile Laboratory for $CH_4$. This is partly expected, as the GAW system is the only one measuring from its own inlet and all the other systems are connected to the same inlet. In addition, the GAW instrument is still measuring the sample wet, while the other instruments are drying the sample to different degrees (ICOS and WCC with a Nafion and the ICOS Mobile Laboratory with a freeze dryer). No significant differences in the LTR was observed for $CO_2$ or for $CH_4$ across the instruments during the audit. However, even with a slightly different sampling location, the measurements between GAW and the other systems agree well, indicating that the air at Pallas is generally well mixed. During the audit the difference between the ICOS and GAW instruments is comparable to the difference over the whole period, when the ICOS instrument is sampling dried air. However, the spread of the differences over the whole period is slightly larger than only during the audit. For $CH_4$, the mean difference as well as the spread over the entire period is larger compared to the audit period. This could be a seasonal effect, as the audit took place in spring when

natural $CH_4$ emissions and $CH_4$ sinks are lower. When filtering the audit data for the wind speed and hourly standard deviation, the spreads between the GAW instrument and the rest generally decrease. Filtering the data by afternoon hours only does not have a large effect on the daily averages when compared to the filtering method based on wind speed and hourly standard deviation. This indicates that the air is generally well mixed during the afternoon hours and that there is little local influence on the mole fractions.

Our results highlight the good accuracy of the measurements conducted at the Pallas station, as the differences between the two measurement systems are small and fit well within the WMO/GAW network compatibility goals.

The better measurement agreement with the two auditing units suggest a better performance of the ICOS system. However, the LTR of the GAW instrument is similar to the LTR of the ICOS instrument, when measuring the ICOS LT cylinders. The LTR of the ICOS ST cylinders is worse, however this could be also caused by cylinder drift. Furthermore, the trend and growth rates at Pallas differ from the NOAA marine boundary layer trend for the Northern Hemisphere, especially for $CH_4$, demonstrating the importance of atmospheric in-situ observations for detecting regional and local variations in greenhouse gas mole fractions.

*Data availability.* ICOS $CO_2$ data were downloaded from the ICOS Carbon portal, DOI: 11676/W1KxBw4QLCVKxiEiOVSPoLCU (Hatakka, J. 2024). ICOS $CH_4$ data were downloaded from the ICOS Carbon portal, DOI: 11676/lMY9pSZLevM3UXmNVKiKl4WH (Hatakka, J. 2024). GAW $CO_2$ data were downloaded from the WDCGG, published as CO2_PAL_surface-insitu_FMI_data1 ver. 2024-06-19-0538 at WDCGG (Reference date: 2024/10/15). GAW $CH_4$ data were downloaded from the WDCGG, published as CH4_PAL_surface-insitu_FMI_data1, at WDCGG ver. 2024-06-19-0538 (Reference date: 2024/10/15). The Mobile Laboratory data and GAW audit data are available from the authors on request.

## Appendix A

| | CAL 1 | CAL 2 | CAL 3 |
|---|---|---|---|
| $CO_2$, assigned [ppm] | 379.24 | 414.46 | 449.39 |
| $CO_2$, measured [ppm] | 379.21 | 414.37 | 449.40 |
| $\Delta CO_2$ [ppm] | -0.03 | -0.09 | 0.01 |
| | | | |
| $CH_4$, assigned [ppb] | 1985.48 | 1799.53 | 2210.77 |
| $CH_4$, measured [ppb] | 1985.36 | 1798.99 | 2210.63 |
| $\Delta CH_4$ [ppb] | -0.12 | -0.54 | -0.14 |

**Table A1.** Cross-calibration of the ICOS Mobile Laboratory calibration standards with the Pallas ICOS instrument: The assigned values of the cylinders, average measured values with the GAW instrument, and the difference of measured value to the assigned value.

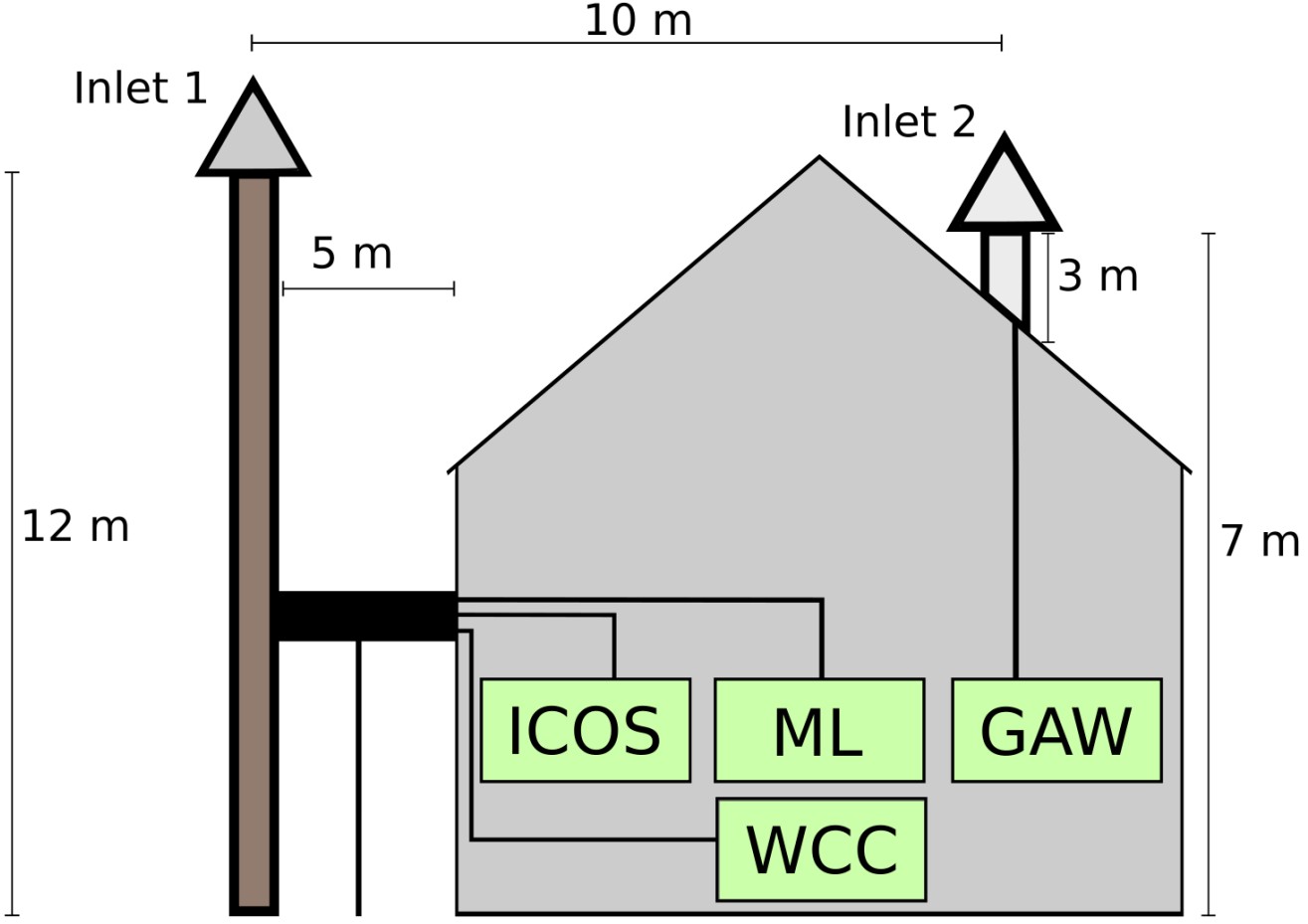

**Figure A1.** Schematics of the measurement setup during the audit at Pallas. ML refers to the ICOS Mobile Laboratory instrument

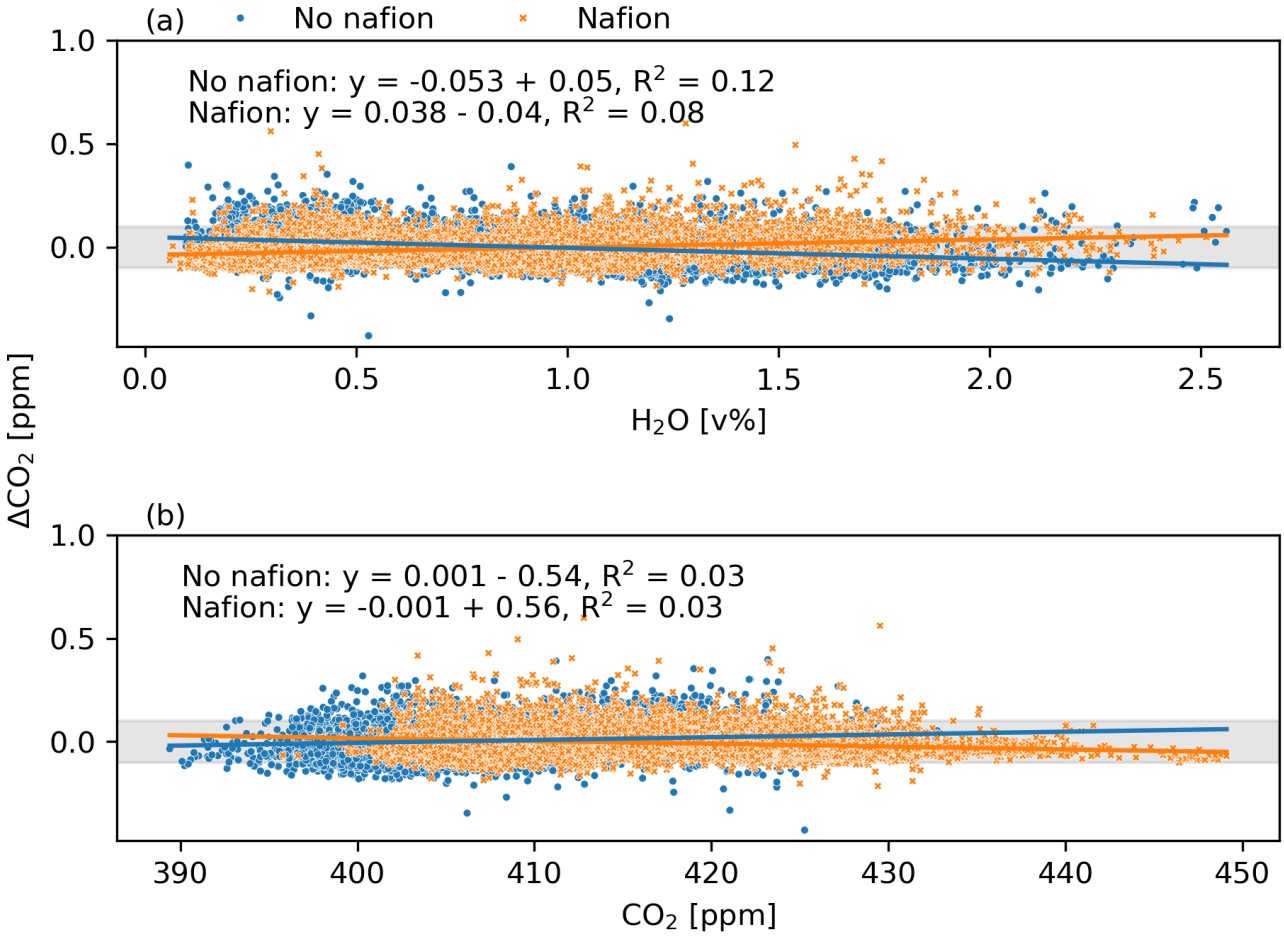

**Figure A2.** Dependency of the $CO_2$ difference (GAW-ICOS) on water vapor concentration (a) and mole fraction (b). The data is split into two groups: before the installation of the Nafion and after.

|                              | CAL 1   | CAL 2   | CAL 3   | CAL 4   | CAL 5   | CAL 6   | CAL 7   |
|------------------------------|---------|---------|---------|---------|---------|---------|---------|
| $CO_2$, assigned [ppm]       | 378.12  | 387.39  | 406.99  | 411.21  | 417.53  | 412.70  | 427.81  |
| $CO_2$, measured [ppm]       | 387.15  | 387.39  | 407.01  | 411.18  | 417.48  | 412.67  | 427.80  |
| $\Delta CO2$ [ppm]           | 0.03    | 0.00    | 0.02    | -0.03   | -0.05   | -0.03   | -0.01   |
|                              |         |         |         |         |         |         |         |
| $CH_4$, assigned [ppb]       | 1883.44 | 1890.78 | 1933.20 | 1953.82 | 1963.81 | 1998.97 | 2191.22 |
| $CH_4$, measured [ppb]       | 1884.16 | 1891.23 | 1933.69 | 1954.13 | 1964.16 | 1999.13 | 2191.50 |
| $\Delta CH_4$ [ppb]          | 0.72    | 0.45    | 0.49    | 0.31    | 0.35    | 0.16    | 0.28    |

**Table A2.** Cross-calibration of the GAW travelling standards with the Pallas GAW instrument: The assigned values of the cylinders, average measured values with the GAW instrument, and the difference of measured value to the assigned value.

*Author contributions.* AL and HA planned the experiment. AL, HA and JH set up the ICOS Mobile Laboratory audit campaign. CZ and JH set up the GAW audit campaign. HA provided the Mobile Laboratory audit data. CZ provided the GAW audit data. AL analyzed the data. AL prepared the manuscript. AL, HA, CZ, AT, TA and JH contributed to the scientific discussion and preparation of the manuscript.

*Competing interests.* The authors declare that they have no conflict of interest.

*Acknowledgements.* This work has been supported by the The Atmosphere and Climate Competence Center (IL: Academy of Finland Flagship funding (grant no. 337552, 357904). We thank the ICOS ATC team for the processing of the ICOS instrument data. We would like to thank the ICOS ATC for providing the data on $CO_2$ and $CH_4$.

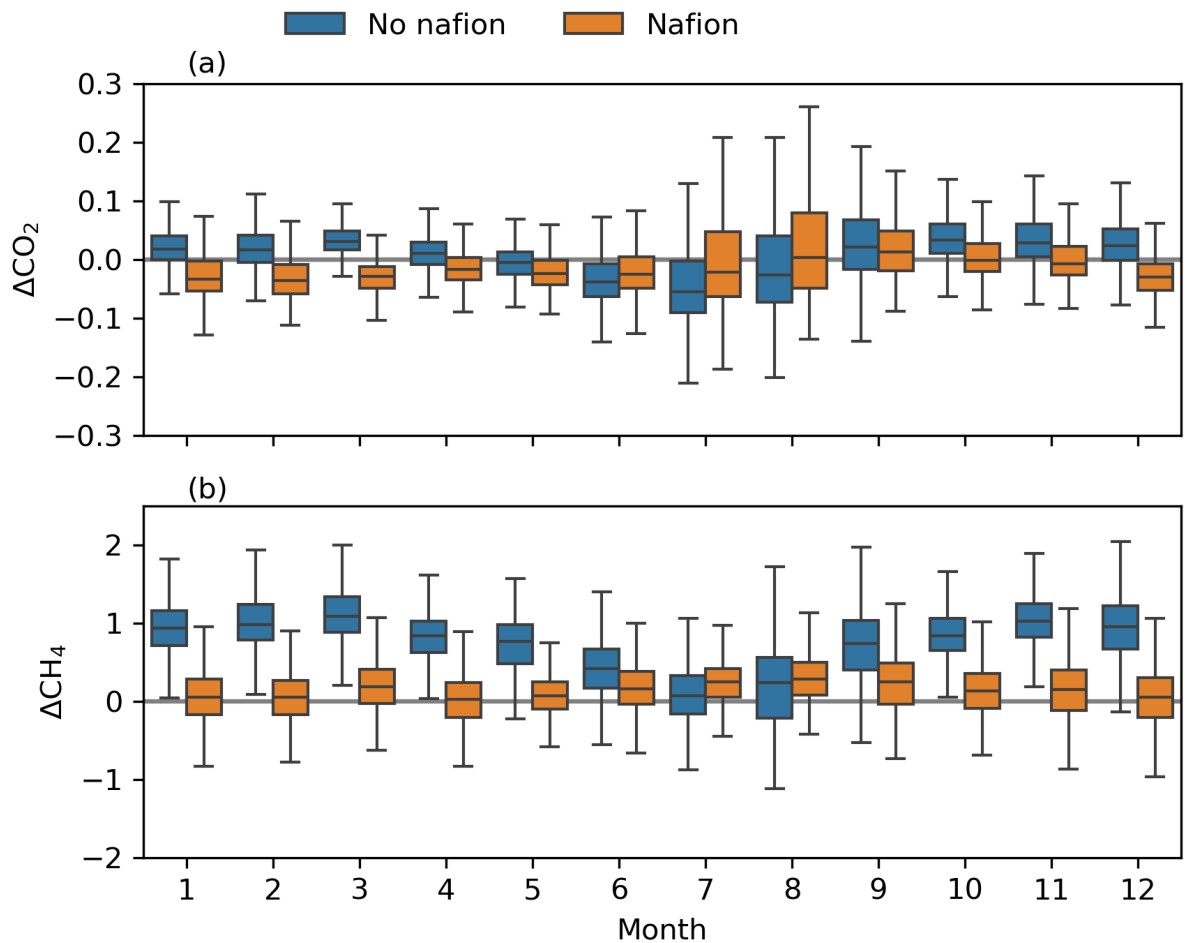

**Figure A3.** Seasonal variation of the differences (GAW-ICOS) in $CO_2$ (a) and $CH_4$ (b) before and after installation of the Nafion.

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

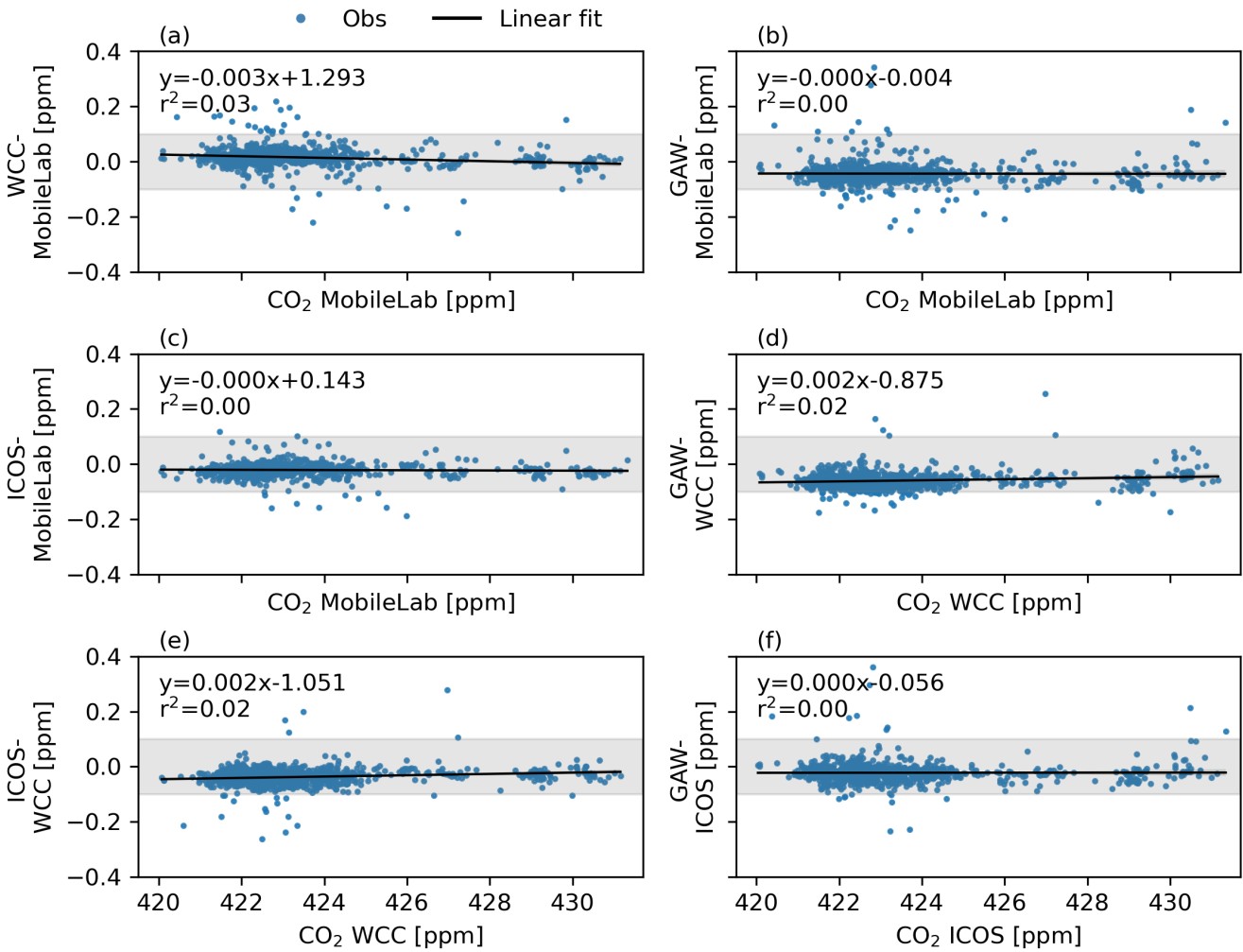

**Figure A4.** Mole fraction dependency of the difference between each comparison pair for $CO_2$. Linear regression fitted to the data.

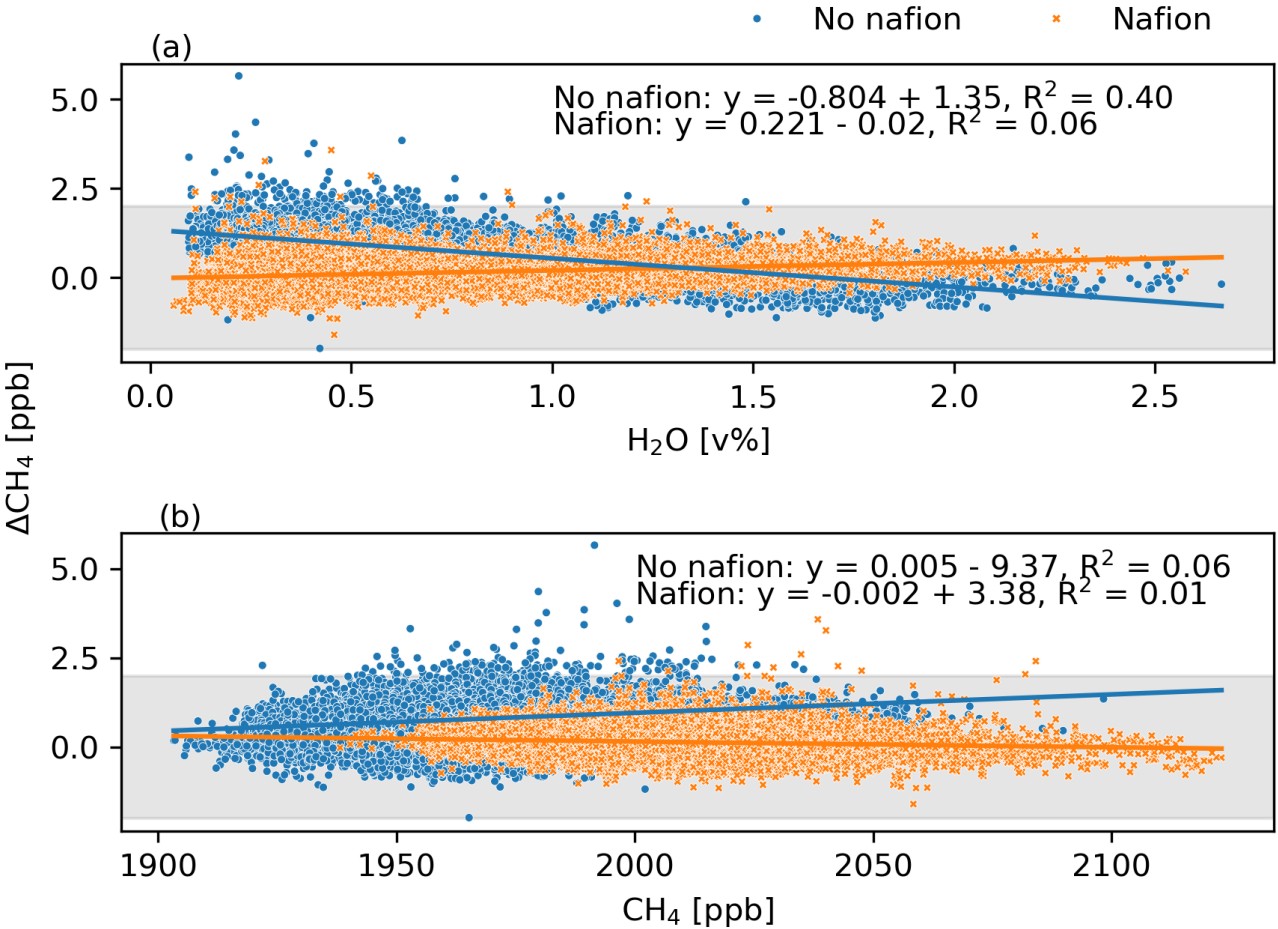

**Figure A5.** Dependency of the CH$_4$ difference (GAW-ICOS) on water vapor concentration (a) and mole fraction (b). The data is split into two groups: before the installation of the Nafion and after.

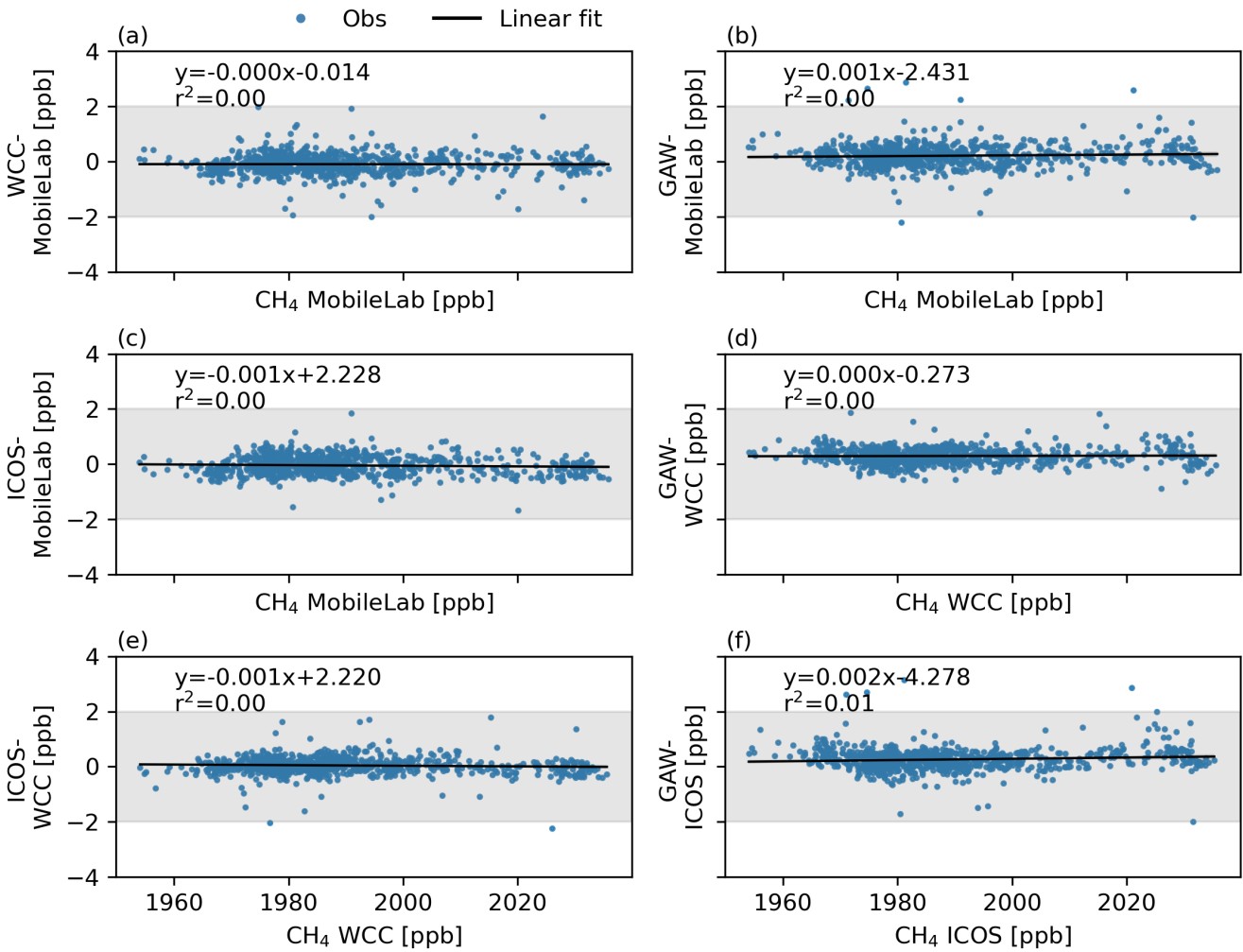

**Figure A6.** Mole fraction dependency of the difference between each comparison pair for $CH_4$. Linear regression fitted to the data.

Andrews, A. E., Kofler, J. D., Trudeau, M. E., Williams, J. C., Neff, D. H., Masarie, K. A., Chao, D. Y., Kitzis, D. R., Novelli, P. C., Zhao, C. L., Dlugokencky, E. J., Lang, P. M., Crotwell, M. J., Fischer, M. L., Parker, M. J., Lee, J. T., Baumann, D. D., Desai, A. R., Stanier, C. O., De Wekker, S. F. J., Wolfe, D. E., Munger, J. W., and Tans, P. P.: $CO_2$ , CO, and $CH_4$ measurements from tall towers in the NOAA Earth System Research Laboratory's Global Greenhouse Gas Reference Network: instrumentation, uncertainty analysis, and recommendations for future high-accuracy greenhouse gas monitoring efforts, Atmospheric Measurement Techniques, 7, 647–687, https://doi.org/10.5194/amt-7-647-2014, 2014.

Byrne, B., Baker, D. F., Basu, S., Bertolacci, M., Bowman, K. W., Carroll, D., Chatterjee, A., Chevallier, F., Ciais, P., Cressie, N., Crisp, D., Crowell, S., Deng, F., Deng, Z., Deutscher, N. M., Dubey, M. K., Feng, S., García, O. E., Griffith, D. W. T., Herkommer, B., Hu, L., Jacobson, A. R., Janardanan, R., Jeong, S., Johnson, M. S., Jones, D. B. A., Kivi, R., Liu, J., Liu, Z., Maksyutov, S., Miller, J. B., Miller, S. M., Morino, I., Notholt, J., Oda, T., O'Dell, C. W., Oh, Y.-S., Ohyama, H., Patra, P. K., Peiro, H., Petri, C., Philip, S., Pollard, D. F., Poulter, B., Remaud, M., Schuh, A., Sha, M. K., Shiomi, K., Strong, K., Sweeney, C., Té, Y., Tian, H., Velazco, V. A., Vrekoussis, M., Warneke, T., Worden, J. R., Wunch, D., Yao, Y., Yun, J., Zammit-Mangion, A., and Zeng, N.: National $CO_2$ budgets (2015–2020) inferred from atmospheric $CO_2$ observations in support of the global stocktake, Earth System Science Data, 15, 963–1004, https://doi.org/10.5194/essd-15-963-2023, 2023.

Dlugokencky, E. J., Myers, R. C., Lang, P. M., Masarie, K. A., Crotwell, A. M., Thoning, K. W., Hall, B. D., Elkins, J. W., and Steele, L. P.: Conversion of NOAA atmospheric dry air $CH_4$ mole fractions to a gravimetrically prepared standard scale, Journal of Geophysical Research: Atmospheres, 110, 2005JD006 035, https://doi.org/10.1029/2005JD006035, 2005.

Feng, L., Palmer, P. I., Parker, R. J., Lunt, M. F., and Bösch, H.: Methane emissions are predominantly responsible for record-breaking atmospheric methane growth rates in 2020 and 2021, Atmospheric Chemistry and Physics, 23, 4863–4880, https://doi.org/10.5194/acp-23-4863-2023, 2023.

Friedlingstein, P., O'Sullivan, M., Jones, M. W., Andrew, R. M., Hauck, J., Landschützer, P., Le Quéré, C., Li, H., Luijkx, I. T., Olsen, A., Peters, G. P., Peters, W., Pongratz, J., Schwingshackl, C., Sitch, S., Canadell, J. G., Ciais, P., Jackson, R. B., Alin, S. R., Arneth, A., Arora, V., Bates, N. R., Becker, M., Bellouin, N., Berghoff, C. F., Bittig, H. C., Bopp, L., Cadule, P., Campbell, K., Chamberlain, M. A., Chandra, N., Chevallier, F., Chini, L. P., Colligan, T., Decayeux, J., Djeutchouang, L. M., Dou, X., Duran Rojas, C., Enyo, K., Evans, W., Fay, A. R., Feely, R. A., Ford, D. J., Foster, A., Gasser, T., Gehlen, M., Gkritzalis, T., Grassi, G., Gregor, L., Gruber, N., Gürses, \., Harris, I., Hefner, M., Heinke, J., Hurtt, G. C., Iida, Y., Ilyina, T., Jacobson, A. R., Jain, A. K., Jarníková, T., Jersild, A., Jiang, F., Jin, Z., Kato, E., Keeling, R. F., Klein Goldewijk, K., Knauer, J., Korsbakken, J. I., Lan, X., Lauvset, S. K., Lefèvre, N., Liu, Z., Liu, J., Ma, L., Maksyutov, S., Marland, G., Mayot, N., McGuire, P. C., Metzl, N., Monacci, N. M., Morgan, E. J., Nakaoka, S.-I., Neill, C., Niwa, Y., Nützel, T., Olivier, L., Ono, T., Palmer, P. I., Pierrot, D., Qin, Z., Resplandy, L., Roobaert, A., Rosan, T. M., Rödenbeck, C., Schwinger, J., Smallman, T. L., Smith, S. M., Sospedra-Alfonso, R., Steinhoff, T., Sun, Q., Sutton, A. J., Séférian, R., Takao, S., Tatebe, H., Tian, H., Tilbrook, B., Torres, O., Tourigny, E., Tsujino, H., Tubiello, F., van der Werf, G., Wanninkhof, R., Wang, X., Yang, D., Yang, X., Yu, Z., Yuan, W., Yue, X., Zaehle, S., Zeng, N., and Zeng, J.: Global Carbon Budget 2024, Earth System Science Data, 17, 965–1039, https://doi.org/10.5194/essd-17-965-2025, 2025.

Hall, B. D., Crotwell, A. M., Kitzis, D. R., Mefford, T., Miller, B. R., Schibig, M. F., and Tans, P. P.: Revision of the World Meteorological Organization Global Atmosphere Watch (WMO/GAW) $CO_2$ calibration scale, Atmospheric Measurement Techniques, 14, 3015–3032, https://doi.org/10.5194/amt-14-3015-2021, 2021.

Hammer, S., Konrad, G., Vermeulen, A. T., Laurent, O., Delmotte, M., Jordan, A., Hazan, L., Conil, S., and Levin, I.: Feasibility study of using a "travelling" $CO_2$ and $CH_4$ instrument to validate continuous in situ measurement stations, Atmospheric Measurement Techniques, 6, 1201–1216, https://doi.org/10.5194/amt-6-1201-2013, 2013.

Hatakka, J.: Atmospheric CH4 at Pallas by Finnish Meteorological Institute , dataset published as CH4_PAL_ surface-insitu_FMI_data1 at WDCGG, ver. 2024-06-19-0538 (Reference date: 2024/10/15), 2024a.

Hatakka, J.: Atmospheric CO2 at Pallas by Finnish Meteorological Institute , dataset published as CO2_PAL_ surface-insitu_FMI_data1 at WDCGG, ver. 2024-06-19-0538 (Reference date: 2024/10/15), 2024b.

Hatakka, J.: ICOS ATC CH4 Release, Pallas (12.0 m), 2017-09-16–2024-03-31, https://doi.org/11676/lMY9pSZLevM3UXmNVKiKl4WH, 2024c.

Hatakka, J.: ICOS ATC CO2 Release, Pallas (12.0 m), 2017-09-16–2024-03-31, https://doi.org/11676/W1KxBw4QLCVKxiEiOVSPoLCU,
2024d.

Hatakka, J., Aalto, T., Aaltonen, V., Aurela, M., Hakola, H., Komppula, M., Laurila, T., Lihavainen, H., Paatero, J., Salminen, K., and Viisanen, Y.: Overview of the atmospheric research activities and results at Pallas GAW station, BOREAL ENVIRONMENT RESEARCH, 8, 365–383, 2003.

Hazan, L., Tarniewicz, J., Ramonet, M., Laurent, O., and Abbaris, A.: Automatic processing of atmospheric $CO_2$ and $CH_4$ mole fractions at
the ICOS Atmosphere Thematic Centre, Atmospheric Measurement Techniques, 9, 4719–4736, https://doi.org/10.5194/amt-9-4719-2016, 2016.

Heiskanen, J., Brümmer, C., Buchmann, N., Calfapietra, C., Chen, H., Gielen, B., Gkritzalis, T., Hammer, S., Hartman, S., Herbst, M., Janssens, I. A., Jordan, A., Juurola, E., Karstens, U., Kasurinen, V., Kruijt, B., Lankreijer, H., Levin, I., Linderson, M.-L., Loustau, D., Merbold, L., Myhre, C. L., Papale, D., Pavelka, M., Pilegaard, K., Ramonet, M., Rebmann, C., Rinne, J., Rivier, L., Saltikoff, E.,
Sanders, R., Steinbacher, M., Steinhoff, T., Watson, A., Vermeulen, A. T., Vesala, T., Vítková, G., and Kutsch, W.: The Integrated Carbon Observation System in Europe, Bulletin of the American Meteorological Society, 103, E855–E872, https://doi.org/10.1175/BAMS-D-19-0364.1, 2022.

ICOS RI: ICOS Atmosphere Station Specifications V2.0 (editor: O. Laurent), p. 2734778, https://doi.org/10.18160/GK28-2188, 2020.

Lan, X., Tans, P., and Thoning, K.: Trends in globally-averaged CO2 determined from NOAA Global Monitoring Laboratory measurements.
Version 2024-04, https://doi.org/10.15138/9N0H-ZH07, 2024a.

Lan, X., Thoning, K., and Dlugokencky, E.: Trends in globally-averaged CH4, N2O, and SF6 determined from NOAA Global Monitoring Laboratory measurements. Version 2024-04, https://doi.org/10.15138/P8XG-AA10, 2024b.

Lauerwald, R., Bastos, A., McGrath, M. J., Petrescu, A. M. R., Ritter, F., Andrew, R. M., Berchet, A., Broquet, G., Brunner, D., Chevallier, F., Cescatti, A., Filipek, S., Fortems-Cheiney, A., Forzieri, G., Friedlingstein, P., Fuchs, R., Gerbig, C., Houweling, S., Ke, P., Lerink, B.
665     J. W., Li, W., Li, W., Li, X., Luijkx, I., Monteil, G., Munassar, S., Nabuurs, G., Patra, P. K., Peylin, P., Pongratz, J., Regnier, P., Saunois, M., Schelhaas, M., Scholze, M., Sitch, S., Thompson, R. L., Tian, H., Tsuruta, A., Wilson, C., Wigneron, J., Winkler, K., Yao, Y., Zaehle, S., and Ciais, P.: Carbon and Greenhouse Gas Budgets of Europe: Trends, Interannual and Spatial Variability, and Their Drivers, Global Biogeochemical Cycles, 38, e2024GB008 141, https://doi.org/10.1029/2024GB008141, 2024.

Levin, I., Karstens, U., Eritt, M., Maier, F., Arnold, S., Rzesanke, D., Hammer, S., Ramonet, M., Vítková, G., Conil, S., Heliasz, M., Kubistin,
D., and Lindauer, M.: A dedicated flask sampling strategy developed for Integrated Carbon Observation System (ICOS) stations based on $CO_2$ and CO measurements and Stochastic Time-Inverted Lagrangian Transport (STILT) footprint modelling, Atmospheric Chemistry and Physics, 20, 11 161–11 180, https://doi.org/10.5194/acp-20-11161-2020, 2020.

McGrath, M. J., Petrescu, A. M. R., Peylin, P., Andrew, R. M., Matthews, B., Dentener, F., Balkovič, J., Bastrikov, V., Becker, M., Broquet, G., Ciais, P., Fortems-Cheiney, A., Ganzenmüller, R., Grassi, G., Harris, I., Jones, M., Knauer, J., Kuhnert, M., Monteil, G., Munassar, S., Palmer, P. I., Peters, G. P., Qiu, C., Schelhaas, M.-J., Tarasova, O., Vizzarri, M., Winkler, K., Balsamo, G., Berchet, A., Briggs, P., Brockmann, P., Chevallier, F., Conchedda, G., Crippa, M., Dellaert, S. N. C., Denier van der Gon, H. A. C., Filipek, S., Friedlingstein, P., Fuchs, R., Gauss, M., Gerbig, C., Guizzardi, D., Günther, D., Houghton, R. A., Janssens-Maenhout, G., Lauerwald, R., Lerink, B., Luijkx, I. T., Moulas, G., Muntean, M., Nabuurs, G.-J., Paquirissamy, A., Perugini, L., Peters, W., Pilli, R., Pongratz, J., Regnier, P., Scholze, M., Serengil, Y., Smith, P., Solazzo, E., Thompson, R. L., Tubiello, F. N., Vesala, T., and Walther, S.: The consolidated European synthesis of $CO_2$ emissions and removals for the European Union and United Kingdom: 1990–2020, Earth System Science Data, 15, 4295–4370, https://doi.org/10.5194/essd-15-4295-2023, 2023.

Nisbet, E. G., Dlugokencky, E. J., Manning, M. R., Lowry, D., Fisher, R. E., France, J. L., Michel, S. E., Miller, J. B., White, J. W. C., Vaughn, B., Bousquet, P., Pyle, J. A., Warwick, N. J., Cain, M., Brownlow, R., Zazzeri, G., Lanoisellé, M., Manning, A. C., Gloor, E., Worthy, D. E. J., Brunke, E.-G., Labuschagne, C., Wolff, E. W., and Ganesan, A. L.: Rising atmospheric methane: 2007-2014 growth and isotopic shift: RISING METHANE 2007-2014, Global Biogeochemical Cycles, 30, 1356–1370, https://doi.org/10.1002/2016GB005406, 2016.

Nisbet, E. G., Manning, M. R., Dlugokencky, E. J., Fisher, R. E., Lowry, D., Michel, S. E., Myhre, C. L., Platt, S. M., Allen, G., Bousquet, P., Brownlow, R., Cain, M., France, J. L., Hermansen, O., Hossaini, R., Jones, A. E., Levin, I., Manning, A. C., Myhre, G., Pyle, J. A., Vaughn, B. H., Warwick, N. J., and White, J. W. C.: Very Strong Atmospheric Methane Growth in the 4 Years 2014–2017: Implications for the Paris Agreement, Global Biogeochemical Cycles, 33, 318–342, https://doi.org/10.1029/2018GB006009, 2019.

Peng, S., Lin, X., Thompson, R. L., Xi, Y., Liu, G., Hauglustaine, D., Lan, X., Poulter, B., Ramonet, M., Saunois, M., Yin, Y., Zhang, Z., Zheng, B., and Ciais, P.: Wetland emission and atmospheric sink changes explain methane growth in 2020, Nature, 612, 477–482, https://doi.org/10.1038/s41586-022-05447-w, 2022.

Petrescu, A. M. R., Qiu, C., McGrath, M. J., Peylin, P., Peters, G. P., Ciais, P., Thompson, R. L., Tsuruta, A., Brunner, D., Kuhnert, M., Matthews, B., Palmer, P. I., Tarasova, O., Regnier, P., Lauerwald, R., Bastviken, D., Höglund-Isaksson, L., Winiwarter, W., Etiope, G., Aalto, T., Balsamo, G., Bastrikov, V., Berchet, A., Brockmann, P., Ciotoli, G., Conchedda, G., Crippa, M., Dentener, F., Groot Zwaaftink, C. D., Guizzardi, D., Günther, D., Haussaire, J.-M., Houweling, S., Janssens-Maenhout, G., Kouyate, M., Leip, A., Leppänen, A., Lugato, E., Maisonnier, M., Manning, A. J., Markkanen, T., McNorton, J., Muntean, M., Oreggioni, G. D., Patra, P. K., Perugini, L., Pison, I., Raivonen, M. T., Saunois, M., Segers, A. J., Smith, P., Solazzo, E., Tian, H., Tubiello, F. N., Vesala, T., van der Werf, G. R., Wilson, C., and Zaehle, S.: The consolidated European synthesis of $CH_4$ and $N_2O$ emissions for the European Union and United Kingdom: 1990–2019, Earth System Science Data, 15, 1197–1268, https://doi.org/10.5194/essd-15-1197-2023, 2023.

Qu, Z., Jacob, D. J., Zhang, Y., Shen, L., Varon, D. J., Lu, X., Scarpelli, T., Bloom, A., Worden, J., and Parker, R. J.: Attribution of the 2020 surge in atmospheric methane by inverse analysis of GOSAT observations, Environmental Research Letters, 17, 094 003, https://doi.org/10.1088/1748-9326/ac8754, 2022.

Rella, C. W., Chen, H., Andrews, A. E., Filges, A., Gerbig, C., Hatakka, J., Karion, A., Miles, N. L., Richardson, S. J., Steinbacher, M., Sweeney, C., Wastine, B., and Zellweger, C.: High accuracy measurements of dry mole fractions of carbon dioxide and methane in humid air, Atmospheric Measurement Techniques, 6, 837–860, https://doi.org/10.5194/amt-6-837-2013, 2013.

Resovsky, A., Ramonet, M., Rivier, L., Tarniewicz, J., Ciais, P., Steinbacher, M., Mammarella, I., Mölder, M., Heliasz, M., Kubistin, D., Lindauer, M., Müller-Williams, J., Conil, S., and Engelen, R.: An algorithm to detect non-background signals in greenhouse gas time series

710   from European tall tower and mountain stations, Atmospheric Measurement Techniques, 14, 6119–6135, https://doi.org/10.5194/amt-14-6119-2021, 2021.

Saunois, M., Martinez, A., Poulter, B., Zhang, Z., Raymond, P., Regnier, P., Canadell, J. G., Jackson, R. B., Patra, P. K., Bousquet, P., Ciais, P., Dlugokencky, E. J., Lan, X., Allen, G. H., Bastviken, D., Beerling, D. J., Belikov, D. A., Blake, D. R., Castaldi, S., Crippa, M., Deemer, B. R., Dennison, F., Etiope, G., Gedney, N., Höglund-Isaksson, L., Holgerson, M. A., Hopcroft, P. O., Hugelius, G., Ito, A., Jain, A. K.,

715   Janardanan, R., Johnson, M. S., Kleinen, T., Krummel, P., Lauerwald, R., Li, T., Liu, X., McDonald, K. C., Melton, J. R., Mühle, J., Müller, J., Murguia-Flores, F., Niwa, Y., Noce, S., Pan, S., Parker, R. J., Peng, C., Ramonet, M., Riley, W. J., Rocher-Ros, G., Rosentreter, J. A., Sasakawa, M., Segers, A., Smith, S. J., Stanley, E. H., Thanwerdas, J., Tian, H., Tsuruta, A., Tubiello, F. N., Weber, T. S., van der Werf, G., Worthy, D. E., Xi, Y., Yoshida, Y., Zhang, W., Zheng, B., Zhu, Q., Zhu, Q., and Zhuang, Q.: Global Methane Budget 2000–2020, Earth System Science Data Discussions, pp. 1–147, https://doi.org/10.5194/essd-2024-115, 2024.

Stevenson, D. S., Derwent, R. G., Wild, O., and Collins, W. J.: COVID-19 lockdown emission reductions have the potential to explain over half of the coincident increase in global atmospheric methane, Atmospheric Chemistry and Physics, 22, 14 243–14 252, https://doi.org/10.5194/acp-22-14243-2022, 2022.

Tenkanen, M. K., Tsuruta, A., Denier Van Der Gon, H., Höglund-Isaksson, L., Leppänen, A., Markkanen, T., Petrescu, A. M. R., Raivonen, M., and Aalto, T.: Partitioning anthropogenic and natural methane emissions in Finland during 2000–2021 by combining bottom-up and

725   top-down estimates, EGUsphere, 2024, 1–38, https://doi.org/10.5194/egusphere-2024-1953, 2024.

Thonat, T., Saunois, M., Bousquet, P., Pison, I., Tan, Z., Zhuang, Q., Crill, P. M., Thornton, B. F., Bastviken, D., Dlugokencky, E. J., Zimov, N., Laurila, T., Hatakka, J., Hermansen, O., and Worthy, D. E. J.: Detectability of Arctic methane sources at six sites performing continuous atmospheric measurements, Atmospheric Chemistry and Physics, 17, 8371–8394, https://doi.org/10.5194/acp-17-8371-2017, 2017.

Thoning, K. W., Tans, P. P., and Komhyr, W. D.: Atmospheric carbon dioxide at Mauna Loa Observatory: 2. Analysis of the NOAA GMCC

730   data, 1974-1985, Journal of Geophysical Research: Atmospheres, 94, 8549–8565, https://doi.org/10.1029/JD094iD06p08549, 1989.

Tsuruta, A., Aalto, T., Backman, L., Krol, M. C., Peters, W., Lienert, S., Joos, F., Miller, P. A., Zhang, W., Laurila, T., Hatakka, J., Leskinen, A., Lehtinen, K. E. J., Peltola, O., Vesala, T., Levula, J., Dlugokencky, E., Heimann, M., Kozlova, E., Aurela, M., Lohila, A., Kauhaniemi, M., and Gomez-Pelaez, A. J.: Methane budget estimates in Finland from the CarbonTracker Europe-CH$_4$ data assimilation system, Tellus B: Chemical and Physical Meteorology, 71, 1565 030, https://doi.org/10.1080/16000889.2018.1565030, 2019.

Vardag, S. N., Hammer, S., O'Doherty, S., Spain, T. G., Wastine, B., Jordan, A., and Levin, I.: Comparisons of continuous atmospheric CH$_4$, CO$_2$ and N$_2$O measurements – results from a travelling instrument campaign at Mace Head, Atmospheric Chemistry and Physics, 14, 8403–8418, https://doi.org/10.5194/acp-14-8403-2014, 2014.

Ward, R. H., Sweeney, C., Miller, J. B., Goeckede, M., Laurila, T., Hatakka, J., Ivakov, V., Sasakawa, M., Machida, T., Morimoto, S., Goto, D., and Ganesan, A. L.: Increasing Methane Emissions and Widespread Cold-Season Release From High-Arctic

740   Regions Detected Through Atmospheric Measurements, Journal of Geophysical Research: Atmospheres, 129, e2024JD040 766, https://doi.org/10.1029/2024JD040766, 2024.

WMO: 16th WMO/IAEA Meeting on Carbon Dioxide, Other Greenhouse Gases, and Related Measurement Techniques (GGMT-2011), Tech. Rep. 206, WMO, Wellington, https://library.wmo.int/idurl/4/51479, 2013.

WMO: System and Performance Audit of Surface Ozone, Carbon Monoxide, Methane, Carbon Dioxide and Nitrous Oxide at the Global

745   GAW Station Pallas, Finland, July 2021, Tech. Rep. 283, WMO, Geneva, https://library.wmo.int/idurl/4/66280, 2022.

WMO: Twenty-First WMO/IAEA Meeting on Carbon Dioxide, Other Greenhouse Gases and Related Measurement Techniques (GGMT-2022), Tech. Rep. 292, WMO, Geneva, https://library.wmo.int/idurl/4/68925, 2024.

Yuan, K., Li, F., McNicol, G., Chen, M., Hoyt, A., Knox, S., Riley, W. J., Jackson, R., and Zhu, Q.: Boreal–Arctic wetland methane emissions modulated by warming and vegetation activity, Nature Climate Change, 14, 282–288, https://doi.org/10.1038/s41558-024-01933-3, 2024.

Yver-Kwok, C., Philippon, C., Bergamaschi, P., Biermann, T., Calzolari, F., Chen, H., Conil, S., Cristofanelli, P., Delmotte, M., Hatakka, J., Heliasz, M., Hermansen, O., Komínková, K., Kubistin, D., Kumps, N., Laurent, O., Laurila, T., Lehner, I., Levula, J., Lindauer, M., Lopez, M., Mammarella, I., Manca, G., Marklund, P., Metzger, J.-M., Mölder, M., Platt, S. M., Ramonet, M., Rivier, L., Scheeren, B., Sha, M. K., Smith, P., Steinbacher, M., Vítková, G., and Wyss, S.: Evaluation and optimization of ICOS atmosphere station data as part of the labeling process, Atmospheric Measurement Techniques, 14, 89–116, https://doi.org/10.5194/amt-14-89-2021, 2021.

Zellweger, C., Emmenegger, L., Firdaus, M., Hatakka, J., Heimann, M., Kozlova, E., Spain, T. G., Steinbacher, M., Van Der Schoot, M. V., and Buchmann, B.: Assessment of recent advances in measurement techniques for atmospheric carbon dioxide and methane observations, Atmospheric Measurement Techniques, 9, 4737–4757, https://doi.org/10.5194/amt-9-4737-2016, 2016.

Zellweger, C., Steinbrecher, R., Laurent, O., Lee, H., Kim, S., Emmenegger, L., Steinbacher, M., and Buchmann, B.: Recent advances in measurement techniques for atmospheric carbon monoxide and nitrous oxide observations, Atmospheric Measurement Techniques, 12, 760   5863–5878, https://doi.org/10.5194/amt-12-5863-2019, 2019.

Zhou, L., Kitzis, D., Tans, P., Masarie, K., and Chao, D.: WMO Round-Robin Inter-comparison: Progress and a New Website, in: 15th WMO/IAEA Meeting of Experts on Carbon Dioxide, Other Greenhouse Gases and Related Tracers Measurement Techniques, no. 194 in GAW Report, pp. 161–164, WMO, Jena, https://library.wmo.int/idurl/4/58718, 2009.