# Peer review of "Long-term observations of atmospheric CO2 and CH4 trends and comparison of two measurement systems at Pallas-Sammaltunturi station in Northern Finland"

_EGUsphere, 2024_

## Author Comment (AC1)

We thank referee #1 for the useful comments to improve the manuscript. We have gone through each comment carefully and answer them below.

*This paper by Laitinen et al. presents two important datasets of long-term atmospheric CO2 and CH4 observations in northern Finland. Although they follow the same general guidelines defined by the WMO for in-situ GHG observations, including the adoption of the same calibration scale, these datasets were produced at the Pallas site using two completely independent observing systems (i.e. different sampling systems, different calibration standards, different data flagging procedures). The manuscript mainly focuses on the comparison of the two datasets (one produced within the GAW/WMO framework and the other within the ICOS-RI) to show the high consistency between them and to briefly discuss the reasons for the small observed deviations.*

*The longer CO2 and CH4 dataset (i.e. the GAW/WMO one) was also compared with the widely used NOAA MBL data product to put it into a global context.*

*The topic of comparison and attribution of deviations in contiguous measurement sites is an important and, I would say, emerging issue (see e.g. https://doi.org/10.1088/1748-9326/abe74a, https://doi.org/10.5194/amt-11-1599-2018, https://doi.org/10.5194/amt-16-2399-2023), so I appreciate the appearance of this paper.*

*Having said that. I recommend publication and my comments can be seen as mostly minor or technical suggestions/recommendations.*

***Specific comments***

*Throughout the paper: I think the nomenclature "mixing ratio" should be changed to "mole fraction" as recommended by WMO.*

Changed the use of "mixing ratio" to "mole fraction"

*An interesting point is that the GAW and ICOS data sets are produced using two different sampling inlets. I suggest that the author also add a figure or map to Figure 1 showing the different locations of the two inlets.*

*Added a schematic figure A1 to Appendix to show the spatial distances of the two inlets.*

*Line 86. "Due to its remote…representative for unpolluted air". Is there a reference to quote?*

Citation **(Hatakka et al, 2003)** added

*Line 88. For the reader not familiar with ICOS, can you briefly explain the difference between the different classes of ICOS stations (Class 1 vs. Class 2)?*

Added further explanation in the Introduction chapter:

**Within the ICOS ATC, the stations are classified into Class 1 and Class 2 stations. The requirements for the Class 2 stations are continuous measurements of $CO_2$ and $CH_4$ complemented by basic meteorological parameters: Air temperature, relative humidity, wind speed and direction as well as atmospheric pressure. The Class 1 stations are required, in addition to the requirements of the Class 2, to have continuous CO and boundary layer height measurements as well as to operate the ICOS flask sampler (described more in detail in Levin et al, 2020). Furthermore, the stations are classified to three types based on their location: continental stations targeting mainly continental air-masses, coastal stations targeting mainly**

**marine air-masses and mountain stations targeting mainly free troposphere during night (ICOS RI, 2020).**

*Section 2.2: For the ICOS instrument, you can report the instrumental uncertainty as indicated by the continuous measurement repeatability and the short - long term repeatability. Are these (or similar) indications available for the GAW instrument?*

Removed Figure 4 showing the LT of ICOS instrument and moved the information to Figures 7 and 13 with target cylinders of GAW instument as well, where also the ST of the ICOS instrument is presented

*Line 120: How did you obtain the correction factors to correct for residual water vapour interference? Did you do the "droplet test"? How often? This should also be reported for the GAW instrument (line 150).*

Added explanation for the water vapor correction assesment to sections 2.2.1 and 2.2.2

**For the ICOS analyzer, the correction coefficients are determined by the ATC during the initial instrument test by first measuring a dry gas stream from a cylinder, and then humidifying the stream for 20 minutes a step with 0.25 v% steps from 0.5 v% to 2v %, 2.5v % and 3 v%. The coefficients for $CO_2$ and $CH_4$ are then determined with the following equation:**

**$C_w/C_d = 1 + aH + bH^2$    (1)**

**Where $C_w$ is the measured wet mole fraction, $C_d$ is the dry mole fraction (measured when H = 0), H is the measured water vapor concentration and a and b are the correction factors.**

And

**Similarly to the ICOS analyzer, the instrument specific water vapor correction factors are determined as well. The approach used by the FMI is similar to that of ATC; a dry gas stream is humidified using a self-build instrument, ranging from 0 to 3.5% (Aaltonen et al, 2016). The coefficient are then calculated using Eq 1.**

And

**These coefficients are evaluated during the ICOS audit, as well as approximately once per year by the station PI, and updated if deemed necessary by the ATC.**

Section 2.2.3. I agree with the comments by Tonatiuh Guillermo Nuñez Ramire that flask results should be shown, especially the flask vs. ambient air comparison. Or at least the authors should refer to them in some way.

This is a good comment, and we agree that the flask data inclusion could bring some additional insights. However, for the scope of this paper we wanted to focus on the CRDS-based systems and their comparison and feel like adding the flask data to the manuscript would widen the scope of the paper too much. Moreover, the flask data is unlikely to reveal additional information about the offsets, considering their small magnitudes. Instead, we decided to leave section discussing the flask measurements out of the paper completely.

**Removed Flask sampling section, moved "Auxiliary measurements" to 2.3**

*Line 186: "...last and first days of the year". Can you be more precise?*

Added further clarification:

... by taking the difference of the values of the **last day (31.12) and the first day (01.01) of the given year. For example, the GR of 2020 would be calculated by taking the difference of the daily trend values of 31.12.2020 and 01.01.2020.**

*Figure 6/12: Since there is an apparent impact of the implementation of Nafion on the ICOS instrument, have you tried to analyse DCO2 (and DCH4) as a function of ambient water vapour values to better quantify possible impacts? This would also help to better attribute the differences for CH4 highlighted in lines 336 - 368 .*

Added to section 4.1:

**There exists a negative correlation between $DCO_2$ and the water vapor concentration of about -0.06 ppm/v% (intercept 0.05 v%, $p < 0.05$) when both of the instruments measure wet air.**

And to section 4.2:

**As for $CO_2$, there exists a negative correlation between $DCH_4$ and the water vapor concentration of about -0.94 ppb/v% (intercept 1.34 v%, $p < 0.05$) when both of the instruments measure wet air.**

*Lines 290 - 292: A "large" spread is also visible for the comparison of ICOS and GAW instruments. As stated by the authors the observed differences are mostly within the WMO compatibility goal but I'm wondering what happen if you subset the data as done for the trend analyses (i.e. only keeping data with standard deviation less than 0.5 ppm). This would in some way minimize the impact of the different inlet locations by catching conditions characterized by low CO2 variability.*

For the long-term comparison between the ICOS and GAW instruments, the same wind speed and STD based filtering has been applied as for the trend analysis. This is explained in section 3.3.

*Lines 298-305 and 362 - 366: These paragraphs have been added to place the results of the intercomparison between ICOS and the GAW instrument at Pallas in a broader context. It would be interesting to add information from the other intercomparisons carried out in other WCC-EMPA audits.*

Added information on other audits to the end of chapters 4.1.3

**In (Zellweger et al., 2016) an overview of results of GAW audits performed in Danum Valley (DMV), Cape Verde (CPVO), Mace Head (MHD) as well as an earlier audit at Pallas, are presented. The audits were performed in 2012 (PAL & CBO) and 2013 (DMV & MHD). The comparison for CO2 was made in PAL, DMV and CVO. The measurement methods were non-dispersive infrared (NDIR) in PAL and DMV, and off-axis integrated cavity output spectroscopy (OA-ICOS) in CVO, compared against the traveling CRDS instrument. Results of the audits show a median deviation (1h aggregation, ± standard deviation) of 0.08 ± 0.03 ppm at PAL, 0.03 ± 0.21 ppm at DMV, 0.06 ppm ± 0.06 at CVO. For Pallas, a comparison of the local CRDS against the traveling CRDS is presented in (Rella et al., 2013). For CO2, the mean deviation was -0.025 ± 0.034 ppm.**

And 4.2.3

**Earlier GAW audits in CVO and MHD for CH4 are presented in (Zellweger et al., 2016). In CVO the same instrument (OA-ICOS) was used as for CO2. In MHD the measurement method was GC. The results show a mean deviation (± standard deviation) of -0.61 ± 0.32 ppb at CVO and 0.22 ± 3.59 ppb and MHD. Comparison of the traveling CRDS against a local CRDS at Pallas in 2012 is presented in (Rella et al., 2013), with a mean deviation of -0.032 ± 0.367 ppb**

*Line 336-338: How can you say "better" performance? Can it not be "different performance"? Or do you have some quantitative test to show that the water vapour correction on the GAW analyser is more accurate?*

We mainly base this claim on the fact that the agreement between the two instruments is improved when the sample air of the ICOS analyzer is dried. Since no change is made for the GAW analyzer, this indicates that the poorer agreement prior is caused by the water correction of the ICOS analyzer, while the GAW analyzer performs well even against the dried ICOS analyzer.

*Line 388 – 391: "The largest differences were observed between the GAW and WCC systems in both CO2 and CH4, however the largest spread (CI range) in the differences was between ICOS and GAW for CO2 and between GAW and the ICOS Mobile Lab for CH4. This is expected, as the GAW system is the only one measuring from its own inlet and all the other systems are connected to the same inlet". A comparable spread for CO2 (Table 2: 0.17 ppm) was also observed for WCC and GAW vs. ICOS Mobile Lab.*

We have added a remark of this:

The largest differences were observed between the GAW and WCC systems in both $CO_2$ and $CH_4$, however the largest spread (CI range) in the differences was between ICOS and GAW **and between WCC and ICOS Mobile Lab** for $CO_2$ and between GAW and the ICOS Mobile Lab for $CH_4$. This is **partly expected**, as the GAW system is the only one measuring from its own inlet

*Line 394: "This is likely due to the filtering of the data based on the wind speed, assuring well mixed air". I think you should apply this filtering to the audit as well and see if the deviations decrease (see also my previous comment about lines 290 – 292).*

Tested the filtering of the audit data for windspeed; we generally notice a decrease in the spreads of the deviations. Added discussion to sections 4.1

**When the data are filtered for wind speed and hourly standard deviation, as done for the analysis for trend and long-term differences, the spreads between GAW and ICOS Mobile Laboratory decreases to 0.14 ppm, between GAW and WCC to 0.09 ppm and between ICOS and GAW to 0.12 ppm, indicating that the different inlet location affects the comparison when the air is not well mixed. The difference between WCC and ICOS Mobile Laboratory decreases as well to 0.15 ppm, but decrease is observed for comparison of ICOS and WCC.**

4.2

**Filtering the audit data for $CH_4$ for wind speed and hourly standard deviation leads to similar results as for $CO_2$, decreasing the spread between GAW and ICOS Mobile Laboratory to 1.40 ppb, between ICOS and WCC to 0.88 ppb and between ICOS and GAW to 1.29 ppb. Between WCC and ICOS Mobile Laboratory the spread is only reduced to 1.31 ppb and between GAW**

**and WCC to 0.98 ppb, and no significant difference in the spread is noticed between ICOS and ICOS Mobile Laboratory.**

and 5.

**When filtering the audit data for the wind speed and hourly standard deviation as well, the spreads between the GAW instrument and the rest generally decrease.**

*Technical comments*

*Data citation: the authors only provide data availability at the end of the manuscript. I recommend that the analysed dataset can be fully cited in the main manuscript, since doi are available for both GAW and ICOS datasets.*

Added citation for GAW and ICOS datasets to the main manuscript

*Line 255, but also in the rest of the manuscript: I would suggest that CI can be expressed as e.g. [-8.0, 11.8] ppm.*

Changed to [-8.0, 11.8] ppm, and similarly in the rest of the manuscript

*Line 262: "(7 bottom) -> (Fig. 7 bottom). The same in line 264. In addition, I suggest that Figure 7 can be moved to Figure 6, as it is referenced earlier in the manuscript.*

As suggested by other reviewer, changed the sentence and referenced to (Fig 7 (a)), and changed accordingly.

Swapped figures 6 and 7.

*Line 271: "before"*

changed to "before" without capitalization

*Line 291: I would suggest to be consistent and continue to use "ICOS Mobile Laboratory" or "ICOS Mobile Lab".*

Changed "Icos Mobile Lab" to "ICOS Mobile Laboratory"

*I suggest moving Figure 9 and Figure 14 to Supplementary Material.*

Moved to supplementary

*Sections 4.1 and 4.2: I would split these two sections into two/tree sub-sections as the following topics are considered: 1) Comparison of Pallas measurements with NOAA PBL dataset, 2) Long-term comparison of GAW and ICOS measurements at Pallas, 3) Combined GAW and ICOS audit.*

Sections 4.1 and 4.2 split into three sub-sections

*Figures 7 and 11: Please describe in the caption what the error bars represent.*

Added description: […] **points are hourly (a) and monthly (b) means with associated standard deviations**

*Line 373: "global development" -> "global tendencies"*

Changed as suggested

**Added figures:**

[Figure]

*Figure 7. CO2 target measurements of ICOS LT, ICOS ST and GAW target cylinders over the whole comparison period. Different cylinders*
*used are marked with distinct colors. The data are given as means of each sequence with the associated standard deviation.*

[Figure]

*Figure 13. CH4 target measurements of ICOS LT, ICOS ST and GAW target cylinders over the whole comparison period. Different cylinders
used are marked with distinct colors. The data are given as means of each sequence with the associated standard deviation*

[Figure]

*Figure A1. Schematics of the measurement setup during the audit at Pallas. ML refers to the ICOS Mobile Laboratory instrument*

---

## Author Comment (AC2)

We thank referee #2 for the useful comments to improve the manuscript. We have gone through each comment carefully and answer them below.

The referee comments are marked in *blue cursive*, the author response in black and text added to the manuscript is **bolded**

*The article "Long-term observations of atmospheric $CO_2$ and $CH_4$ trends and comparison of twomeasurement systems at Pallas-Sammaltunturi station in Northern Finland" by Laitinen et al. presents the atmospheric observations of the greenhouse gases $CO_2$ and $CH_4$ from two measurement programs, GAW and ICOS, conducted at a boreal station in northern Finland. The authors intercompared the measurements and also performed a concurrent audit of WCC-EMPA and the ICOS Mobile Lab. Long-term trends were compared and discussed, including a comparison with the Northern Hemispheric mean derived by NOAA. The results show that the comparability and compatibility of the network, including their audit, are mostly fulfilled. Deviations from the Northern Hemispheric trends and growth rates were also addressed.*

*Overall, the paper is well-structured and makes an important contribution to the comparability and compatibility of the European ICOS and global GAW networks. This is a valuable contribution to the user community relying on greenhouse gas observation data. However, the readability of the paper could be improved, and further explanations could provide more clarity.*

*I recommend publication after the following comments are addressed:*

*Page 1, Line 17: In the sentence "We also compared the long time series with the marine boundary layer reference values in the Northern Hemisphere", it is unclear to which station or network this refers. I recommend providing additional context or specifying the station or network being referenced.*

Changed the sentence to refer to the NOAA data product:

"...marine boundary layer reference values, **derived by NOAA based on the weekly air sample measurements**, in the Northern Hemisphere."

Furthermore, we have given a more detailed explanation with reference to the data set used as the reference.

*Page 2, Line 22-25: Referring to "Especially in situ measurements of greenhouse gas mixing ratios are needed for quantifying the long-term trends of the greenhouse gases, as well as annual and interannual variations. They are also crucial for top-down emission estimates using atmospheric inverse models, which aim to optimize fluxes based on measured mixing ratios (Peiro et al. (2022), Crowell et al. (2019))", it would be helpful to elaborate on why in situ measurements in particular are needed for quantifying long-term trends, annual, and interannual variations in greenhouse gases. Remote sensing technologies have advanced and it is unclear from the text why they cannot also provide sufficient data for these applications. A more detailed explanation of the advantages or necessity of in situ measurements compared to remote sensing would strengthen this argument.*

Added further explanation:

**While remote sensing techniques can also be used for this purpose, only in situ measurements can be directly calibrated to the WMO scales for $CO_2$ and $CH_4$, and can be used to link the remote sensing observations to accepted scales (Byrnet et al, 2023)**

*Page 2, Line 39: "Within the GAW network, the station is referred to as Pallas-Sammaltunturi (station id: PAL) and it reports data on $CO_2$ and $CH_4$. Meanwhile, under the ICOS network, the station is named Pallas (station id: PAL), and it provides data not only on $CO_2$ and $CH_4$ but also on CO and $N_2O$. This data is also available as GAW data, as ICOS is a contributing network to GAW": It would be helpful to clarify the term "contributing network" and the relationship between the GAW and ICOS networks. This would make the expected comparability of data from both networks clearer.*

Added a further explanation:

"**Contributing networks have signed a Letter of Agreement with WMO, detailing the list and characteristics of the stations to be included in the GAW network as contributing stations. The data from these stations is subsequently available through the GAW data portal**."

*Page 3, Line 54: The statement, "To ensure that the station's measurements are compatible with the WMO/GAW goals, they must be compared against other instruments to ensure the differences are within acceptable limits" implies that direct comparisons with other instruments are necessary. However, round robin comparisons might already be sufficient for this purpose. Could you clarify why direct comparison with other instruments is deemed essential, or whether round robin exercises or other methods could achieve the same objective?*

Added further explanation:

**While travelling cylinders can be used to ensure that the measurement scale is transferred correctly, they do not account for potential biases arising from the sampling system (WMO, 2022). With a co-located measurements with a travelling instrument, the whole sampling system can be evaluated.**

*Page 3, line 84: "The mean wind speed is 6.9 m/s." Please specify the averaging time period over which this mean wind speed was calculated to provide the corresponding standard deviation as an indication of the variability.*

Changed to:

**The mean wind speed (1996 - 2022) is 6.9 m/s (± 0.5 m/s).**

*Page 5, line 100: The sentence mentions that both ICOS instruments have been tested at the ICOS Atmosphere Thematic Centre (ATC) before being set up at the station. What about the GAW instruments? Could you clarify their validation procedure?*

The GAW instruments are usually checked for normal operation and an initial calibration was performed before deployment at a station. However, more important is the verification procedure during the field measurements, which includes regular measurements of calibration and target standards.

*Page 5, line 105: The phrase "the maximum bias tolerable when measuring well-mixed background air" is used in Table 1, but it is unclear what the bias refers to. Could you please specify what is meant by "bias" in this context?*

Added explanation:

(the maximum bias **between different datasets** tolerable when measuring well-mixed background air)

*Page 6, Table 1: Table 1 shows "the WMO Compatibility goals for $CO_2$ and $CH_4$ measurements." Could you please clarify the specific requirements for the ICOS network in relation to these goals?*

Added further explanation in Section 1:

… measuring well-mixed background air) and the measurement ranges are presented(WMO, 2024). **The ICOS network aims for the same goals, however covering a wider range (ICOS RI, 2020)**

*Page 6, Line 120: The sentence states, "Before that time, the air was measured as wet, and corrections to convert to mole fractions in dry air were applied." It is not clear whether the correction refers only to the dilution factor or if corrections for cross-sensitivities have already been applied. Additionally, it is unclear what the remaining water concentration is after passing through the Nafion dryer and whether any further corrections are applied afterward. Please provide additional information on these points.*

Quantified the water content of the sample air in section 2.2:

… in dry air were applied. **Without the dryer, the sample water content was, on average, 0.59 v % (+/- 0.33 v%). With the dryer installed, the remaining water content was on average 0.06 v % (+/-  0.01 v%).**

In addition, removed statement from line 130:

**and corrections to convert to mole fractions in dry air were applied**

For clarity, as even with the dryer the remaining water vapor effect is still corrected as explained on in section 2.2.1.

*Page 6, line 125: "To monitor the quality of the measurements, long-term and short-term target cylinders are measured to identify any drift in the measurements between calibrations." Could you please provide further information on why two target cylinders (LTT and STT) were used instead of just one?*

Added explanation:

...measured directly after each calibration. **The purpose of the short-term target is to ensure quality on daily basis, while the long-term target can ensure the continuity of the quality control as the cylinder should last over a decade (Ywer-Kwok et al, 2021).**

*Page 6, line 130: "The dry mole fractions can be obtained by sufficiently drying the sample […]". Please specify what is meant by sufficiently.*

Added explanation:

… sufficiently drying the sample (**dew points of at most -50 ºC (WMO, 2013)**)

*Page 7, Figure 3: The abbreviations for the cylinders in the schematic flow diagram (STWS, C1, C2, C3, C4) are either not explained or differ from those used in the text (STT, LTT). Please clarify the terms and ensure consistency between the diagram and the text.*

Changed the cylinder abberviations to match the schematic and inlcuded the missing cylinder terms

*Page 8, line 134: The sentence states "[…] used at ICOS, the correction coefficients are determined for each instrument individually […]." Could you clarify why this approach is used instead of applying the standard instrument water correction?*

Added clarification:

**However, as the pressure broadening effect caused by the water vapor in the sample is different for each instrument, the ICOS strategy is to determine the correction coefficient for each instrument individually and apply the correction in the ICOS database (Hazan et al, 2016).**

*Page 9, line 153: The sentence states, "The calibration standard cylinders used for the GAW analyzers are filled by NOAA, and the STT and LTT cylinders are filled by the FMI." Could you clarify how the FMI concentration is connected to NOAA, and how the assigned concentrations for the STT and LTT cylinders are obtained? Additionally, could you provide more information about the quality assurance process for GAW, similar to your description of ICOS? For instance, NOAA serves as the central calibration laboratory (CCL) for $CO_2$ and $CH_4$ in the GAW framework.*

*Added clarification for the GAW QA process:*

**The GAW QA process includes regular system and performance audits carried out by WCC-EMPA for CO2 and CH4 (Zellweger et al, 2016)**

Here we fix a mistake in the text: All the cylinders used for the GAW instrumentation are filled by the FMI but calibrated against NOAA standards:

***All the standard cylinders used for the GAW instrument are filled by the FMI, and calibrated against a set of four standard cylinders prepared by NOAA.***

*Page 9, line 157: The paragraph "2.2.3 Flask sampling" provides further information on the ICOS sampling strategy, but it is unclear how this information contributes to the analysis in the main paper. Could you clarify its relevance in your analysis or consider to leave it out for better flow?*

As in response to reviewer 1, we decided to omit the Flask sampling section from the manuscript.

**Removed section Flask sampling, moved Auxiliary measurements to 2.3**

*Page 10, Line 202: The sentence states "In order to account for a potential drift of the travelling standards, they are regularly compared to laboratory standards between the audit campaigns". It is unclear what is meant by "laboratory standards". Are those provided by the CAL-FCL and how current are the assigned values? Could you please provide further clarification on these points?*

Clarified:

...they are regularly compared to **the NOAA standards used to calibrate the GAW instument standard cylinders** between the audit campaigns.

Furthermore, in Section 2.2.2:

… **calibrated at the FMI laboratory against a set of four standard cylinders prepared at the NOAA Global Monitoring Laboratory (GML) before being send to the station. These cylinders are regularly calibrated at the GML and the latest calibration for the FMI standards was in July 2018.**

*Page 11, line 223: "The WCC-Empa analyzer sampled air from a location close to the ICOS Picarro and the ICOS Mobile Laboratory inlets […]". Could you please quantify what is meant by "close"?*

Added a schematic figure A1 to show the inlet locations during the audit

*Page 11, line 229: The sentence states, "The lower limit for the wind speed is 3 m/s during summertime (June-August) and 4 m/s during wintertime." Could you please clarify why the difference is made between the wind speed limits for summertime and wintertime?*

Added explanation:

... hourly measurements **based on wind statistic, as defined by (Aalto et al, 2015). Due to the differing wind speeds between summer and winter, the criteria is defined separately for the seasons.**

*Page 12, line 261:" $CO_2$ sinks at Pallas are mostly vegetation, and the effect can be seen in the seasonal cycle (7, bottom)." How is this effect distinguished from the seasonal cycle observed in background $CO_2$ concentrations, such as those in the NOAA marine boundary layer data?*

While the seasonal cycle is mainly driven by increased vegetation activity during summer, we agree that distinguishing this effect from the background seasonal cycle is difficult. Instead, we refer now refer to the diurnal cycle which should capture the local effect better.

 … effect can be seen in the **diurnal** cycle (**Fig.** 7 **(a)**)

*Page 13, Figure 5: In Figure 5, specific years (e.g., 2001) show larger deviations in the CO₂ growth rate at Pallas compared to the MBL growth rate. Could you elaborate on the potential causes of these deviations?*

Added discussions:

… growth rate (GR( of about 2 ppm/year at Pallas is comparable to the globally observed changes in $CO_2$ (Fig. 5). **Measurements at Pallas show, however, larger deviations in the $CO_2$ GR than the northern hemisphere averages. Partly this can be explained by noting that we compare measurements from one location to an averaged product, which naturally leads to higher variation. A negative GR is observed at Pallas in 2001; this is caused by elevated $CO_2$ mole fractions during late 2000, and lower values during late 2001. The exact reason is difficult to quantify based on atmospheric measurements alone, however the fall 2000 was warm with little precipitation, which could influence the $CO_2$ emissions. The summer 2006 and autumn 2010 were both dry with less precipitation than normally, likely influencing the emissions.**

*Page 14, line 271: I do not understand why the average hourly difference differs from the average daily difference. If I understand correctly, as described in page 11, line 229 ff, the data was filtered based on hourly averages using wind speed, the standard deviation of the hourly measurements, and a minimum of 60 minutes of data. Since the calculation of the mean is a linear operation, I would expect the averaged deviation between the two instruments to be the same over the same time*

*period, whether calculated from hourly or daily averages (excluding the confidence intervals, which will naturally differ). Is there an additional filtering process applied? Please clarify and provide further details in the corresponding paragraphs.*

Added explanation of the phenomena in Section 3.3:

**When comparing the data on hourly and daily resolution, it could be expected that the mean difference remains the same on both resolutions. However, as we filter the hourly data and later aggregate this filtered time series to daily values, days with different amount of hourly data points are represented differently in the final daily timeseries, leading to small differences in the comparison of daily and hourly values (i.e., a day with 24 hourly data points would be weighted twice as much as a day with 12 hourly data points in hourly means, while after aggregating to daily means both days would be weighted equally). The hourly data can also be unequally distributed within the days.**

*Page 14, line 282: The statement "Results of the combined ICOS and GAW audit are presented in Fig. 8" refers to the results obtained by the mobile lab. However, it is not clear whether the results are based on the measurements from both instruments (Picarro G2401 and Spectronus) individually or if they represent the average of both. Please clarify to which instruments the numbers refer.*

Added explanation:

… Fig. 8. **In the analysis the ICOS Mobile Laboratory data is measured with G2401 analyzer.**

*Page 14, line 284: "As the ICOS inlet is in a slightly different location than the GAW inlet […]". Whilst there is information on the location of each inlet in sections 2.2.1 and 2.2.2, the relative location of both inlets is not provided. Please include this information to give the reader a better understanding of their spatial relationship.*

Added figure A1 to provide information on the spatial relationship of the inlets

*Page 14, line 293: It is discussed that different flow rates might cause the larger spread in the differences observed between the co-located WCC-EMPA and ICOS Mobile Lab measurements. Have you considered correcting for the different flow rates to validate this hypothesis?*

After further thought on the flow rates, we have estimate that the impact on the comparison is negligible as the lines at the site are rather short (compared to for example tall towers) and we compare data on hourly and daily means.

Explained at the text:

***While the flow rates of the different instruments are slightly different, the effect of the flow rate is likely small as the lines at the stations are rather short and we are comparing hourly aggregated data.***

*Page 14, line 298: Please provide information on the type of reference instrument used at HEI, as well as the instruments at CBW and OPE.*

Included the types of instruments used at the stations

...fourier transform infrared (FTIR) based travelling instrument to a **GC** reference instrument at Heidelberg (HEI) as well as to local **CRDS** instruments

*Page 15, line 312: "There is no significant difference between the cold and warm seasons as in $CO_2$, indicating little influence of the local vegetation to $CH_4$". While it is stated that there is no significant difference between the cold and warm seasons for $CH_4$, background concentrations of methane are generally expected to show some seasonal variation, particularly due to changes in the OH sink. Figure 11 also shows higher mean values in the cold season compared to the warm season. However, it is unclear what the error bars represent and whether they could account for the lack of significant difference.*

We have clarified this in the answer to the comment below:

*Page 15, line 314: "A seasonal cycle is visible in $CH_4$". This seems contradictory to the previous statement on line 312. Could you clarify how the observed seasonal cycle aligns with the statement that there is no significant difference between the cold and warm seasons.*

Here we are referring to the difference between the Pallas observations and the Northern Hemisphere average, and between those two the difference does not vary from cold to warm season, unlike CO2.  We clarify this with the following change:

**Unlike for $CO_2$, there is no significant variation in the differences between the Pallas observations and the Northern Hemisphere averages between the cold and the warm season, indicating little influence of the local vegetation to the $CH_4$ mixing ratios.**

*Page 16, Fig. 7: Could you please provide information on the error bars in the figure? What do they represent?*

Added explanation to the caption:

… from the trend line) (b). **Data points are hourly (a) and monthly (b) means with associated standard deviations.**

***Page 19, line 326: "However, the increase measured at Pallas is significantly higher than on the average in the northern hemisphere, indicating a strong increase of local and regional emissions". Do you have any ideas on the potential sources of these emissions?***

Added explanation:

… regional emissions. **These emissions are most likely caused by increase in wetlands or anthropogenic sources (Tenkanen et al, 2024).**

*Page 19, line 335: "As the drying process eliminates most of the moisture from the sample, the variation is reduced". Please quantify the remaining moisture in the sample in section 2.2.1.*

Quantified the water content of the sample air in section 2.2.1:

… in dry air were applied. **Without the dryer, the sample water content was, on average, 0.59 v % (± 0.33 v%). With the dryer installed, the remaining water content was on average 0.06 v% (± 0.01 v%).**

In addition, removed statement from line 130:

**and corrections to convert to mole fractions in dry air were applied**

For clarity, as even with the dryer the remaining water vapor effect is still corrected as explained on line 148.

*Page 19, Line 336: "The discrepancy could also be caused by better performance of water vapor correction of the GAW analyzer compared to the ICOS analyzer". Is the procedure for determining the water vapour correction different for the GAW and ICOS analysers, which might explain why the GAW instrument performs better?*

Added explanation for the water vapor correction assesment to sections 2.2.1 and 2.2.2

**For the ICOS analyzer, the correction coefficients are determined by the ATC during the initial instrument test by first measuring a dry gas stream from a cylinder, and then humidifying the stream for 20 minutes a step with 0.25 v% steps from 0.5 v% to 2v %, 2.5v % and 3 v%. The coefficients for $CO_2$ and $CH_4$ are then determined with the following equation:**

**$C_w/C_d = 1 + aH + bH^2$    (1)**

**Where $C_w$ is the measured wet mole fraction, $C_d$ is the dry mole fraction (measured when H = 0), H is the measured water vapor concentration and a and b are the correction factors.**

And

**Similarly to the ICOS analyzer, the instrument specific water vapor correction factors are determined as well. The approach used by the FMI is similar to that of ATC; a dry gas stream is humidified using a self-build instrument, ranging from 0 to 3.5% (Aaltonen et al, 2016). The coefficient are then calculated using Eq 1.**

 Added further discussion:

**CO2, the GR of CH4 varies more for Pallas than for the northern hemisphere average. This can be expected, as local variations in sources and sinks affect the mole fractions at the site more than the hemisphere averages. Especially in 2006 and 2010 the GRs observed at Pallas were negative. A study by Tsuruta et al. (2019) found that during those years the CH4 emissions in Finland were lower than usually, which is a likely explanation for the lower mole fractions observerd at Pallas during those years.**

*Page 21, Line 356: "Furthermore, all the instruments are calibrated using a separate set of calibration standards, which could cause differences in the calibrated values". Have you considered performing cross-calibrations of the cylinders to confirm this hypothesis?*

Added two tables with results from cross-calibrations to the Appendix A1 and A2 as well as discussion to chapter 3

**To account for possible differences in the calibration standards, the ICOS Mobile Laboratory cylinders were measured with the ICOS analyzer, and the WCC travelling cylinders were measured with the GAW analyzer. The results of the measurements are presented in table A1 and table A2. The calibration cylinder measurements show that for $CO_2$ the ICOS analyzers**

**measurements differ on average from -0.09 ppm to 0.01 ppm to the assigned cylinder values, and the GAW analyzer differs from -0.05 ppm to 0.03 ppm.**

and 4

...all the instruments are calibrated using a separate set of calibration standards, **leading to slight differences in the calibrated values. To quantify this effect, during the audit the ICOS analyzer measured the ICOS Mobile Laboratory calibration standards, and the GAW analyzer measured the WCC travelling standards. The results of the calibrations are presented in tables A1 and A2. For CH₄, the ICOS analyzer measured 0.12 to 0.54 ppb lower values than the assigned values of the cylinder, and the GAW analyzer measured 0.16 to 0.72 ppb higher values.**

*Page 25, line 391: "[…] and all the other systems are connected to the same inlet." I understand that the inlet lines are closely located but distinct (e.g., section 3.2). Could you please clarify this point?*

Clarified this point in section 3.2 as well as added a figure A1:

The WCC-Empa analyzer sampled air from **the same inlet** as the ICOS and the ICOS Mobile Laboratory instruments

*Minor changes:*

*Page 1, Line 13:  For clarity, I recommend changing "World Calibration Centre (WCC)" to "World Calibration Centre (WCC-EMPA)" as there are multiple World Calibration Centres (WCCs) within the GAW program.*

Changed according to the comment

*Page 2, Line 21:  Please change "Accurate, long-term observations of the atmospheric greenhouse gas […] to "Accurate, long-term observations of atmospheric greenhouse gases […]" to reflect the plural of the gases being studied.*

Changed

*Page 2, Line 22: I recommend changing "[…] in the composition of the atmosphere […]" to "[…] atmospheric composition […]" for better fluency and conciseness.*

Changed

*Page 2, Line 50: "[…] ICOS aims to capture the entire carbon cycle. This includes atmospheric mixing ratio observations of different greenhouse gases, as well as […]": For clarity, I would recommend explicitly mentioning the key components of the carbon cycle, such as CO2 and CH4, as not all ICOS-related observations (e.g., SF6) are part of the carbon cycle: "[…] atmospheric mixing ratio observations of CO2 and CH4, as well as […]"*

Changed to:

[…] of **$CO_2$ and $CH_4$,** as well as [...]

*Page 3, line 58: "One of the central facilities of the ICOS ATC is the ICOS Mobile Laboratory, is tasked with this exact purpose: auditing the different atmosphere stations by means of parallel measurements and cross-comparisons.": The term "central facilities" may cause confusion if it is not the official terminology used by ICOS. To avoid misinterpretation, I suggest rephrasing the sentence as follows: "The ICOS ATC is composed of various components, including the ICOS Mobile Laboratory, which is tasked with this exact purpose: auditing the different atmospheric stations through parallel measurements and cross-comparisons."*

Changed the sentence as suggested

*Page 3, Line 83: For clarity, I suggest to rephrase the sentence "The prevailing wind direction atop the Sammaltunturi is in the West - South axis (Fig. 2 (B)), with very little wind coming in from North" to "The prevailing wind direction atop Sammaltunturi is along the west-south axis (Fig. 2B), with very little wind coming from the north."*

Changed the sentence as suggested

*Page 5, line 90: I suggest to replace "in sense" to "in terms": "During the last 25 years, greenhouse gas instrumentation has undergone substantial improvements in terms of precision, measurement frequency, and user-friendliness."*

Changed as suggested

*Page 5, Line 93: Please remove "based" in the sentence: "Later, in January 2009, both instruments were replaced by a single cavity ring-down spectroscopy (CRDS) instrument capable of measuring both species simultaneously"*

Removed word "based" from the sentence

*Page 5, Line 95: The sentence states, "These instruments were producing data for the GAW network, which was later supplemented by a separate CRDS-based instrument producing data for the ICOS network." To improve clarity, please specify the time frame for "later".*

Changed to:

… was  **in 2017** supplemented by a separate CRDS instrument producing data for the ICOS network.

*Page 5, line 98: I suggest to use "dry mole fraction" instead of "mole fraction" to improve clarity: "These commercially available CRDS instruments are capable of measuring dry mole fractions of […]"*

Changed as suggested

*Page 6, line 113: Please remove  "  in the sentence "[…] directly after each calibration."*

Removed

*Page 6, line 125: The sentence states, "To monitor the quality of the measurements, a long-term and short-term target cylinders are measured to identify any drift in the measurements." Please add "between calibrations" at the end of the sentence, as the calibrations will correct for any drifts.*

Changed as suggested

*Page 6, line 128: For clarity I would reorder the sentences and add "with varying ambient water content" to: "The water vapor present in the sample air dilutes the mixing ratios of $CO_2$ and $CH_4$, as well as broadening the absorption peaks. In order to make the measured mixing ratios comparable between different stations with varying water content, the effect of water vapor in the sample must be removed. The resulting dry mole fraction is the comparable physical quantity to report."*

Changed as suggested

*Page 8, line 135: The sentence states, "For the GAW analyzer at Pallas, the coefficients are determined by the FMI." This information would be more appropriately placed in the GAW section for better clarity and structure.*

Moved to section 2.2.2

*Page 9, line 147: Typo. Please correct "ever" to "every".*

Corrected

*Page 11, line 252: The sentence states, "Consistent with the global trend, the $CO_2$ levels have risen [...]". For clarity, I would recommend adding "at Pallas": "Consistent with the global trend, the $CO_2$ levels at Pallas have risen [...]"*

Added as suggested

*Page 12, line 254: Does the "average in the Northern Hemisphere" refer to the average over the entire Northern Hemisphere or specifically the marine Northern Hemisphere? Please clarify.*

Added the clarification for "MBL" (marine boundary layer)

*Page 12, line 255 (and following paragraphs): The format"'95% CI: -8.0 ppm–11.8 ppm" may be misleading, as it could be interpreted differently by some readers. To clarify that the interval ranges from -8.0 ppm to 11.8 ppm, I suggest using a more explicit format, such as "95% CI: [-8.0 ppm, 11.8 ppm]" or "95% CI: (-8.0 ppm to 11.8 ppm)". This suggestion also applies to the following paragraphs where a similar format is used to describe confidence intervals.*

Changed the confidence intervals to format: **95% CI: [-8.0 ppm, 11.8 ppm]**

*Page 12, Line 256: "[...] with a mean difference of 4.10 ppm uses an additional leading digit. Consider simplifying to "4.1 ppm" for consistency.*

Changed as suggested

*Page 13, line 280: Typo in "[...] is 0.01 ppm ppm [...]". Please remove one "ppm".*

Removed extra "ppm"

*Page 16, Line 314: Typo: "(Fig 11" with missing ")".*

Added missing bracket

*Page 16, line 314: Typo: Change "Amplitude of the diurnal cycle" to "The amplitude [...]"*

Corrected typo to : **The amplitude**

*Page 19, line 345: Please change "When the data is filtered […]" to "When the data are filtered […]"*

Changed to: **When the data are filtered [...]**

*Corrected*

**Added figures:**

[Figure]

*Figure A1. Schematics of the measurement setup during the audit at Pallas. ML refers to the ICOS Mobile Laboratory instrument.*

**Added tables**

| | CAL 1 | CAL 2 | CAL 3 |
|---|---|---|---|
| $CO_2$, assigned [ppm] | 379.24 | 414.46 | 449.39 |
| $CO_2$, measured [ppm] | 379.21 | 414.37 | 449.40 |
| $\Delta CO_2$ [ppm] | -0.03 | -0.09 | 0.01 |
| | | | |
| $CH_4$, assigned [ppb] | 1985.48 | 1799.53 | 2210.77 |
| $CH_4$, measured [ppb] | 1985.36 | 1798.99 | 2210.63 |
| $\Delta CH_4$ [ppb] | -0.12 | -0.54 | -0.14 |

**Table A1.** Cross-calibration of the ICOS Mobile Laboratory calibration standards with the Pallas ICOS analyzer: The assigned values of the cylinders, average measured values with the GAW analyzer, and the difference of measured value to the assigned value.

| | CAL 1 | CAL 2 | CAL 3 | CAL 4 | CAL 5 | CAL 6 | CAL 7 |
|---|---|---|---|---|---|---|---|
| $CO_2$, assigned [ppm] | 378.12 | 387.39 | 406.99 | 411.21 | 417.53 | 412.70 | 427.81 |
| $CO_2$, measured [ppm] | 387.15 | 387.39 | 407.01 | 411.18 | 417.48 | 412.67 | 427.80 |
| $\Delta CO2$ [ppm] | 0.03 | 0.00 | 0.02 | -0.03 | -0.05 | -0.03 | -0.01 |
| | | | | | | | |
| $CH_4$, assigned [ppb] | 1883.44 | 1890.78 | 1933.20 | 1953.82 | 1963.81 | 1998.97 | 2191.22 |
| $CH_4$, measured [ppb] | 1884.16 | 1891.23 | 1933.69 | 1954.13 | 1964.16 | 1999.13 | 2191.50 |
| $\Delta CH_4$ [ppb] | 0.72 | 0.45 | 0.49 | 0.31 | 0.35 | 0.16 | 0.28 |

**Table A2.** Cross-calibration of the GAW travelling standards with the Pallas GAW analyzer: The assigned values of the cylinders, average measured values with the GAW analyzer, and the difference of measured value to the assigned value.

---

## Author Comment (AC3)

We thank referee #3 for the useful comments and the encouragement for deeper exploration of the data. We have considered each point carefully and we hope that our responses add clarification to the manuscript.

The referee comments are marked in *blue cursive*, the author response in black and text added to the manuscript is **bolded**

*The manuscript Laitinen et al. presents experimental evidence that allows to assess the compatibility of two independent measurement programmes at the Pallas measurement site. The provision of longterm quality control (QC) data giving evidence of the data quality of observations is crucial to evaluate the consistency of the global observational data set. The information the authors have compiled for their study is exemplary. The two independent set-ups have each independent methodological choices concerning aspects like drying, calibration approaches or the frequency of QC measurements. The offsets that are visible in the CO2 and CH4 comparisons document a very satisfactory performance. In additition, both programmes maintain each their proper external auditing entitities. The core data set of this manuscript is the result of a coordinated auditing campaign to have four parallel measurements. This data set provides a great opportunity to learn about causes for biases or methodological weaknesses. So first of all I highly welcome the publication of such a study. On the other hand I feel that currently the manuscript does not evaluate the data set sufficiently. It has drawn just few conclusions without strong ambitions to use the findings to learn about which methodological approaches are more successful than others. The manuscript would be of much more interest if it could give guidance what limitations might be cause for measurement offsets. Therefore, I would strongly encourage the authors to further explore the data available to them as suggested below.*

*General comments*

- *The four parallel measurements done during the audit campaign have very clear differences in their calibration approaches (no. of calibration points between 2 and 9; frequency between daily and every 3 months). They all also have there own internal quality control set-up (mostly using target standard gas analysis to monitor the success of the efforts to achieve maximum accuracy). Those internal QC records should contain information on what internal reproducibility the two observational programmes (GAW and ICOS) but also the two audit programmes (by the ICOS mobile laboratory and GAW WCC) deliver. From my point of view, those target data records should also be included into the manuscript and a thorough data uncertainty assessment for the individual measurement programmes be made (for the continuous station measurements as well as for the auditing programmes). An estimate of the expected measurement compatibility between the various systems based on such an assessment is needed to judge if the observed comparison mean offsets and their spread presented in Table 2 point to additional systematic analytical limitations.*

  Thank you for the comment. This is a very valid point and we have put the effort in analyzing the target cylinders records more carefully. We have addressed the individual comments regarding this in the specific comments below. In addition, we have added a section 3.4: Quality assurance

**3.4 Quality assurance**

For quality assurance (QA) of the different measurement instruments, we analyzed their respective calibrated target cylinder measurements. Using the calibrated values allows for a comprehensive evaluation of the instrument as well as the calibration cylinders and calibration method, which also varies by instrument. As a measure of long-term repeatability (LTR), we calculate the deviation of the target cylinder measurement from the assigned value of the cylinder, and calculate the standard deviation of these values. This approach is used to account for different target cylinders used over time. For cases where only one cylinder is used, this value is equal to simply taking the standard deviation of the target cylinder measurements. For short-term repeatability (STR), we use the standard deviation of the individual cylinder measurement sequences. In addition, we calculate the mean bias of the measured values to the assigned target mole fractions for each instrument. For stabilization, only the last minutes of the injection are used for the analysis. For ICOS and GAW instruments, each measure is calculated for the whole time period of concurrent measurements as well as for the audit period. For the ICOS instrument, the LT is used for the long-term comparison QA and the ST for the audit period. For GAW instrument, one target cylinder is used for QA and a short-term working standard is used for drift correction. For the audits, one target cylinder is analyzed for each travelling instrument. In order to to get a table measurement, only the last minutes of each cylinder measurement is used for calculating the means. The different cylinders are presented in tables 3 (CO2) and 5 (CH4). The assigned values, measurement times as well as the number of minute data points used for averaging are presented, as well as STR, LTR and mean biases.

- *The focus of the manuscript could be sharpened. There is general site descriptions that have already been published (in a reference that is cited) and are not essential to the understanding of the main topic of the study.*
  While we agree that the focus of the manuscript is quite wide, we feel like having e.g. the general site descriptions still in the manuscript is helpful for people to understand the location and the site specific characteristics. Especially the cited paper describes the whole Pallas area, while we want to focus especially on the atmospheric station.

- *Throughout the document it remains a challenge to understand at which points different names for the site are used as synonyms and where those specific names cover a different meaning. Please try to harmonize and avoid terms that are used as synonyms.*

  We will harmonized the terms used to refer to the station throughout the manuscript.

*Specific comments*

- *l. 25: It is not clear why exactly those two references have been selected for the use of observational data for atmospheric inverse models and the top-down emission estimates.*
  We have added different citations that hopefully reflect better the need for measurements:
  **McGrath et al. (2023), https://doi.org/10.5194/essd-15-4295-2023)**
  **Petrescu et al. (2023), *https://doi.org/10.5194/essd-15-1197-2023*)**
  **Lauerwald et al.(2024), *https://doi.org/10.1029/2024GB008141*)**
  **Saunois et al. (2024), *https://doi.org/10.5194/essd-2024-115*) ;**
  **Friedlingstein et al. (2025), https://doi.org/10.5194/essd-14-4811-2022)**

- *l. 30: The term supersite does not have a clear definition. Either define it or omit it.*
  Included definition of supersite on line 50 and changed "supersite" on line 30 to "site"

- *l. 41: "This data is also available as GAW data" is a bit confusing. Consider rephrasing as:"This data is also submitted to the WMO World Data Centre for Greenhouse Gases..."*
  Changed as suggested

- *l. 54: The following sentence does not fully meet the meaning of WMO Expert Group's recommendations: "To ensure that the station's measurements are compatible with the WMO/GAW goals, they must be compared against other instruments to ensure the differences are within acceptable limits." I suggest to rephrase like "Assessing the compliance to the WMO network compatibility goals requires comparison of station measurements with other laboratory's measurements". Please add Andrews et al (www.atmos-meas-tech.net/7/647/2014/) as reference.*
  Changed as suggested

- *l. 111: "Pallas was labelled as an ICOS Class 1 atmosphere station": The meaning of "labelled as an ICOS class 1 station" ought to be explained and doi.org/10.5194/amt-14-89-2021 should be provided as reference.*
  Added clarification:
  **Within the ICOS ATC, the stations are classified into Class 1 and Class 2 stations (Ywer-Kwok et al, 2021).**

- *Figure 4a: The caption should indicate that LTT only is displayed. The trend in an increasing DCO2 for D348367 is substantial with respect to the mean bias numbers in Table 2. A discussion of this should be offered some part in the manuscript. As the LTT are analysed only every 15 days after the calibration the STT record would be of greater interest to compare it with the observed measurement system differences.*
  Removed Figure 4 and moved the information to Figures 7 and 13, where also the ST of the ICOS instrument is now presented.

- *l. 134-136: Are the water vapour correction coefficients assumed to be constant over time (i.e. established just once) or are those re-established according to a certain schedule?*
  The correction coefficients are monitored during the audit as well as approximately once per year by the station PI, and updated if deemed necessary by the ATC. Added explanation to the text:
  **These coefficients are evaluated during the ICOS audit, as well as approximately once per year by the station PI, and updated if deemed necessary by the ATC.**

- *l. 145: The GAW Picarro is calibrated every 2.5-3 months which assumes response stability throughout that period. When was the calibration done during the auditing campaign?*
  The GAW picarro was calibrated at the beginning and end of the audit period. Added in section 3.2:
  **The GAW instrument was calibrated at the beginning and at the end of the joint audit period.**

- *l. 154: "STT and LTT cylinders.." A similar figure as Figure 4 displaying the GAW programme's analyzer QC record should be presented.*
  Added figures 7 and 13 to show these for GAW as well, removed figure 4.

- *l. 200 ff: Same comment for the Mobile Laboratory measurements: the target measurement results should be presented similarly to Fig. 4.*
  Added figures 9 and 15 to show these measurements

- *l. 213: What are the results of the assessment the validity of the internal water vapor correction coefficient?*
  Here we noticed a mistake; The audit is validating the coefficients determined by the ATC before the instrument is being shipped to the station. Also added assesment of the coefficients.
  [...] the validity of the water vapor correction coefficients **determined by ATC. For Pallas ICOS instrument the coefficient were deemed valid and no change was required.**

- *l. 219: The WCC calibration approach is not fully clear: the description states that there is a two point calibration performed (zero plus one atmospheric reference standard). The function of the working standard to account for "drift" (analyzer response drift or $CO_2$ drift?) is unclear. Is the working standard effectively being used as a second calibration standard?*
  Changed the explanation the be more detailed:
  **The zero reading of the WCC-Empa travelling instrument has been calibrated with $CO_2$ and $CH_4$ free air (or nitrogen 6.0) prior to field use by adjusting the offsets in the user calibration file of the instrument. During the field use of the TI, only one working standard (WS) is used to calibrate the instrument for $CO_2$ and $CH_4$. In a first step, a Loess function is fitted to the WS (measured every 1445 min) to correct for drift. The resulting drift correction is then applied to all TI data in a second step. The drift corrected WS is then used to apply a calibration factor to the data using the assigned**

**value of the WS based on calibration against the CCL standards before and after field use. Two target standards are measured to verify the drift correction.**

- *l. 221: Same comment as above for the WCC audit measurements: the target measurement results should be presented similarly to Fig. 4.*
  Added figures 9 and 15 to show these measurements

- *Figure 5 caption: "the MBL data" and "the NOAA MBL data" refer to the same data set. Rather use only one term – I think "the NOAA mean MBL data" describes it best.*
  Changed term "MBL data" to NOAA mean MBL data"

- *l. 275-276: I cannot judge if a change from 94.1 % to 95.3 % is significant. The authors should consider adding a figure displaying the inter-instrument bias vs. the measured water content to reveal if there is any influence of the different drying and water vapour correction functions. Likewise a figure displaying the bias dependence on the mole fraction would be appreciated that might give an indication of a potential calibration offset contributing to the offset.*
  We have added discussion on the dependency of the difference on the mole fraction and water vapor concentration, as well as two figures A4 and A6 to illustrate.

  **$CO_2$:**
  **There is a negative correlation between the $DCO_2$ and the water vapor concentration of about -0.05 ppm/v% (intercept 0.05 v%, $p < 0.05$, $R^2$: 0.12) when both of the instruments measure wet air (Fig. A3, (a)). The correlation changes to slightly positive when the ICOS sample air is dried (slope 0.04 ppm/v%, intercept = -0.04, $p < 0.05$, $R^2$= 0.08). Additionally, only a very weak correlation with mole fraction was observed on the differences (Fig. A3, (b)).**

  **$CH_4$**
  **As for $CO_2$, there is a negative correlation between $DCH_4$ and the water vapor concentration of about -0.80 ppb/v% (intercept 1.35 v%, $p < 0.05$, $R^2 = 0.40$) when both of the instruments measure wet air (Fig. A5, (a)). After the ICOS analyzer sample is dried, the correlation is slightly positive of about 0.22 ppb/v% (intercept -0.02, $p < 0.05$, $R^2 = 0.06$). When measuring wet samples, there is a slight correlation with mole fraction of about 0.01 ppb/ppb (intercept -9.37, $R^2 = 0.06$), and when the ICOS analyzer sample is dried no significant mole fraction dependency is observed (Fig. A5, (b)).**

- *l. 279: For readability please do not abbreviate "growth rates" ("GRs").*
  Changed instances of GRs to growth rates and GR to growth rate

- *Figure 6 / Figure 12: Please state explicitly which of the two data is subtracted from which (ICOS-GAW or GAW-ICOS) either in the figure caption or in the y-axis label.*
  Added explanation (GAW-ICOS) to captions

- *Figure 8: In the presentation of three records in one graph the visibility of the blue data points is too limited. Three rows of graphs should be considered (e.g. grouping GAW-ICOS and WCC-MobLab / ICOS-WCC and ICOS-MobLab / GAW-WCC and GAW-Moblab.*
  We have added a third row to the figure for visibility.

- *I do not see the additional value of Fig. 9 compared to Table 2. I would see additional information, though, if the offset between the measurement result pairs would be depicted vs. the mole fraction. The same applies to Fig. 14 and Table 3.*
  Moved Fig 9 and 14 to appendix and changed them to show the mole fraction biases.

- *l. 356: The authors reflect on the source of the biases and suggest calibration being one cause. It is unclear why this is written in a bit speculative manner as the audit of the Mobile Laboratory includes a cross-comparison of the reference gases by the station instrument and the Mobile Laboratory (see l. 195). The results of this is calibration gas assignment bias is not presented - why? Were the reference standards also exchanged between GAW, WCC and Mobile Laboratory? This is data that is necessary information and should be provided in an additional table (or in a supplement).*
  Only two cross-comparisons were made: Measuring the WCC TCs with the GAW instrument and the ICOS Mobile Lab TC with the ICOS instrument.  Added tables A1 and A2 to present  the calibration cross-comparisons of GAW and ICOS calibration gases. Provided discussion in the manuscript:

  **$CO_2$**
  **To account for possible differences in the calibration standards, the ICOS Mobile Laboratory cylinders were measured with the ICOS analyzer, and the WCC travelling cylinders were measured with the GAW analyzer. The results of the measurements are presented in table A1 and table A2. The calibration cylinder measurements show that for $CO_2$ the ICOS analyzers measurements differ on average from -0.09 ppm to 0.01 ppm to the assigned cylinder values, and the GAW analyzer differs from -0.05  ppm to 0.03 ppm.**

  **$CH_4$**
  **To quantify this effect, during the audit the ICOS analyzer measured the ICOS Mobile Laboratory calibration standards, and the GAW analyzer measured the WCC travelling standards. The results of the calibrations are presented in tables A1 and  A2. For $CH_4$, the  ICOS analyzer measured 0.12 to 0.54 ppb lower values than the assigned values of the cylinder, and the GAW analyzer measured 0.16 to 0.72 ppb higher values.**

- *l. 383-384: "In the CO2 data the effect [of the Nafion dryer] is less clear, with the difference before adding the Nafion being 0.02 ppm and -0.02 ppm afterward". Fig. 6a clearly shows that a seasonal cycle of ca 0.1 ppm in the offset was much reduced after using the Nafion on the ICOS system. This has been described for CH4 (l. 332) but should also be mentioned for CO2. It is clear evidence that a problem has been remedied even if the absolute number of the offset has not changed.*
  We have added this discussion to the $CO_2$ section:
  **Before the Nafion was installed, there exists a seasonal variation in the differences between the instrument.** With the addition of the Nafion dryer to the ICOS analyzer inlet at the end of 2020**, the seasonal variation is reduced** [...]

- *l. 389-391: "the largest spread (CI range) in the differences was between ICOS and GAW for CO2 and between GAW and the ICOS Mobile Lab for CH4. This is expected, as the GAW system is the only one measuring from its own inlet". This conclusion is not fully convincing as long as the insignificance of other factors is not proven. Factors that might result in GAW measurements being slightly different could be for instance: 1) GAW measurements are also the only ones that are not made with dry air. So the application of the water vapour correction makes GAW different to all others. 2) Respective assignment errors (or instable CO2) of reference standards might result in mole fraction dependent offsets. 3) The reproducibility of the GAW system might be inferior due to less frequent calibrations. This could be easily checked by an evaluation of the target records.*
  We have added discussion on the topic.

  This is **partly** expected, as the GAW system is the only one measuring from its own inlet and all the other systems are connected to the same inlet. **In addition, the GAW instrument is still measuring the sample wet, while the other instruments are drying the sample to different degrees (ICOS instrument and WCC TI with a Nafion and the ICOS Mobile Laboratory with ICOS dryer). No significant differences in the LTR was observed for $CO_2$ or for $CH_4$ across the instruments during the audit.**

- *l. 393-394: "Compared to the differences between the ICOS and GAW analyzers over the whole period, the CO2 difference during the audit is slightly higher." It seems more reasonable to compare the audit mean difference to the mean difference since December 2020 when the Nafion has been added to the ICOS system. Else this aspects needs to be mentioned in the discussion here.*
  Changed to:
  **During the audit the difference between the ICOS and GAW instruments is comparable to the difference over the whole period, when the ICOS instrument is sampling dried air.**

- *l. 401-402: "The audit period also indicates that the ICOS system is performing slightly better". Phrase like: "The better measurement agreement with the two auditing units suggest a better performance of the ICOS system". It would be very useful to add if this corresponded to a superior reproducibility estimate for the ICOS measurements based on the record of its short term target compared to the respective target records of the GAW system.*

  Changed to:

  **The better measurement agreement with the two auditing units suggest a better performance of the ICOS system. However, the LTR of the GAW instrument is similar to the LTR of the ICOS instrument, when measuring the ICOS LT cylinders. The LTR of the ICOS ST cylinders is worse, however this could be also caused by cylinder drift.**

- *l. 403: "but the effect of theNafion dryer on the differences between the two systems is still unclear". As pointed out above (comment to l. 383) this statement is not justified. The Nafion dryer removed a stronger systematic seasonality in the offset. There might be a smaller seasonality remaining after December 2020 with the phase having shifted but it is difficult to capture this just visually from Fig. 6a. This could point to a remaining smaller humidity related offset now caused by the wet air analysis of the GAW system and also a non-perfect water vapour correction. This is something the authors should check.*

  We have added a figure A3 to show the seasonal variation in the differences before and after the Nafion was taken in use. It shows significantly reduced seasonal variation after the Nafion is installed, especially in $CH_4$ but also $CO_2$. There is more relatively more variation in the $CO_2$ measurements, especially during summer which to some extend mask the effect of the water vapor.

  Added to $CO_2$

  **There could be a remaining seasonal variation after the ICOS sample is dried, however the stronger variation during summer in $CO_2$ masks this effect.**

  Changed in Summary:

  […] addition of Nafion dryer on the intake line of the ICOS instrument. Especially for the $CH_4$ measurements the improvement is clear, the difference before drying the sample is 0.76 ppb on average and 0.21 ppb after. In the $CO_2$ measurements the effect is less **pronounced**, with the difference before […]

**New figures added:**

[Figure]

*Figure 7. CO2 target measurements of ICOS LT, ICOS ST and GAW target cylinders over the whole comparison period. Different cylinders*
*used are marked with distinct colors. The data are given as means of each sequence with the associated standard deviation.*

[Figure]

*Figure 13. CH4 target measurements of ICOS LT, ICOS ST and GAW target cylinders over the whole comparison period. Different cylinders*
*used are marked with distinct colors. The data are given as means of each sequence with the associated standard deviation*

[Figure]

*Figure 15. CH4 target measurements for ICOS, GAW, WCC and Mobile Laboratory instruments during the audit period. For ICOS both ST and LT are presented. The data are given as means of each sequence with the associated standard deviation.*

[Figure]

*Figure A2. Dependency of the CO2 difference (GAW-ICOS) on water vapor concentration (a) and mole fraction (b). The data is split into two groups: before the installation of the Nafion and after.*

[Figure]

*Figure A3. Seasonal variation of the differences (GAW-ICOS) in CO2 (a) and CH4 (b) before and after installation of the Nafion.*

[Figure]

*Figure A4. Mole fraction dependency of the difference between each comparison pair for CO2. Linear regression fitted to the data.*

[Figure]

*Figure A5. Dependency of the CH4 difference (GAW-ICOS) on water vapor concentration (a) and mole fraction (b). The data is split into two groups: before the installation of the Nafion and after.*

[Figure]

*Figure A6. Mole fraction dependency of the difference between each comparison pair for CH4. Linear regression fitted to the data.*

**Added tables:**

| | Cylinder | Purpose | LTR (ppm) | STR (ppm) | Bias (ppm) | Conc (ppm) | Nb | Measure time (min) |
|---|---|---|---|---|---|---|---|---|
| **Full period** | | | | | | | | |
| ICOS | D348367 | LT | 0.02 | 0.01 | 0.03 | 450.80 | 10 | 20 |
| | D920975 | LT | 0.01 | 0.01 | 0.03 | 461.97 | 10 | 20 |
| | D348358 | ST | 0.02 | 0.01 | -0.05 | 409.76 | 10 | 20 |
| | D348368 | ST | 0.03 | 0.01 | -0.03 | 399.71 | 10 | 20 |
| | D920974 | ST | 0.02 | 0.01 | -0.03 | 415.02 | 10 | 20 |
| GAW | D489481 | | 0.02 | 0.01 | 0.01 | 418.30 | 9 | 18 |
| | D489486 | | 0.01 | 0.01 | 0.04 | 396.61 | 9 | 18 |
| | D489487 | | 0.02 | 0.01 | 0.01 | 411.07 | 9 | 18 |
| **Audit period** | | | | | | | | |
| ICOS | D348367 | LT | 0.02 | 0.01 | 0.05 | 450.80 | 10 | 20 |
| ICOS | D348368 | ST | 0.02 | 0.01 | -0.04 | 414.64 | 10 | 20 |
| GAW | D489487 | | 0.01 | 0.01 | 0.03 | 411.07 | 9 | 18 |
| WCC | 180318_FF61508 | | 0.01 | 0.01 | -0.05 | 417.57 | 4 | 9 |
| MobileLab | D748303 | | 0.01 | 0.01 | 0.02 | 411.94 | 8 | 20 |

**Table 3.** Results of the target measurements of $CO_2$ for each instrument for the full period of comparisons between ICOS and GAW instruments and for the audit period. Nb refers to the number of data points used for averaging and measure time is the total time each cylinder is measured during one injection.

| | Cylinder | Purpose | LTR (ppb) | STR (ppb) | Bias (ppb) | Conc (ppb) | Nb | Measure time (min) |
|---|---|---|---|---|---|---|---|---|
| **Full period** | | | | | | | | |
| ICOS | D348367 | LT | 0.18 | 0.11 | -0.03 | 2097.11 | 10 | 20 |
| | D920975 | LT | 0.14 | 0.10 | -0.01 | 2196.16 | 10 | 20 |
| | D348358 | ST | 0.24 | 0.11 | -0.31 | 1949.49 | 10 | 20 |
| | D348368 | ST | 0.30 | 0.12 | -0.24 | 1948.65 | 10 | 20 |
| | D920974 | ST | 0.22 | 0.10 | -0.04 | 1936.69 | 10 | 20 |
| GAW | D489481 | | 0.21 | 0.11 | -0.12 | 1961.42 | 9 | 18 |
| | D489486 | | 0.15 | 0.10 | -0.01 | 1686.86 | 9 | 18 |
| | D489487 | | 0.18 | 0.10 | -0.06 | 1938.79 | 9 | 18 |
| **Audit period** | | | | | | | | |
| ICOS | D348367 | LT | 0.14 | 0.11 | -0.06 | 2097.11 | 10 | 20 |
| ICOS | D348368 | ST | 0.20 | 0.11 | -0.12 | 1974.51 | 10 | 20 |
| GAW | D489487 | | 0.17 | 0.10 | -0.04 | 1938.79 | 9 | 18 |
| WCC | 180318_FF61508 | | 0.18 | 0.09 | <0.01 | 1963.81 | 4 | 9 |
| MobileLab | D748303 | | 0.25 | 0.10 | 0.23 | 1937.38 | 8 | 20 |

**Table 5.** Results of the target measurements of $CH_4$ for each instrument for the full period of comparisons between ICOS and GAW instruments and for the audit period. Nb refers to the number of data points used for averaging and measure time is the total time each cylinder is measured during one injection.

|  | CAL 1 | CAL 2 | CAL 3 |
|---|---|---|---|
| $CO_2$, assigned [ppm] | 379.24 | 414.46 | 449.39 |
| $CO_2$, measured [ppm] | 379.21 | 414.37 | 449.40 |
| $\Delta CO_2$ [ppm] | -0.03 | -0.09 | 0.01 |
|  |  |  |  |
| $CH_4$, assigned [ppb] | 1985.48 | 1799.53 | 2210.77 |
| $CH_4$, measured [ppb] | 1985.36 | 1798.99 | 2210.63 |
| $\Delta CH_4$ [ppb] | -0.12 | -0.54 | -0.14 |

**Table A1.** Cross-calibration of the ICOS Mobile Laboratory calibration standards with the Pallas ICOS analyzer: The assigned values of the cylinders, average measured values with the GAW analyzer, and the difference of measured value to the assigned value.

|  | CAL 1 | CAL 2 | CAL 3 | CAL 4 | CAL 5 | CAL 6 | CAL 7 |
|---|---|---|---|---|---|---|---|
| $CO_2$, assigned [ppm] | 378.12 | 387.39 | 406.99 | 411.21 | 417.53 | 412.70 | 427.81 |
| $CO_2$, measured [ppm] | 387.15 | 387.39 | 407.01 | 411.18 | 417.48 | 412.67 | 427.80 |
| $\Delta CO2$ [ppm] | 0.03 | 0.00 | 0.02 | -0.03 | -0.05 | -0.03 | -0.01 |
|  |  |  |  |  |  |  |  |
| $CH_4$, assigned [ppb] | 1883.44 | 1890.78 | 1933.20 | 1953.82 | 1963.81 | 1998.97 | 2191.22 |
| $CH_4$, measured [ppb] | 1884.16 | 1891.23 | 1933.69 | 1954.13 | 1964.16 | 1999.13 | 2191.50 |
| $\Delta CH_4$ [ppb] | 0.72 | 0.45 | 0.49 | 0.31 | 0.35 | 0.16 | 0.28 |

**Table A2.** Cross-calibration of the GAW travelling standards with the Pallas GAW analyzer: The assigned values of the cylinders, average measured values with the GAW analyzer, and the difference of measured value to the assigned value.